# ROBUSTNESS OF QUANTUM ALGORITHMS FOR NONCONVEX OPTIMIZATION

**Weiyuan Gong**[*]
School of Engineering and Applied Science
Harvard University
Allston, MA 02134
`wgong@g.harvard.edu`

**Chenyi Zhang**[*]
Computer Science Department
Stanford University
Stanford, CA 94305
`chenyiz@stanford.edu`

**Tongyang Li**[†]
Center on Frontiers of Computing Studies
and School of Computer Science
Peking University
Beijing, 100871
`tongyangli@pku.edu.cn`

## ABSTRACT

In this paper, we systematically study quantum algorithms for finding an $\epsilon$-approximate second-order stationary point ($\epsilon$-SOSP) of a $d$-dimensional nonconvex function, a fundamental problem in nonconvex optimization, with noisy zeroth- or first-order oracles as inputs. We first prove that, up to noise of $O(\epsilon^{10}/d^5)$, perturbed accelerated gradient descent equipped with quantum gradient estimation takes $O(\log d/\epsilon^{1.75})$ quantum queries to find an $\epsilon$-SOSP. We then prove that standard perturbed gradient descent is robust to the noise of $O(\epsilon^6/d^4)$ and $O(\epsilon/d^{0.5+\zeta})$ for any $\zeta > 0$ on the zeroth- and first-order oracles, respectively, which provides a quantum algorithm with poly-logarithmic query complexity. Furthermore, we propose a stochastic gradient descent algorithm using quantum mean estimation on the Gaussian smoothing of noisy oracles, which is robust to $O(\epsilon^{1.5}/d)$ and $O(\epsilon/\sqrt{d})$ noise on the zeroth- and first-order oracles, respectively. The quantum algorithm takes $O(d^{2.5}/\epsilon^{3.5})$ and $O(d^2/\epsilon^3)$ queries to the two oracles, giving a polynomial speedup over the classical counterparts. As a complement, we characterize the domains where quantum algorithms can find an $\epsilon$-SOSP with poly-logarithmic, polynomial, or exponential number of queries in $d$, or the problem is information-theoretically unsolvable even with an infinite number of queries. In addition, we prove an $\Omega(\epsilon^{-12/7})$ lower bound on $\epsilon$ for any randomized classical and quantum algorithm to find an $\epsilon$-SOSP using either noisy zeroth- or first-order oracles.

---

[*]Equal contribution.
[†]Corresponding author.

# 1 INTRODUCTION

Optimization theory is a central topic in computer science and applied mathematics, with wide applications in machine learning, operations research, statistics, and many other areas. Currently, various quantum algorithms for optimization have been proposed, ranging from linear programs (Casares & Martin-Delgado, 2020; Rains, 1999) and semidefinite programs (Brandão & Svore, 2017; Brandão et al., 2019; van Apeldoorn & Gilyén, 2019; van Apeldoorn et al., 2020b) to general convex optimization (Chakrabarti et al., 2020; van Apeldoorn et al., 2020a; Sidford & Zhang, 2023) and non-convex optimization (Zhang et al., 2021; Liu et al., 2023; Leng et al., 2023; Chen et al., 2023).

A crucial factor of quantum optimization algorithms is their robustness. On the one hand, current quantum applications suffer from noises generated by near-term quantum devices (Preskill, 2018), which may create adversarial perturbations in the worst-case that result in disastrous failures. To address this issue, certain quantum algorithms or their components—such as some adiabatic quantum algorithms (Childs et al., 2001), quantum gates (Harrow & Nielsen, 2003), and machine learning algorithms (Liu et al., 2021; Cross et al., 2015; Lu et al., 2020)—are specifically designed to be robust against experimental noise or noisy quantum queries (Buhrman et al., 2007). An alternative solution is to develop error correction (Gottesman, 1997) or error mitigation (Endo et al., 2018; 2021) mechanisms to reduce the influences of experimental noises. In the context of nonconvex optimization, developing robust quantum algorithms is essential for future practical implementations of these algorithms on near-term devices.

On the other hand, robustness is a natural and crucial requirement for solving classical optimization problems. For instance, in statistical machine learning, we are given data drawn from an underlying probability distribution $\mathcal{D}$ (i.e., the population), and the goal is to optimize an objective function known as the *population risk* $F$, defined as

$$F(\theta) = \mathbb{E}_{\mathbf{z} \sim \mathcal{D}}[L(\theta; \mathbf{z})], \tag{1}$$

where the expectation is taken over all possible continuous loss functions $\{L(\cdot; \mathbf{z})\}$ with $\mathbf{z} \sim \mathcal{D}$. Since direct access to $F$ is unavailable, it is commonly approximated by the *empirical risk function* $f(\theta) = \frac{1}{n} \sum_{i=1}^{n} L(\theta; \mathbf{z}_i)$, which is computed from a finite set of $n$ samples. The task of optimizing $F$ using access to its empirical counterpart $f$ is known as *empirical risk minimization* (Belloni et al., 2015; Jin et al., 2018a; Vapnik, 1991). In this setting, noisy evaluations of $F$ can give rise to poorly behaved landscapes, potentially containing an exponential number of shallow local minima—even when $F$ itself exhibits favorable properties such as smoothness or Lipschitz continuity (Auer et al., 1995; Brutzkus & Globerson, 2017).

Based on this observation, a line of prior works on classical optimization (see e.g. Bartlett & Mendelson (2002); Boucheron et al. (2013)) studies the setting where we have access to a function $f$ that is pointwise close to the actual objective function $F$,

$$\|F - f\|_{\infty} \leq \nu, \tag{2}$$

where the error $\nu$ usually decays with the number of samples. Under this assumption, $f$ may still be non-smooth and contain additional shallow local minima independent of $F$. Nevertheless, the pointwise closeness between $f$ and $F$ can be leveraged to escape highly suboptimal local minima that exist only in $f$ and to approximate a local minimum of $F$.

Another related setting involves finding local minima of $F$ using empirical first-order information (Jin et al., 2018a). Similar to Eq. 2, we assume access to a noisy gradient $\nabla f$ that remains uniformly close to the actual gradient $\nabla F$. This model is widely used in stochastic settings, where gradient estimations are obtained through a sampling procedure using zeroth-order function values. A well-known example is stochastic gradient descent (Jin et al., 2021; Sun, 2019), in which an approximate gradient is obtained by sampling mini-batch function values. As the mini-batch size $m$ increases, the gradient estimate converges to the true gradient with high probability:

$$\|\nabla F - \nabla f\|_{\infty} \leq \tilde{\nu}, \tag{3}$$

where the error $\tilde{\nu}$ typically decreases with the mini-batch size $m$. Here, $\|\nabla F - \nabla f\|_{\infty}$ denotes the maximum infinity norm taken over both the input $\mathbf{x}$ and the $d$ entries of the gradient at $\mathbf{x}$.

Various approaches (Belloni et al., 2015; Zhang et al., 2017; Jin et al., 2018a; Risteski & Li, 2016; Singer & Vondrák, 2015; Karabag et al., 2021; Roy et al., 2020; Zhang et al., 2022) have been

developed to analyze the robustness of optimization algorithms from different perspectives. For example, Belloni et al. (2015) proposed an algorithm for finding an $\epsilon$-approximate minimum of an approximate convex function, where $\epsilon$ is the precision guarantee for the optimization output (see Assumption 1.1 and Assumption 1.2 for the formal definition). This algorithm requires $\tilde{O}(d^{7.5}/\epsilon^2)$[1] queries to the stochastic noisy function evaluation oracle, which has zero-mean and sub-Gaussian distributed noise. More recently, Li & Zhang (2022) developed a quantum algorithm with query complexity $\tilde{O}(d^5/\epsilon)$ for the same task, giving a polynomial quantum speedup. In another direction, Singer & Vondrák (2015) established an information-theoretic lower bound for any convex optimization algorithm that seeks minima within an $\epsilon$ multiplicative error using noisy function evaluation oracles. In the same setting, Risteski & Li (2016) developed an algorithm whose dependence on $d$ matches this lower bound.

In nonconvex optimization, Chen et al. (2020); Zhang et al. (2017) developed efficient classical algorithms to escape noise-induced "shallow" local minima using simulated annealing and stochastic gradient Langevin dynamics (SGLD). These algorithms assume access to an oracle with bounded noise $\nu \leq O(\epsilon^2/d^8)$, where $\epsilon$ is the precision and $d$ is the dimension of $F$. More recently, Jin et al. (2018a) proposed algorithms for escaping saddle points using a polynomial number of queries to zeroth- or first-order noisy oracles, with bounded noise of order $O(\epsilon^{1.5}/d)$ and $O(\epsilon/\sqrt{d})$, respectively. On the quantum side, Zhang & Li (2021); Childs et al. (2022) developed quantum algorithms for escaping saddle points via quantum simulation. Their approach leverages a precise quantum evaluation oracle and applies Jordan's algorithm to compute the gradient exponentially faster than classical algorithms using classical evaluation oracles. While quantum evaluation oracles inherently suffer from both empirical loss and experimental noise in physical implementations, the study of nonconvex optimization using quantum algorithms with noisy evaluation oracles remains largely unexplored. This motivates us to systematically investigate the robustness of quantum algorithms in escaping saddle points under such noise. Moreover, we discuss the noise threshold to guarantee the existence of polynomial-query classical algorithms for finding an $\epsilon$-approximate local minimum of $F$.

**Nonconvex Optimization with Noisy Oracle.** In this work, we consider a twice-differentiable target function $F\colon \mathbb{R}^d \to \mathbb{R}$ satisfying

- $F$ is $B$-bounded: $\sup_{\mathbf{x} \in \mathbb{R}^d} |F(\mathbf{x})| \leq B$;
- $F$ is $\ell$-smooth ($\ell$-gradient Lipschitz): $\|\nabla F(\mathbf{x}_1) - \nabla F(\mathbf{x}_2)\| \leq \ell \|\mathbf{x}_1 - \mathbf{x}_2\|$, $\forall \mathbf{x}_1, \mathbf{x}_2 \in \mathbb{R}^d$;
- $F$ is $\rho$-Hessian Lipschitz: $\left\| \nabla^2 F(\mathbf{x}_1) - \nabla^2 F(\mathbf{x}_2) \right\| \leq \rho \|\mathbf{x}_1 - \mathbf{x}_2\|$, $\forall \mathbf{x}_1, \mathbf{x}_2 \in \mathbb{R}^d$.

The goal is to find an $\epsilon$-approximate second-order stationary point ($\epsilon$-SOSP)[2] such that

$$\|\nabla F(\mathbf{x})\| \leq \epsilon, \qquad \lambda_{\min}(\nabla^2 F(\mathbf{x})) \geq -\sqrt{\rho\epsilon}. \tag{4}$$

Instead of directly querying $F$, we assume access to a noisy function $f$ that is pointwise close to $F$.

**Assumption 1.1** (Noisy evaluation query). The target function $F$ is $B$-bounded, $\ell$-smooth, and $\rho$-Hessian Lipschitz, and we can query a noisy function $f$ that is $\nu$-pointwise close to $F$:

$$\|F - f\|_\infty \leq \nu. \tag{5}$$

Moreover, we consider the problem of finding an $\epsilon$-SOSP of $F$ with an alternative assumption, where the gradient $\nabla f$ is pointwise close to $\nabla F$.

**Assumption 1.2** (Noisy gradient query). The target function $F$ is $B$-bounded, $\ell$-smooth, and $\rho$-Hessian Lipschitz, and we can query the gradient $\mathbf{g} := \nabla f$ of an $L$-smooth function $f$. The gradient of $f$ is pointwise close to gradient of $F$:

$$\|\nabla F - \nabla f\|_\infty \leq \tilde{\nu}. \tag{6}$$

---

[1] The $\tilde{O}$ notation omits poly-logarithmic terms, i.e., $\tilde{O}(g) = O(g\,\mathrm{poly}(\log g))$.

[2] A more general target is to find an $(\epsilon, \gamma)$-SOSP $\mathbf{x}$ such that $\|F(\mathbf{x})\| \leq \epsilon$ and $\lambda_{\min}(\nabla^2 F(\mathbf{x})) \geq -\gamma$. The definition of an $\epsilon$-SOSP in (4) was proposed first by Nesterov & Polyak (2006) and has been taken as a standard assumption in the subsequent papers (Jin et al., 2017; 2018b; Xu et al., 2017; 2018; Carmon et al., 2018; Agarwal et al., 2017; Tripuraneni et al., 2018; Fang et al., 2019; Jin et al., 2021; Zhang et al., 2021).

In quantum computing, the oracles are unitary operators rather than classical procedures. Under Assumption 1.1, one can query a *quantum evaluation oracle* (quantum zeroth-order oracle) $U_f$, which can be represented as

$$U_f(|\mathbf{x}\rangle \otimes |0\rangle) \to |\mathbf{x}\rangle \otimes |f(\mathbf{x})\rangle, \qquad \forall \mathbf{x} \in \mathbb{R}^d. \tag{7}$$

Furthermore, quantum oracles allow coherent *superpositions* of queries. Given $m$ vectors $|\mathbf{x}_1\rangle, \ldots,$ $|\mathbf{x}_m\rangle \in \mathbb{R}^d$ and a coefficient vector $\mathbf{c} \in \mathbb{C}^m$ such that $\sum_{i=1}^m |\mathbf{c}_i|^2 = 1$, the quantum oracle outputs $U_f(\sum_{i=1}^m \mathbf{c}_i |\mathbf{x}_i\rangle \otimes |0\rangle) \to \sum_{i=1}^m \mathbf{c}_i |\mathbf{x}_i\rangle \otimes |f(\mathbf{x}_i)\rangle$. Compared to the classical evaluation oracle, the ability to query different locations simultaneously in superposition is the essence of quantum speedup.

Similarly, in the first-order scenario we assume that one can access the *quantum gradient oracle* $U_\mathbf{g}$ under Assumption 1.2, which can be represented as

$$U_\mathbf{g}(|\mathbf{x}\rangle \otimes |\mathbf{0}\rangle) \to |\mathbf{x}\rangle \otimes |\nabla f(\mathbf{x})\rangle, \qquad \forall \mathbf{x} \in \mathbb{R}^d. \tag{8}$$

We remark that the quantum oracles defined above are natural extensions of their classical counterparts. Similar definitions of quantum oracles are widely used in prior works (Chakrabarti et al., 2020; Zhang & Li, 2021; Apeldoorn et al., 2020; Garg et al., 2021a; Zhang & Li, 2023). Notably, if a classical oracle is established by a classical arithmetic circuit (composed of additions, multiplications, divisions, etc.), it is standard to construct a quantum circuit of the same size for the arithmetic calculations (quantum additions, multiplications, divisions, etc. that enable these calculations in superposition) as the corresponding quantum oracle. Consequently, we believe our quantum algorithms hold promise to outperform black-box classical algorithms in low-dimensional settings where the oracle is given as an explicit circuit. In other words, we are not assuming the oracle for free, but we consider the same black-box optimization setting as the classical counterpart, with essentially the same cost of constructing the oracle.

**Contributions.** In this paper, we conduct a systematic study of quantum algorithms for nonconvex optimization using noisy oracles. Using zeroth- or first-order oracles as inputs, we rigorously characterize different domains where quantum algorithms can find an $\epsilon$-SOSP using poly-logarithmic, polynomial, or exponential number of queries, respectively. We identify the domain where it is information-theoretically unsolvable to find an $\epsilon$-SOSP even using an infinite number of queries.

In some of the domains, we further develop lower bounds on the query complexity for any classical algorithms, and thus establish polynomial or exponential quantum speedups compared to either the classical lower bounds or the complexities of corresponding state-of-the-art classical algorithms. We summarize our main results under Assumption 1.1 and Assumption 1.2 in Table 1 and Table 2, respectively. We (informally) introduce our results in this main body of the paper and provide the intuition for the proofs. We leave the formal version of our results, the corresponding algorithms, and the proofs to the full version of the paper (see supplementary materials).

| Noise Strength | Classical Bounds | Quantum Bounds | Speedup in $d$ |
|---|---|---|---|
| $\nu = \Omega(\epsilon^{1.5})$ | Unsolvable (Jin et al., 2018a) | Unsolvable (Theorem 3.2) | N/A |
| $\nu = O(\epsilon^{1.5}), \nu = \tilde{\Omega}(\epsilon^{1.5}/d)$ | $O(\exp(d)), \Omega(d^{\log d})$ (Jin et al., 2018a) | $\Omega(d^{\log d})$ (Theorem 3.1) | N/A |
| $\nu = O(\epsilon^{1.5}/d), \nu = \tilde{\Omega}(\epsilon^6/d^4)$ | $\tilde{O}(d^4/\epsilon^5)$ (Jin et al., 2018a; Zhang et al., 2017) | $\tilde{O}(d^{2.5}/\epsilon^{3.5})$ (Theorem 2.5) | Polynomial |
| $\nu = \tilde{O}(\epsilon^6/d^4), \nu = \tilde{\Omega}(\epsilon^{10}/d^5)$ | $\Omega(d/\log d)$ (Theorem 3.3) | $O(\log^4 d/\epsilon^2)$ (Theorem 2.3) | Exponential |
| $\nu = \tilde{O}(\epsilon^{10}/d^5)$ | $\Omega(d/\log d)^*$ (Theorem 3.3) | $O(\log d/\epsilon^{1.75})$ (Theorem 2.2) | Exponential* |

Table 1: A summary of our results and comparisons with the state-of-the-art classical upper and lower bounds under Assumption 1.1. The query complexities are highlighted in terms of the dimension $d$ and the precision $\epsilon$. ($*$) In the last row, we can obtain the desired classical lower bound and thus an exponential speedup in the query complexity when $\nu = \tilde{\Omega}(\text{poly}(1/d, \epsilon))$ as Theorem 3.3 works for $\nu = \tilde{\Omega}(\text{poly}(1/d, \epsilon))$.

## 2 UPPER BOUNDS

In this section, we propose different quantum nonconvex optimization algorithms that are robust against different levels of noise. The starting point is the quantum perturbed accelerated gradient

| Noise Strength | Classical Bounds | Quantum Bounds | Speedup in $d$ |
|---|---|---|---|
| $\tilde{\nu} = \Omega(\epsilon)$ | Unsolvable (Theorem 3.4) | Unsolvable (Theorem 3.4) | N/A |
| $\tilde{\nu} = O(\epsilon), \tilde{\nu} = \tilde{\Omega}(\epsilon/d^{0.5})$ | $\Omega(d^{\log d})$ (Theorem 3.4) | $\Omega(d^{\log d})$ (Theorem 3.4) | N/A |
| $\tilde{\nu} = \Theta(\epsilon/d^{0.5})$ | $O(d^3/\epsilon^4)$ (Jin et al., 2018a) | $O(d^2/\epsilon^3)$ (Theorem 2.6) | Polynomial |
| $\tilde{\nu} = O(\epsilon/d^{0.5+\zeta})$ | $O(\log^4 d/\epsilon^2)$ (Corollary 2.4) | $O(\log^4 d/\epsilon^2)$ (Corollary 2.4) | No |

Table 2: A summary of our results and comparisons with the state-of-the-art classical upper and lower bounds under Assumption 1.2. In the last line, $\zeta > 0$ and $\zeta = \Omega(1/\log(d))$ (for instance, this is satisfied for any constant $\zeta > 0$).

descent (PAGD) with accelerated negative curvature finding algorithm, which is inspired by the noiseless nonconvex optimization algorithm in Zhang & Li (2021). To find an $\epsilon$-SOSP of $F$ using quantum evaluation oracle specified in (7), an important step is to approximate the gradient at each iteration. An ingenious quantum approach initiated by Jordan (2005) takes a uniform mesh around the point and queries the quantum evaluation oracle (in uniform superposition) in phase using the standard phase kickback technique (Chakrabarti et al., 2020; Gilyén et al., 2019). Then by the Taylor expansion, we have

$$\sum_{\mathbf{x}} \exp(if(\mathbf{x}))\mathbf{x} \approx \sum_{\mathbf{x}} \bigotimes_{k=1}^{d} \exp\left(i\frac{\partial f}{\partial \mathbf{x}_k}\mathbf{x}_k\right)\mathbf{x}_k. \tag{9}$$

The algorithm finally recovers all the partial derivatives by a quantum Fourier transformation (QFT). We refer to Chakrabarti et al. (2020) for a precise version of Jordan's gradient estimation algorithm.

**Lemma 2.1** (Lemma 2.2, Chakrabarti et al. (2020)). *Given a target function $F$ and its noisy evaluation $f$ satisfying Assumption 1.1 with noise rate $\nu$, there exists a quantum algorithm that uses one query to the noisy oracle defined in (7) and outputs a vector $\tilde{\nabla}F(\mathbf{x})$ such that*

$$\Pr\left[\left\|\tilde{\nabla}F(\mathbf{x}) - \nabla F(\mathbf{x})\right\| \geq 400\omega d\sqrt{\nu\ell}\right] \leq \min\left\{\frac{d}{\omega-1}, 1\right\}, \qquad \forall \omega > 1. \tag{10}$$

This lemma shows that with probability at least $1 - \delta$, one can use one query to the noisy zeroth-oracle and obtain a vector $\tilde{\nabla}F(\mathbf{x})$ such that

$$\left\|\tilde{\nabla}F(\mathbf{x}) - \nabla F(\mathbf{x})\right\| \leq O(d^2\sqrt{\nu\ell}/\delta). \tag{11}$$

Our quantum perturbed accelerated gradient descent (PAGD) algorithm (see Algorithm 1 in the supplementary material for details) replaces the gradient queries in Perturbed Accelerated Gradient Descent (Zhang & Li, 2021; Zhang & Gu, 2022) with Jordan's gradient estimation. The cost of this substitution is analyzed using Lemma 2.1.

## 2.1 TINY NOISE: ROBUSTNESS OF PERTURBED ACCELERATED GRADIENT DESCENT

We consider adding tiny noise to the oracles in quantum gradient descent algorithms. In particular, we consider the function pair $(F, f)$ satisfying Assumption 1.1 and assume that one can access the function values of the noisy evaluation function $f$. We remark that $f$ may even be non-differentiable or non-smooth. In addition, the noise between $f$ and the target function $F$ might introduce additional SOSPs. Nevertheless, recent work Zhang & Gu (2022) indicates that the performance of PAGD (Jin et al., 2018b; Zhang & Li, 2021) persists when the gradients are inexact. We rigorously prove that the perturbed accelerated gradient descent algorithm with accelerated negative curvature (Zhang & Li, 2021) equipped with Jordan's algorithm for quantum gradient estimation (Jordan, 2005) is robust to the tiny noise on zeroth-order oracles as follow:

**Theorem 2.2** (Informal). *Given a target function $F$ and a noisy function $f$ satisfying Assumption 1.1 with $\nu = \Omega(\epsilon^{10}/d^5)$, there exists a quantum algorithm that finds an $\epsilon$-SOSP of $F$ with high probability using $\tilde{O}(\log d/\epsilon^{1.75})$ queries to the noisy zeroth-order oracle $U_f$.*

We leave the formal version of Theorem 2.2, the corresponding algorithm, and the proof to Section 2.1 of the supplementary material. Theorem 2.2 demonstrates that if the noise is small

enough, the impact on PAGD algorithm will not lead to an increase on the query complexity. If $\nu = \Omega(\text{poly}(\epsilon, 1/d))$, we further demonstrate that this robustness only exists for quantum algorithms by proving a polynomial lower bound in Theorem 3.3 for any classical algorithm.

## 2.2 SMALL NOISE: ROBUSTNESS OF QUANTUM GRADIENT ESTIMATION

When the strength of noise increases, the negative curvature estimation in standard PAGD will fail. In this case, we show the robustness of the gradient descent algorithm with quantum gradient estimation against the noise. We consider the function pair $(F, f)$ satisfying Assumption 1.1 when we can access noisy function $f$. Zhang et al. (2021); Chakrabarti et al. (2020) conveyed the conceptual message that perturbed gradient descent (PGD) (Jin et al., 2021) algorithm with Jordan's gradient estimation (Jordan, 2005) possesses a certain degree of robustness to noise. In this work, we formalize this intuition by presenting a concrete analysis on the convergence rate of PGD with noise, and obtain the following result:

**Theorem 2.3** (Informal). *Given a target function $F$ and a noisy function $f$ satisfying Assumption 1.1 with $\nu \leq \tilde{O}(\epsilon^6/d^4)$, there exists a quantum algorithm that finds an $\epsilon$-SOSP of $F$ with high probability using $\tilde{O}(\log^4 d/\epsilon^2)$ queries to the noisy zeroth-order oracle $U_f$.*

The formal version of Theorem 2.3, the corresponding algorithms, and the proof are given in Section 2.2 of the supplementary material. Theorem 2.3 demonstrates if the noise on the zeroth-order oracle is below a certain threshold, a quantum algorithm can find an $\epsilon$-SOSP of $F$ within a number of queries that is poly-logarithmic in terms of the dimension $d$. Similar to Theorem 2.2, this robustness only exists in quantum algorithms and provides an exponential quantum speedup in the query complexity compared to the classical counterpart. The key insight for such a speedup is as follows. While a polynomial number of queries in dimension $d$ is required for classical algorithms to compute the gradient using zeroth-order oracles, quantum zeroth-order queries enable querying function values at different positions in parallel due to quantum superposition. As a result, Jordan's algorithm for quantum gradient estimation computes the gradient using exponentially fewer queries compared to classical counterparts, and such speedup is robust to noise up to $O(\epsilon^6/d^4)$.

It seems that Theorem 2.2 and Theorem 2.3 provide exponential speed-ups only when the noise thresholds that decay as $O(\epsilon^{10}/d^5)$ or $O(\epsilon^6/d^4)$. However, we will prove later (see Theorem 3.1) that when the noise rate is larger than $\Omega(\epsilon^{1.5}/d)$, a superpolynomial query complexity lower bound of $\Omega(d^{\log d})$ queries for both classical and quantum algorithms is proved in our work. This suggests that we have to focus on the noise regime below $O(\epsilon^{1.5}/d)$ where one can find an algorithm (either classical or quantum) that only requires polynomially many queries in $d$. Compared to this threshold, the noise threshold of $O(\epsilon^{10}/d^5)$ or $O(\epsilon^6/d^4)$ where quantum algorithms have exponential speed-ups compared to classical ones is only polynomially small. We can thus conclude that the noise regime where an exponential quantum speedup is within a reasonable amplitude compared to the noise regime where polynomial classical and quantum algorithms for this task exist.

We further extend Theorem 2.3 to function pair $(F, f)$ satisfying Assumption 1.2. In particular, we prove the following corollary indicating that the classical PGD iteration is robust against the noise of $\tilde{\nu} \leq O(\epsilon/d^{0.5+\zeta})$ on the first-order gradient information, where $\zeta = \Omega(1/\log(d))$. In addition, we can use the techniques above to prove the algorithmic upper bound for function pair $(F, f)$ satisfying Assumption 1.2 with $\tilde{\nu} \leq O(\epsilon/d^{0.5+\zeta})$ for $\zeta > 0$ and $\zeta = \Omega(1/\log(d))$. We provide the following corollary corresponding to the last line in Table 2.

**Corollary 2.4.** *Given a target function $F$ and a noisy function $f$ satisfying Assumption 1.2 with $\tilde{\nu} \leq O(\epsilon/d^{0.5+\zeta})$ for $\zeta > 0$ and $\zeta = \Omega(1/\log(d))$. There exists an algorithm that outputs an $\epsilon$-SOSP of $F$ satisfying Eq. (4) using $O(\log^4 d/\epsilon^2)$ queries to $U_{\mathbf{g}}$ in (8) with high probability.*

## 2.3 INTERMEDIATE NOISE: SPEEDUP FROM QUANTUM MEAN ESTIMATION

When the strength of noise keeps increasing, the robustness of Jordan's algorithm will also fail to handle the gap between the noisy function $f$ and the target function $F$. To address this issue, we develop a quantum algorithm based on the Gaussian smoothing of $f$ inspired by Jin et al. (2018a). We consider function pairs $(F, f)$ satisfying Assumption 1.1. We sample the value $\mathbf{z}[f(\mathbf{x} + \mathbf{z}) - f(\mathbf{x})]/\sigma^2$, where $\mathbf{z} \sim \mathcal{N}(0, \sigma^2 I)$ is chosen from the standard Gaussian distribution with parameter

$\sigma^2$ (Duchi et al., 2015). We then apply quantum mean estimation to approximate the gradient from the samples of stochastic gradients. The performance of the algorithm is guaranteed as follows:

**Theorem 2.5** (informal). *Given a target function $F$ and a noisy function $f$ satisfying Assumption 1.1 with $\nu \leq O(\epsilon^{1.5}/d)$, there exists a quantum algorithm that finds an $\epsilon$-SOSP of $F$ with high probability taking $\tilde{O}(d^{2.5}/\epsilon^{3.5})$ queries to the noisy zeroth-order oracle $U_f$.*

The formal version of Theorem 2.5, the corresponding algorithms, and the proof are given in Section 3.1 of the supplementary material. Theorem 2.5 indicates that the quantum algorithm can find an $\epsilon$-SOSP of $F$ using polynomial number of queries to $f$ with bounded noise strength $\nu \leq O(\epsilon^{1.5}/d)$. As the state-of-art classical algorithm (Jin et al., 2018a) solves this problem with the same noise strength $\nu \leq O(\epsilon^{1.5}/d)$ using $O(d^4/\epsilon^5)$ queries, our algorithm provides a polynomial improvement compared to this classical result in both the dimension $d$ and the precision $\epsilon$.

We consider the problem of finding an $\epsilon$-SOSP of functions $F$ taking queries to the quantum gradient oracle in (8). Our approach leverages the fact that $\nabla f(\mathbf{x} + \mathbf{z})$ is a stochastic gradient function of the Gaussian smoothing of $f$ where $\mathbf{z} \sim \mathcal{N}(0, \sigma^2 I)$ is chosen from Gaussian distribution with parameter $\sigma^2$. Similar to the zeroth-order scenario, we apply quantum mean estimation to approximate the gradient of the Gaussian smoothing, which leads to the following algorithmic upper bound.

**Theorem 2.6** (Informal). *Given a target function $F$ and the gradient information of a noisy function $f$ satisfying Assumption 1.2 with $\tilde{\nu} \leq O(\epsilon/d^{0.5})$, there exists a quantum algorithm that finds an $\epsilon$-SOSP of $F$ with high probability using $\tilde{O}(d^2/\epsilon^3)$ queries to the noisy gradient oracle $U_\mathbf{g}$.*

The formal version of Theorem 2.6, the corresponding algorithms, and the proof are given in Section 3.2 of the supplementary material. The tolerance on $\tilde{\nu}$ and the query complexity is larger compared to Theorem 2.5, where we access a zeroth-order oracle. The best-known classical algorithm finding an $\epsilon$-SOSP requires $O(d^3/\epsilon^4)$ queries. Hence, this quantum algorithm also provides a polynomial reduction in the sample complexity compared to the classical result.

## 3 LOWER BOUNDS

| Input Oracle | Noise Strength | Deterministic Classical Lower Bounds | Randomized Classical and Quantum Lower Bounds |
|---|---|---|---|
| Zeroth-order | $\nu = 0$ | | N/A |
| Zeroth-order | $\nu = \Omega(\epsilon^{-16/7}/d)$ | $\Omega(\epsilon^{-12/7})$ | $\Omega(\epsilon^{-12/7})$ (Theorem 3.5) |
| First-order | $\tilde{\nu} = 0$ | (Carmon et al., 2021) | N/A |
| First-order | $\tilde{\nu} = \Omega(\epsilon^{-8/7}/\sqrt{d})$ | | $\Omega(\epsilon^{-12/7})$ (Theorem 3.5) |

Table 3: A summary of our results on classical and quantum query complexity lower bounds in $\epsilon$ under Assumption 1.1 or Assumption 1.2, respectively. The query complexities are highlighted in terms of the dimension $d$ and the precision $\epsilon$.

### 3.1 LARGE NOISE: QUANTUM QUERY COMPLEXITY LOWER BOUND IN $d$

In this work, we also provide lower bounds concerning $d$ on the query complexity required for any classical and quantum algorithms under Assumption 1.1 and Assumption 1.2. In particular, we construct a hard instance inspired by Jin et al. (2018a) (as shown in the left of Figure 1 (a)): we define a target function $F$ in a hypercube and use the hypercube to fill the entire space $\mathbb{R}^d$. We start from a "scale free" version of function pair $(F, f)$, where we assume $\rho = 1$ and $\epsilon = 1$. Denote $\sin \mathbf{x} := (\sin(x_1), ..., \sin(x_d))$ and $\mathbb{I}(A)$ as the indicator function that has value 1 when $A$ is true and 0 otherwise. We choose some constant $\mu$ and define the target function as

$$F(\mathbf{x}) := h(\sin \mathbf{x}) + \|\sin \mathbf{x}\|^2, \tag{12}$$

where $h(\mathbf{x}) := h_1(\mathbf{v}^\top \mathbf{x}) \cdot h_2\left(\sqrt{\|\mathbf{x}\|^2 - (\mathbf{v}^\top \mathbf{x})^2}\right)$ for some polynomials such that the landscape of the function is given as the right of Figure 1(a). Here, the vector $\mathbf{v}$ is uniformly chosen from the $d$-dimensional unit sphere. In addition, we can split the domain into different regions upon which analysis and constructions are made separately:

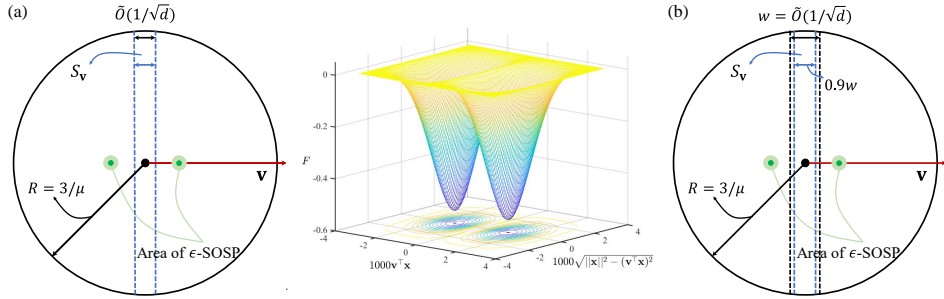

Figure 1: (a) The Sketch of the hard instance function for Theorem 3.1. The left figure illustrates the construction in the domain and the right figure shows a two-dimensional example. (b) The construction of the hard instance function for Theorem 3.4.

- "ball" $\mathbb{B}(0, 3/\mu) = \{\mathbf{x} \in \mathbb{R}^d : \|x\| \leq 3/\mu\}$ is the $d$-dimensional hyperball with radius $3/\mu$.
- "band" $S_{\mathbf{v}} = \{\mathbf{x} \in \mathbb{B}(0, 3/\mu) : \langle \sin \mathbf{x}, \mathbf{v} \rangle \leq \log d/\sqrt{d}\}$.

We provide the landscape of $F$ and the region division in Figure 1 (a). The above construction happens within a hyperball and we cannot fill the entire space $\mathbb{R}^d$ with hyperballs. Therefore, we embed this hyperball into a hypercube and add two regions.

- "hypercube" $H = [-\pi/2, \pi/2]^d$ is the $d$-dimensional hypercube with length $\pi$.
- "padding" $S_2 = H - \mathbb{B}(0, 3/\mu)$.

With the above construction, we can fill the space $\mathbb{R}^d$ using these hypercubes. Meanwhile, the noisy function $f$ is defined as

$$f(\mathbf{x}) = \begin{cases} \|\sin \mathbf{x}\|^2, & \mathbf{x} \in S_{\mathbf{v}}, \\ F(\mathbf{x}), & \mathbf{x} \notin S_{\mathbf{v}}. \end{cases} \tag{13}$$

The "band" region $S_{\mathbf{v}}$ is known as the non-informative region as any query to $f$ in this area will obtain no information regarding $\mathbf{v}$. Intuitively, the metric of the non-informative area approaches 1 as $d$ increases according to the measure of concentration. It is hard for any algorithm (both classical and quantum) to find a point out of this region. In particular, the probability of classically querying a point on $S_{\mathbf{v}}$ is bounded below by

$$\frac{\text{Area}(S_{\mathbf{v}})}{\text{Area}(\mathbb{B}(0, 3/\mu))} \geq 1 - O(d^{-\log d}). \tag{14}$$

By adding noise to the zeroth- or first-order oracle $f$, we can erase the information of $F$ such that a limited number of classical or quantum queries cannot find any $\epsilon$-SOSPs with high probability. For a function pair $(F, f)$ satisfying Assumption 1.1, our first result in this part is the following quasi-polynomial lower bound.

**Theorem 3.1** (Informal). *We can find functions $F$ and $f$ satisfying Assumption 1.1 with $\nu = \tilde{\Theta}(\epsilon^{1.5}/d)$ such that any quantum algorithm requires at least $\Omega(d^{\log d})$ queries to $U_f$ to find any $\epsilon$-SOSP of $F$ with high probability.*

Here, we provide the intuition. Consider a function $F$ in a hyperball $\mathbb{B}(0, r)$ and embed the hyperball into a hypercube, with which we can cover the whole space. Next, we introduce noise to create $f$ with a non-informative area around $0$ (in the sense that any query to this area will obtain no information about any SOSPs of the target function $F$). Then, we transfer this problem into an unstructured search problem. The final lower bound for nonconvex optimization is obtained by applying the quantum lower bound for unstructured search. We mention that the $\epsilon$ and the $d$ dependence for $\nu$ in Theorem 3.1 are tight up to logarithmic factors. The classical version of Theorem 3.1 is proved in Jin et al. (2018a). The parallelism in quantum algorithms possesses the potential to

query different points in superposition. However, Theorem 3.1 demonstrates that the same query complexity lower bound holds even for quantum algorithms.

If the noise $\nu$ keeps increasing, we can further prove the following lower bound that prevents any quantum algorithm from finding any $\epsilon$-SOSPs of target function $F$:

**Theorem 3.2** (Informal). *For any quantum algorithm, there exists a pair of functions $(F, f)$ satisfying Assumption 1.1 with $\nu = \tilde{\Theta}(\epsilon^{1.5})$ such that it will fail, with large probability, to find any $\epsilon$-SOSP of $F$ given access to $f$.*

Despite the quantum lower bound, we also propose a classical lower bound concerning nonconvex optimization using zeroth-order oracle with noise strength $\nu = O(1/\operatorname{poly}(d))$.

**Theorem 3.3.** *For any $\epsilon \leq \epsilon_0 < 1$, where $\epsilon_0$ is some constant, there exists a function pair $(F, f)$ satisfying Assumption 1.1 with $\nu = \Omega(1/\operatorname{poly}(d))$, such that any classical algorithm that outputs an $\epsilon$-SOSP of $F$ with high probability requires at least $\Omega(d/\log d)$ classical queries to $f$.*

We prove Theorem 3.3 using an information-theoretic argument inspired by Chakrabarti et al. (2020). Theorem 2.2, Theorem 2.3, and Theorem 3.3 establish the exponential separation between classical and quantum query complexities required for nonconvex optimization using oracles with noise $\nu = \tilde{\Omega}(\operatorname{poly}(\epsilon, 1/d))$. This separation originates from the Jordan's gradient estimation algorithm (Jordan, 2005). Classically, querying the evaluation oracle can only provide information at one point. Quantumly, however, one can take the superposition on different points and query the quantum evaluation oracle in parallel (Gilyén et al., 2019; Chakrabarti et al., 2020).

Moreover, we can extend the above lower bound to function pairs $(F, f)$ satisfying Assumption 1.2. If the noise increases by even a factor that is logarithmic in $d$ from $\tilde{\Theta}(\epsilon/d^{0.5})$, we can prove an exponential lower bound for any classical or quantum algorithm through a similar construction of hard instance used in Theorem 3.1 (as shown in Figure 1 (b)). Unlike Theorem 3.1, we cannot directly apply the hard instance $(F, f)$ defined above because $f$ is not differentiable (or more strictly, not continuous). To address this problem, we construct a different noisy function $f$. For simplicity, we focus on the "scale free" version here are leave the details in the full version of the paper. We still define the target function $F(\mathbf{x}) = h(\sin \mathbf{x}) + \|\sin \mathbf{x}\|^2$, which is the same with (12). We uniformly choose $\mathbf{v}$ and divide the "hypercube" into different regions including: "hypercube" $H = [-\pi/2, \pi/2]^d$ is the $d$-dimensional hypercube with length $\pi$; "ball" $\mathbb{B}(0, 3/\mu) = \{\mathbf{x} \in \mathbb{R}^d : \|x\| \leq 3/\mu\}$ is the $d$-dimensional ball with radius $3/\mu$; "band" $S = \{\mathbf{x} \in \mathbb{B}(0, 3/\mu) : \langle \sin \mathbf{x}, \mathbf{v} \rangle \leq w\}$ with $w = O(\log d/\sqrt{d})$; and "padding" $S_2 = H - \mathbb{B}(0, 3/\mu)$. In particular, a new region is added as

- "non-informative band" $S_{\mathbf{v}} = \{\mathbf{x} \in \mathbb{B}(0, 3/\mu) : \langle \sin \mathbf{x}, \mathbf{v} \rangle \leq 0.9w\}$.

Meanwhile, the noisy function $f$ is defined as

$$f(\mathbf{x}) = \begin{cases} \|\sin \mathbf{x}\|^2, & \mathbf{x} \in S_{\mathbf{v}}, \\ \|\sin \mathbf{x}\|^2 + g(\mathbf{x}), & \mathbf{x} \in S - S_{\mathbf{v}} \\ F(\mathbf{x}), & \mathbf{x} \notin S, \end{cases} \tag{15}$$

where $g(\mathbf{x}) = h_3(\mathbf{x}) \cdot h_2(\sqrt{\|\sin \mathbf{x}\|^2 - (\mathbf{v}^\top \sin \mathbf{x})^2})$ and $h_3(\mathbf{x}) = h_1(\mathbf{v}^\top \sin(10\mathbf{x} - 9w\mathbf{v}/2))$.

Moreover, if the noise increases to $\Omega(\epsilon)$, there exists a similar hard instance with Theorem 3.2 that prevents any classical or quantum algorithm from finding any $\epsilon$-SOSP of $F$. Formally, we can extend Theorem 3.1 and Theorem 3.2 in the context of Assumption 1.2:

**Theorem 3.4** (Informal). *We can find functions $F$ and $f$ satisfying Assumption 1.2 with $\tilde{\nu} = \tilde{\Theta}(\epsilon/d^{0.5})$ such that any quantum (classical) algorithm that finds an $\epsilon$-SOSP of $F$ with high probability requires at least $\Omega(d^{\log d})$ queries to $U_{\mathbf{g}}$. Moreover, we can find functions $F$ and $f$ satisfying Assumption 1.2 with $\tilde{\nu} = \Theta(\epsilon)$ with any quantum (classical) algorithm failing with high probability.*

### 3.2 QUANTUM QUERY COMPLEXITY LOWER BOUND IN $\epsilon$

Finally, we establish query complexity lower bounds for classical and quantum nonconvex optimization algorithms under Assumption 1.1 or Assumption 1.2, respectively (summarized in Table 3).

**Theorem 3.5** (informal). *There exists a function pair $F$ and $f$ satisfying either Assumption 1.1 or Assumption 1.2 with $\nu = \Omega(\epsilon^{-16/7}/d)$ or $\tilde{\nu} = \Omega(\epsilon^{-8/7}/\sqrt{d})$, respectively, and additionally*

$F(\mathbf{0}) - \inf_{\mathbf{x}} F(\mathbf{x}) \leq \Delta$ *for some constant* $\Delta$*, such that any classical or quantum algorithm with query complexity* $\Omega\big(\epsilon^{-12/7}\big)$ *will fail with high probability to find an* $\epsilon$*-SOSP of target function* $F$.

The intuition for Theorem 3.5 employs the hard instance inspired by Carmon et al. (2020; 2021). Previously, there have been two lower bounds concerning $\epsilon$ dependence that apply to classical algorithms for nonconvex optimization. In Carmon et al. (2020), it is proved that at least $\Omega(\epsilon^{-3/2})$ queries are required by finding $\epsilon$-SOSPs of a Hessian Lipshitz function $F$ even provided both zeroth- and first-order oracles for either random or deterministic classical algorithms. Using similar techniques, Carmon et al. (2021) further proved that deterministic classical algorithms using first-order *noiseless* oracle require $\Omega(\epsilon^{-12/7})$ queries to find an $\epsilon$-SOSP of a Hessian Lipshitz function $F$.

On the other hand, despite recent results on quantum lower bounds for optimization (Garg et al., 2021a;b; Zhang & Li, 2023), there are settings where quantum lower bounds remain unresolved. Specifically, although Zhang & Li (2023) showed that there is no quantum speedup for finding an $\epsilon$-FOSP of a $p$-th order smooth function with $p$-th order oracle for any $p$, it remains unclear whether a quantum speedup exists for finding an $\epsilon$-SOSP of a Hessian-Lipschitz function with only a gradient oracle, where the order of the function smoothness and the given oracle differ. Notably, this problem has been widely investigated in previous works on classical nonconvex optimization, see e.g., Agarwal et al. (2017); Allen-Zhu & Li (2018); Liu et al. (2018). In this paper, we fill this conceptual gap by extending the classical deterministic lower bound by Carmon et al. (2021) to all classical randomized algorithms and even quantum algorithms, given that noise exists in the function evaluation. In particular, noise allows us to construct a hard instance by creating a non-informative area around $\mathbf{0}$. According to the concentration of measure phenomenon, the non-informative area will occupy an overwhelming proportion of the whole space. Technically, our hard instance is also different from Carmon et al. (2021) in the sense that we replace its quadratic scaling function by a sine function as we have an additional $B$-bounded assumption on the function. Similar to Garg et al. (2021a;b); Zhang & Li (2023), the hard instance we construct here exhibits a similar property that, if the number of quantum queries is below a certain threshold, in expectation the output state will barely change if we replace the quantum oracle by an oracle that only encodes "partial" information of the objective function, where the missing information is crucial for any (classical or quantum) algorithm to find an $\epsilon$-SOSP of $F$. Moreover, our lower bound result in Theorem 3.5 can be extended to finding $\epsilon$-SOSPs when we waive the $B$-bounded requirement on $F$.

## 4 DISCUSSIONS

In the paper, we conduct a systematic study on the query complexity for quantum algorithms to find $\epsilon$-SOSP of a nonconvex function using noisy oracles. We prove a series of upper and lower bounds under different noise regimes for first-order and second-order oracles. Our paper leaves several open questions for future investigations:

- Can we further improve quantum algorithms for the task of nonconvex optimization using noisy oracles? Can we obtain a quantum algorithm with better dependence on $d$ and $\epsilon$ compared to Theorem 2.5?

- Can we characterize optimal quantum query complexity for nonconvex optimization? We would like to point out that for all noise ranges, nonconvex optimization in the classical setting does not have optimal bounds either, as shown in Table 1 and Table 2. As a result, proving optimal quantum query complexity bounds may solicit more techniques and hard instances. It is helpful to investigate sublinear or poly-logarithmic quantum lower bounds in dimension $d$ on general optimization problems using either noiseless or noisy oracles.

- We employ a simple model on the noise: only the upper bound of noise strength is considered. In general, can we demonstrate the robustness and speedups for nonconvex optimization algorithms analytically under other noise assumptions (say, more practical quantum noise models or stochastic noise models)?

- A further question one may ask is the quantum version of minimax optimal query complexities. Although this is better understood in classical information theory, in the quantum setting this is widely open in general. Some existing results can calculate the success probability of quantum algorithms when we have a fixed number of quantum queries (Zhandry, 2015; 2021), but their minimax properties are still open.

ACKNOWLEDGEMENT

We thank Yizhou Liu for helpful discussions about the quantum tunneling walk in Liu et al. (2023). TL was supported by the National Natural Science Foundation of China (Grant Numbers 92365117 and 62372006), and also the Tianyan Quantum Computing Program.

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
