papers (Garg et al., 2021a;b) studying quantum lower bounds on convex optimization, quantum lower bounds on nonconvex optimization are still widely open. In this paper, we fill this conceptual gap by extending the classical deterministic lower bound (Carmon et al., 2021) to all classical randomized algorithms and even quantum algorithms, given that noise exists in the function evaluation. In particular, noise allows us to construct a hard instance by creating a non-informative area around $\mathbf{0}$. According to the concentration of measure phenomenon, the non-informative area will occupy an overwhelming proportion of the whole space. Although its intuition and structure are different from the hard instance in Refs. (Garg et al., 2021a;b) constructed via performing maximization, the hard instance we construct here exhibits a similar property that, if the number of quantum queries is below a certain threshold, in expectation the output state will barely change if we replace the quantum oracle by an oracle that only encodes "partial" information of the objective function, where the missing information is crucial for any (classical or quantum) algorithm to find an $\epsilon$-SOSP of $F$.

Moreover, we note that our lower bound result in Theorem 1.11 can be extended to the case where the goal is merely to find an $\epsilon$-SOSP if we waive the $B$-bounded requirement on $F$, which may be of independent interest.

## 1.3 DISCUSSION AND OPEN QUESTIONS

In conclusion, we conduct a systematic study of quantum algorithms for nonconvex optimization using noisy oracles. We would like to provide two meaningful contexts for the noise levels considered in our work from classical optimization and quantum computing perspectives, respectively.

From the perspective of classical optimization, nonconvex optimization arises naturally when we study the landscape of the loss functions for training neural networks with a bounded number of data samples. In practice, loss functions are usually nonconvex and nonsmooth. However, according to the PAC learning framework, the loss function is approximately smooth and its distance (i.e. the empirical noise level) from a smooth function is proportional to the inverse of the number of data samples. In particular, roughly $O(d/\epsilon^{1.5})$ samples can guarantee empirical risk of amplitude $O(\epsilon^{1.5}/d)$ for dimension $d$ and accuracy parameter $\epsilon$. By varying the dependence of the sample size on the dimension $d$ and accuracy parameter $\epsilon$, we can generate empirical noise of different levels of $\mathrm{poly}(d, \mathbf{1}/\epsilon)$. Understanding the complexity of optimizing approximately smooth nonconvex functions of different noise rates of $\mathrm{poly}(d, \mathbf{1}/\epsilon)$ offers insights into the necessary data sample number for training neural networks.

From the perspective of quantum computing, the realization of future quantum computers heavily depends on achieving fault-tolerant quantum computation. A standard approach involves con-

catenating multiple levels of quantum error-correcting codes. The number of concatenation levels typically scales poly-logarithmically with the target noise rate, leading to an inverse polynomial overhead in the noise rate. For a noise rate of order $\mathrm{poly}(d, \mathbf{1}/\epsilon)$, achieving fault tolerance requires $\mathrm{poly}(d, \mathbf{1}/\epsilon)$ levels of concatenation. This overhead is generally considered practical. However, it can vary significantly depending on the specific noise level within this regime. Understanding the effects of different noise levels is crucial for optimizing the number of concatenation levels and reducing experimental resource requirements.

Our paper leaves several open questions for future investigations:

- Can we give quantum algorithms for the task of nonconvex optimization with better performance using noisy oracles? For instance, can we obtain a quantum algorithm with better dependence on $d$ and $\epsilon$ compared to Theorem 1.5?

- Can we establish optimal quantum query complexity bounds for nonconvex optimization? We would like to point out that for all noise ranges, nonconvex optimization in the classical setting does not have optimal bounds neither, as shown in Table 1 and Table 2. As a result, proving optimal quantum query complexity bounds may solicit more techniques and hard instances. It might be helpful to investigate sublinear or poly-logarithmic quantum lower bounds in dimension $d$ on general optimization problems using either noiseless or noisy oracles.

- We employ a simple model on the noise: only the upper bound of noise strength is considered. In general, can we demonstrate the robustness and speedups for nonconvex optimization algorithms analytically under other noise assumptions (say, more practical quantum noise models or stochastic noise models)?

- A further question one may ask is the quantum version of minimax optimal query complexities. Although this is better understood in classical information theory, in the quantum setting this is widely open in general. Some existing results can calculate the success probability of quantum algorithms when we have a fixed number of quantum queries (Zhandry, 2015; 2021), but their minimax properties are still open.

## 1.4 ORGANIZATION

The rest of the paper is organized as follows:

- In Section 2, we prove the robustness for the standard gradient-based algorithms. In particular, we consider the tiny noise case and prove the robustness of PAGD equipped with quantum gradient estimation in Section 2.1. In Section 2.2, we consider the small noise case and prove the robustness of standard PGD equipped with quantum gradient estimation.

- In Section 3, we consider the intermediate noise case and propose the stochastic gradient descent algorithm using Gaussian smoothing and quantum mean estimation, which provides a polynomial speedup compared to classical algorithms under Section 3.1 and Section 3.2, respectively.

- In Section 4, we prove lower bounds concerning dimension $d$ for classical and quantum algorithms under different noise strengths. Specifically, in Section 4.1 and Section 4.2, we prove the existence of hard instances under Assumption 1.1 for any (polynomial) quantum algorithm when $\nu = \tilde{\Theta}(\epsilon^{1.5}/d)$ ($\nu = \tilde{\Theta}(\epsilon^{1.5})$). In Section 4.3, we prove the $\Omega(d/\log d)$ classical query complexity lower bound using zeroth-order oracles with $\nu = \Omega(1/\mathrm{poly}(d))$. We prove the lower bound under Assumption 1.2 in Section 4.4.

- In Section 5, we prove lower bounds concerning the precision $\epsilon$ for both (possibly randomized) classical algorithms and quantum algorithms.

- In the appendices, we introduce necessary existing tools for our proofs in Appendix A. Technical lemmas for the main text are given in Appendix B. Additional information and extended discussions on PGD equipped with quantum simulation and quantum tunneling walk are provided in Appendix C and Appendix D, respectively.

## 2 ROBUSTNESS OF QUANTUM AND CLASSICAL ALGORITHMS WITH SMALL NOISE

In this section, we propose two quantum nonconvex optimization algorithms that are robust for tiny noise and small noise, respectively. These algorithms find to an $\epsilon$-SOSP of $F$ using only polylogarithmic queries to noisy empirical function $f$.

### 2.1 ROBUSTNESS OF CLASSICAL PERTURBED ACCELERATED GRADIENT DESCENT WITH TINY NOISE

To begin with, we introduce the quantum perturbed accelerated gradient descent (PAGD) with accelerated negative curvature finding algorithm, which is inspired by the noiseless nonconvex optimization algorithm in Ref. (Zhang & Li, 2021). To find an $\epsilon$-SOSP of $F$ using quantum evaluation oracle specified in (7), an important step is to approximate the gradient at each iteration. An ingenious quantum approach initiated by Ref. (Jordan, 2005) takes a uniform mesh around the point and queries the quantum evaluation oracle (in uniform superposition) in phase using the standard phase kickback technique (Chakrabarti et al., 2020; Gilyén et al., 2019). Then by the Taylor expansion, we have

$$\sum_{\mathbf{x}} \exp(if(\mathbf{x}))\mathbf{x} \approx \sum_{\mathbf{x}} \bigotimes_{k=1}^{d} \exp\left(i\frac{\partial f}{\partial \mathbf{x}_k}\mathbf{x}_k\right)\mathbf{x}_k. \tag{9}$$

The algorithm finally recovers all the partial derivatives by applying a quantum Fourier transformation (QFT). We refer to Ref. (Chakrabarti et al., 2020) for a precise version of Jordan's gradient estimation algorithm with the following performance guarantee:

**Lemma 2.1** (Lemma 2.2, Ref. (Chakrabarti et al., 2020)). *Given a target function $F$ and its noisy evaluation $f$ satisfying Assumption 1.1 with noisy rate $\nu$, there exists a quantum algorithm that uses one query to the noisy oracle defined in (7) and outputs a vector $\tilde{\nabla} F(\mathbf{x})$ such that*

$$\Pr\left[\left\|\tilde{\nabla} F(\mathbf{x}) - \nabla F(\mathbf{x})\right\| \geq 400\omega d\sqrt{\nu\ell}\right] \leq \min\left\{\frac{d}{\omega-1}, 1\right\}, \qquad \forall \omega > 1. \tag{10}$$

This lemma indicates that with probability at least $1 - \delta$, one can use one query to the noisy zeroth-oracle and obtain a vector $\tilde{\nabla} F(\mathbf{x})$ such that

$$\left\|\tilde{\nabla} F(\mathbf{x}) - \nabla F(\mathbf{x})\right\| \leq O(d^2\sqrt{\nu\ell}/\delta). \tag{11}$$

Now, we are ready to introduce our first algorithm as shown in Algorithm 1. This algorithm replaces the gradient queries in Perturbed Accelerated Gradient Descent (Zhang & Li, 2021; Zhang & Gu, 2022) with Jordan's gradient estimation in Lemma 2.1. The negative curvature exploitation (NCE) subroutine as shown in Algorithm 2 is applied if the following condition holds.

$$f(\mathbf{x}_t) \leq f(\mathbf{y}_t) + \left\langle \tilde{\nabla} F(\mathbf{y}_t), \mathbf{x}_t - \mathbf{y}_t \right\rangle - \frac{\gamma}{2}\|\mathbf{x}_t - \mathbf{y}_t\|^2. \tag{12}$$

The intuition for NCE (Algorithm 2) will be discussed later.

We prove that Algorithm 1 has the following performance guarantee:

**Theorem 2.2** (Formal version of Theorem 1.3). *Consider a target function $F$ and its noisy evaluation $f$ satisfying Assumption 1.1 with $\nu \leq \tilde{O}(\delta^2\epsilon^{10}/d^5)$. Algorithm 1 can find an $\epsilon$-SOSP of $F$ satisfying Eq. (4) with probability at least $1 - \delta$, using*

$$\tilde{O}\left(\frac{\ell B}{\epsilon^{1.75}} \cdot \log d\right) \tag{13}$$

*queries to $U_f$ defined in (7), under the following parameter choices:*

$$\eta = \frac{1}{4\ell}, \quad \theta = \frac{1}{4\sqrt{\kappa}}, \quad \gamma = \frac{\theta^2}{\eta}, \quad s = \frac{\gamma}{4\rho}, \quad \delta_0 = \frac{\delta\epsilon^{1.75}}{c_\delta \ell B} \cdot \log d, \tag{14}$$

$$r = \frac{\delta_0\epsilon}{c_r}\sqrt{\frac{\pi}{\rho d}}, \quad \mathcal{T} = c_r\sqrt{\kappa}\log\left(\frac{\ell\sqrt{d}}{\delta_0\sqrt{\rho\epsilon}}\right), \quad \mathcal{F} = \sqrt{\frac{\epsilon^3}{\rho}}c^{-7}, \tag{15}$$

*where $c$, $c_r$, and $c_\delta$ are some large enough constants, and $\kappa = \ell/\sqrt{\rho\epsilon}$.*

---

**Algorithm 1:** Perturbed Accelerated Gradient Descent with Accelerated Negative Curvature Finding and Quantum Gradient Computation

---

**Input:** $\mathbf{x}_0$, learning rate $\eta$, noise ratio $r$, parameters $\mathscr{T}$, $\iota$, $\theta$, $\gamma$, and $s$ to be fixed later.

1: $t_{\text{perturb}} \leftarrow -\mathscr{T} - 1$, $\mathbf{y}_0 \leftarrow \mathbf{x}_0$, $\tilde{\mathbf{x}} \leftarrow \mathbf{x}_0$, and $\iota \leftarrow \mathbf{0}$
2: **for** $t = 0, 1, \ldots, T$ **do**
3:     Apply Lemma 2.1 to compute an estimation $\tilde{\nabla}F(\mathbf{x})$ of $\nabla F(\mathbf{x})$
4:     **if** $\left\|\tilde{\nabla}F(\mathbf{x})\right\| \leq 3\epsilon/4$ and $t - t_{\text{perturb}} > \mathscr{T}$ **then**
5:         $\tilde{\mathbf{x}} \leftarrow \mathbf{x}_t$
6:         $\mathbf{x}_t = \tilde{\mathbf{x}} + \xi_t$
7:         $\mathbf{y}_t = \mathbf{x}_t$, $\iota = \tilde{\nabla}F(\tilde{\mathbf{x}})$, $t_{\text{perturb}} \leftarrow t$
8:     **end if**
9:     **if** $t_{\text{perturb}} \neq -\mathscr{T} - 1$ and $t - t_{\text{perturb}} = \mathscr{T}$ **then**
10:         $\hat{\mathbf{e}} \leftarrow (\mathbf{x}_t - \tilde{\mathbf{x}})/\|\mathbf{x}_t - \tilde{\mathbf{x}}\|$
11:         $\mathbf{x}_t \leftarrow \arg\min_{\mathbf{x} \in \{\tilde{\mathbf{x}} - \frac{1}{4}\sqrt{\frac{\epsilon}{\rho}}\hat{\mathbf{e}}, \tilde{\mathbf{x}} + \frac{1}{4}\sqrt{\frac{\epsilon}{\rho}}\hat{\mathbf{e}}\}} f(\mathbf{x})$
12:         $\mathbf{y}_t = \mathbf{x}_t$, $\iota = 0$
13:     **end if**
14:     $\mathbf{x}_{t+1} = \mathbf{y}_t = \eta(\tilde{\nabla}F(\mathbf{y}_t) - \iota)$
15:     $\mathbf{v}_{t+1} = \mathbf{x}_{t+1} - \mathbf{x}_t$
16:     $\mathbf{y}_{t+1} = \mathbf{x}_{t+1} + (1 - \theta)\mathbf{v}_{t+1}$
17:     **if** $t_{\text{perturb}} \neq -\mathscr{T} - 1$ and $t - t_{\text{perturb}} \leq \mathscr{T}$ **then**
18:         $(\mathbf{y}_{t+1}, \mathbf{x}_{t+1}) = \tilde{\mathbf{x}} + r \cdot \left(\frac{\mathbf{y}_{t+1} - \tilde{\mathbf{x}}}{\|\mathbf{y}_{t+1} - \tilde{\mathbf{x}}\|} \cdot \frac{\mathbf{x}_{t+1} - \tilde{\mathbf{x}}}{\|\mathbf{x}_{t+1} - \tilde{\mathbf{x}}\|}\right)$
19:     **else if** $f(\mathbf{x}_{t+1}) \leq f(\mathbf{y}_{t+1}) + \left\langle \tilde{\nabla}F(\mathbf{y}_{t+1}), \mathbf{x}_{t+1} - \mathbf{y}_{t+1}\right\rangle - \frac{\gamma}{2}\|\mathbf{x}_{t+1} - \mathbf{y}_{t+1}\|^2$ **then**
20:         $(\mathbf{x}_{t+1}, \mathbf{v}_{t+1}) \leftarrow \text{NCE}(\mathbf{x}_{t+1}, \mathbf{v}_{t+1}, s)$
21:         $\mathbf{y}_{t+1} \leftarrow \mathbf{x}_{t+1} + (1 - \theta)\mathbf{v}_{t+1}$
22:     **end if**
23: **end for**

---

**Algorithm 2:** Negative Curvature Exploitation (NCE) $(\mathbf{x}_t, \mathbf{v}_t, s)$

---

1: **if** $\|\mathbf{v}_t\| \geq s$ **then**
2:     $\mathbf{x}_{t+1} \leftarrow \mathbf{x}_t$
3: **else**
4:     $\delta = s \cdot \mathbf{v}_t/\|\mathbf{v}_t\|$
5:     $\mathbf{x}_{t+1} \leftarrow \arg\min_{\mathbf{x} \in \{\mathbf{x}_t + \delta, \mathbf{x}_t - \delta\}} f(\mathbf{x})$
6: **end if**

---

For simplicity, we denote the error of Jordan's gradient estimation as $\hat{\nu}$. To solve the problem of monotonic decrease for function value in momentum-based nonconvex optimization problems, we consider the Hamiltonian of the function (Jin et al., 2018b) in our proof, which is defined as

$$E_t = F(\mathbf{x}_t) + \frac{1}{2\eta}\|\mathbf{v}_t\|^2. \tag{16}$$

The Hamiltonian composes a potential energy term and a kinetic energy term. It monotonically decreases in the continuous-time scenario. To prove Theorem 2.2, we consider the dynamics of Algorithm 1 in the two different cases depending on whether (12) holds. If it does not hold, the following lemma holds by using Lemma 4 of Ref. (Zhang & Gu, 2022) and replacing the zeroth-order queries to $F(\mathbf{x}_t)$ and $F(\mathbf{y}_t)$ with the noisy queries $f(\mathbf{x}_t)$ and $f(\mathbf{y}_t)$.

**Lemma 2.3** (Adaptive version of Lemma 3, Ref. (Zhang & Gu, 2022))**.** *We consider $F(\cdot)$ is $\ell$-smooth and $\rho$-Hessian Lipschitz. Assume one can access the zeroth-order oracle with noise $\nu$ and the first-order oracle with noise $\hat{\nu}$. Set the learning rate $\eta \leq 1/4\ell$, $\theta \in [2\eta\gamma, 1/2]$. For each iteration $t$ where (12) does not hold, running Algorithm 1 will decrease the Hamiltonian defined in (16) by*

$$E_{t+1} \leq E_t - \frac{\theta}{2\eta}\|\mathbf{v}_t\|^2 - \frac{\eta}{4}\|\nabla f(\mathbf{y}_t)\|^2 + O(\eta\hat{\nu}^2) + O(\nu). \tag{17}$$

On the other hand, if (12) holds, the function has an approximate large negative curvature between $\mathbf{y}_t$ and $\mathbf{x}_t$. The accelerated gradient step might not decrease the value for the Hamiltonian. We thus call the negative curvature exploitation subroutine (Algorithm 2) to further decrease the Hamiltonian. In particular, when choosing large enough constant $c_r$, the following lemme holds by replacing the zeroth-order query to $F(\mathbf{x}_t)$ and $F(\mathbf{y}_t)$ with the noisy query $f(\mathbf{x}_t)$ and $f(\mathbf{y}_t)$ and noise term $O(\nu)$, respectively, in Lemma 4 of Ref. (Zhang & Gu, 2022).

**Lemma 2.4** (Adapted version of Lemma 4, Ref. (Zhang & Gu, 2022)). *Assume that $F(\cdot)$ is $\ell$-smooth, $\rho$-Hessian Lipschitz, and we are given the zeroth-order oracle with noise strength $\nu$ and the first-order oracle with noise strength $\hat{\nu}$. Set the learning rate $\eta \leq 1/4\ell$, $\theta \in [2\eta\gamma, 1/2]$. For each iteration $t$ where (12) holds, running Algorithm 1 wiil decrease the Hamiltonian defined in (16) by*

$$E_{t+1} \leq E_t - \min\left\{\frac{s^2}{2\eta}, \frac{1}{2}\gamma s^2 - \rho s^3 - O\left(\frac{\hat{\nu}^2}{\gamma}\right)\right\} + O(\nu). \tag{18}$$

We set an additional parameter $\mathscr{T}' = \Theta(\sqrt{\kappa})$. Based on Lemma 2.3 and Lemma 2.4, and proper choices of $\hat{\nu}$ and $\nu$, Lemma 5 of Ref. (Zhang & Gu, 2022) carries over as the below lemma when the norm of the estimated gradient is large enough, i.e. $\left\|\tilde{\nabla}F(\mathbf{x}_t)\right\| \geq 3\epsilon/4$.

**Lemma 2.5** (Adaptive version of Lemma 5, Ref. (Zhang & Gu, 2022)). *If $\left\|\tilde{\nabla}F(\mathbf{x}_t)\right\| \geq 3\epsilon/4$ and the noise strengths are bounded by $\nu, \hat{\nu} \leq O(\epsilon^{1.25})$ for all $\tau \in [0, \mathscr{T}']$, Algorithm 1 can decrease the Hamiltonian by $E_{\mathscr{T}'} - E_0 \leq -\mathscr{F}$ using*

$$\mathscr{T}' = \sqrt{\kappa}\chi c \tag{19}$$

*iterations in Algorithm 1, where $\chi = \max\{1, \log(d\ell B/\rho\epsilon\delta_0)\}$, and $c$ is a large enough constant given in Theorem 2.2*

On the other hand, when the estimated gradient is small, we obtain the following adaptive version of Lemma 7 of Ref. (Zhang & Gu, 2022).

**Lemma 2.6** (Adaptive version of Lemma 7, Ref. (Zhang & Gu, 2022)). *Suppose $\left\|\tilde{\nabla}F(\mathbf{x}_t)\right\| \leq 3\epsilon/4$ and the noise strengths are bounded by $\nu, \hat{\nu} \leq O(\epsilon^{3.25}/d^{0.5})$, $\lambda_{\min}(\nabla F(\mathbf{x}_t)) \leq -\sqrt{\rho\epsilon}$. For any $0 \leq \delta_0 \leq 1$, we set the parameters as Theorem 1.3. Suppose no perturbation is added in the iterations $[t - \mathscr{T}, t]$. By running Algorithm 1 for $\mathscr{T}$ iterations, we have*

$$\hat{\mathbf{e}}^\top \nabla^2 F(\mathbf{x}_t)\hat{\mathbf{e}} \leq -\frac{\sqrt{\rho\epsilon}}{4}, \tag{20}$$

*with probability at least $1 - \delta_0$.*

Furthermore, the following lemma from Ref. (Zhang & Li, 2021) indicates that the function value of $F$ will decrease fast along the direction of $\hat{\mathbf{e}}$.

**Lemma 2.7** (Lemma 6, Ref. (Zhang & Li, 2021)). *Suppose the function $F$ is $\ell$-smooth and $\rho$-Hessian Lipschitz. For any point $\mathbf{x}_t$, if there exists a unit vector $\hat{\mathbf{e}}$ satisfying $\hat{\mathbf{e}}^\top F(\mathbf{x}_t)\hat{\mathbf{e}} \leq -\sqrt{\rho\epsilon}/4$, we have*

$$F\left(\mathbf{x}_t - \frac{F'_{\hat{\mathbf{e}}}(\mathbf{x}_t)}{4\left|F'_{\hat{\mathbf{e}}}(\mathbf{x}_t)\right|} \cdot \sqrt{\frac{\epsilon}{\rho}}\right) \leq F(\mathbf{x}_t) - \frac{1}{384}\sqrt{\frac{\epsilon^3}{\rho}}, \tag{21}$$

*where $F'_{\hat{\mathbf{e}}}(\mathbf{x}_t)$ is the entry of the derivative along $\hat{\mathbf{e}}$.*

Now, we are ready to prove Theorem 2.2.

*Proof.* We first set $\nu \leq \frac{C_0}{\ell} \cdot \frac{\epsilon^{10}}{d^5}$ for some small enough constant $C_0$. According to Lemma 2.1, we bound $\hat{\nu} \leq O(\epsilon^{3.25}/d^{0.5})$ with probability at least $1 - \delta_0$. Assume Algorithm 1 starts at point $\mathbf{x}_0$ and the local minimum of $F$ has value $F^*$. Since $F$ is $B$-bounded, $F(\mathbf{x}_0) - F^* \leq 2B$. Set the total number of iterations $T$ to be:

$$T = 3\max\left\{\frac{2B\mathscr{T}'}{\mathscr{F}}, 768B\mathscr{T} \cdot \sqrt{\frac{\rho}{\epsilon^3}}\right\}. \tag{22}$$

Suppose for some iterations $\mathbf{x}_t$, we have $\tilde{\nabla} F(\mathbf{x}_t) \leq 3\epsilon/4$ and $\lambda_{\min}(\nabla^2 F(\mathbf{x}_t)) \leq -\sqrt{\rho\epsilon}$. The error probability of this assumption is given later. Under this assumption, the function value decreases for $\frac{1}{384} \cdot \sqrt{\frac{\epsilon^3}{\rho}}$ after each $\mathscr{T}'$ iterations. The number of such iterations when Lemma 2.7 can be called is bounded by $T/3$ times, for otherwise the function value will decrease greater than $2B \geq F(\mathbf{x}_0) - F^*$, which is impossible. The failure probability is composed of two parts: the failure probability of estimating the gradients in Lemma 2.1 and the failure probability of Lemma 2.7. In each iteration, the probability of failure is bounded by $2\delta_0$ according to the union bound. When we choose a large enough constant $\mathbf{c}_\delta$, the overall probability that Algorithm 1 fails to indicate a negative curvature is upper bounded by

$$\frac{T}{3} \cdot 2\delta_0 \leq \frac{\delta}{2}. \tag{23}$$

Excluding the iterations that Lemma 2.7 is applied, there are $2T/3$ iterations left. We consider the iterations $\mathbf{x}_t$ with large gradients, $\tilde{\nabla} F(\mathbf{x}_t) \geq 3\epsilon/4$. According to Lemma 2.5, the function value decreases by at least $\mathscr{F}$ with probability at least $1 - \delta_0$ in $\mathscr{T}'$ iterations. Thus there can be at most $T/3$ steps with large gradients, for otherwise, the function value will decrease greater than $2B \geq F(\mathbf{x}_0) - F^*$, which is impossible. The fail probability is bounded by

$$\frac{T}{3} \cdot \delta_0 \leq \frac{\delta}{2}. \tag{24}$$

In summary, we can deduce that with probability at least $1 - \delta$, there are at most $T/3$ iterations within which the neighboring $\mathscr{T}$ iterations have small gradients but large negative curvatures, and at most $T/3$ iterations with large gradients. Therefore, the rest $T/3$ iterations must be $\epsilon$-SOSPs of target function $F$. The number of queries is thus bounded by

$$T \leq \tilde{O}\left( \frac{B\ell}{\epsilon^{1.75}} \cdot \log d \right). \tag{25}$$

$\square$

## 2.2 ROBUSTNESS OF QUANTUM PERTURBED GRADIENT DESCENT

When the noise rate increases but is still bounded by $\nu \leq \tilde{O}(\epsilon^6/d^4)$, some quantum algorithms using perturbed gradient descent (PGD) for noiseless cases are robust against such noise. We introduce the quantum PGD algorithm, which is the one of the standard methods used for noiseless nonconvex optimization (Zhang et al., 2021).

Algorithm 3 replaces the gradient queries in PGD (Jin et al., 2021) by Jordan's gradient estimations in Lemma 2.1.

---
**Algorithm 3:** Perturbed Gradient Descent with Quantum Gradient Computation

---
**Input:** $\mathbf{x}_0$, learning rate $\eta$, noise ratio $r$
  1: **for** $t = 0, 1, \ldots, T$ **do**
  2:     Apply Lemma 2.1 to compute an estimation $\tilde{\nabla} F(\mathbf{x})$ of $\nabla F(\mathbf{x})$
  3:     $\mathbf{x}_{t+1} \leftarrow \mathbf{x}_t - \eta(\tilde{\nabla} F(\mathbf{x}) + \xi_t)$, $\xi_t$ uniformly $\sim B_0(r)$
  4: **end for**

---

We prove that Algorithm 3 has the following performance guarantee:

**Theorem 2.8** (Formal version of Theorem 1.4). *Suppose we have a target function $F$ and its noisy evaluation $f$ satisfying Assumption 1.1 with $\nu \leq \tilde{O}(\delta^2\epsilon^6/d^4)$. Algorithm 3 can find an $\epsilon$-SOSP of F satisfying Eq. (4) with probability at least $1 - \delta$, using*

$$\tilde{O}\left( \frac{\ell B}{\epsilon^2} \cdot \log^4 d \right) \tag{26}$$

*queries to $U_f$ defined in (7), under the following parameter choices:*

$$\eta = \frac{1}{\ell}, \quad \delta_0 = \frac{\delta\epsilon^2}{32\ell B}\chi^{-4}, \quad r = \epsilon\chi^{-3}c^{-6}, \quad \mathscr{T} = \frac{\chi c}{\eta\sqrt{\rho\epsilon}}, \quad \mathscr{F} = \sqrt{\frac{\epsilon^3}{\rho}}\chi^{-3}c^{-5}, \quad (27)$$

*where $c$ is some large enough constant and $\chi = \max\{1, \log(d\ell B/\rho\epsilon\delta_0)\}$.*

To prove Theorem 2.8, we consider two cases where the current iteration $\mathbf{x}_t$ is not an $\epsilon$-SOSP of $F$. In the first case, the gradient $\|\nabla F(\mathbf{x}_t)\| \geq \epsilon$ is larger than $\epsilon$. In the second case, the gradient $\|\nabla F(\mathbf{x}_t)\| \leq \epsilon$ but the minimal eigenvalue of the Hessian matrix satisfies $\lambda_{\min}(\nabla^2 F(\mathbf{x}_t)) \leq -\sqrt{\rho\epsilon}$. Intuitively, the proof of Theorem 2.8 is composed of the performance guarantees regarding both cases. For Algorithm 3, it takes $\mathscr{T} = O(\log d)$ queries to $U_f$ to decrease the function value by $\mathscr{F} = O(1/\log^3 d)$ (Jin et al., 2018a).

We first set $\nu \leq \frac{C_0}{\ell} \cdot (\frac{\delta\epsilon^3}{Bd^2\chi^4\ell})^2$ for some small enough constant $C_0$. Formally, we introduce the following lemma characterizing the performance of Algorithm 3 when the gradient is large:

**Lemma 2.9.** *Under the setting of Theorem 2.8, for any iteration $t$ of Algorithm 3 with $\|\nabla F(\mathbf{x}_t)\| \geq \epsilon$, we have $F(\mathbf{x}_{t+1}) - F(\mathbf{x}_t) \leq -\eta\epsilon^2/4$ with probability at least $1 - \delta_0$, where $\delta_0$ is defined in Eq. (27).*

*Proof.* We set $\omega = 2d/\delta_0$ and choose $C_0$ small enough such that

$$\left\|\tilde{\nabla}F(\mathbf{x}) - \nabla F(\mathbf{x})\right\| \leq \frac{\epsilon}{20} \quad (28)$$

with probability at least $1 - \delta_0$ according to Lemma 2.1.

Next, we choose $c$ such that $\left\|\tilde{\nabla}F(\mathbf{x}) - \nabla F(\mathbf{x})\right\| \leq \epsilon/20$. Recall that the perturbation $\xi_t$ is chosen from $B_0(r)$, the stochastic part in each iteration $\kappa_t = \tilde{\nabla}F(\mathbf{x}) - \nabla F(\mathbf{x}) + \xi_t$ is bounded by $\|\kappa_t\| = \epsilon/10$. According to the update rule $\mathbf{x}_{t+1} = \mathbf{x}_t - \eta(\nabla F(\mathbf{x}_t) + \kappa_t)$ of Algorithm 3, we have

$$F(\mathbf{x}_{t+1}) \leq F(\mathbf{x}_t) + \langle\nabla F(\mathbf{x}_t), \mathbf{x}_{t+1} - \mathbf{x}_t\rangle + \frac{\ell}{2}\|\mathbf{x}_{t+1} - \mathbf{x}_t\|^2$$

$$\leq F(\mathbf{x}_t) - \eta\left[\|\nabla F(\mathbf{x}_t)\|^2 - \|\nabla F(\mathbf{x}_t)\|\|\kappa_t\|\right] + \frac{\eta^2\ell}{2}\left[\|\nabla F(\mathbf{x}_t)\|^2 + 2\|\nabla F(\mathbf{x}_t)\|\|\kappa_t\| + \|\kappa_t\|^2\right]$$

$$\leq F(\mathbf{x}_t) - \eta\|\nabla F(\mathbf{x}_t)\|\left[\frac{1}{2}\|\nabla F(\mathbf{x}_t)\| - 2\|\kappa_t\|\right] + \frac{\eta}{2}\|\kappa_t^2\|$$

$$\leq F(\mathbf{x}_t) - \frac{\eta\epsilon^2}{4}. \quad (29)$$

$\square$

In addition, we can generalize the following lemma in Ref. (Jin et al., 2018a).

**Lemma 2.10** (Lemma 67, Ref. (Jin et al., 2018a)). *Suppose we are given a oracle that outputs an gradient estimation $\tilde{\nabla}F(\mathbf{x}_t)$ such that $\left\|\tilde{\nabla}F(\mathbf{x}_t) - \nabla F(\mathbf{x}_t)\right\| \leq \epsilon/20$. Consider a iteration $t$ of Algorithm 3 with $\|\nabla F(\mathbf{x}_t)\| \geq \epsilon$. By using the PGD update rule $\mathbf{x}_{t+1} = \mathbf{x}_t - \eta(\tilde{\nabla}F(\mathbf{x}_t) + \xi_t)$, we have $F(\mathbf{x}_{t+1}) - F(\mathbf{x}_t) \leq -\eta\epsilon^2/4$ with probability at least $1 - \delta_0$, where $\delta_0$ is defined in Eq. (27).*

When the gradient is small but the minimal eigenvalue of the Hessian matrix is large, i.e., the function has a large negative curvature at the current iteration, we have the following lemma from Ref. (Jin et al., 2018a).

**Lemma 2.11** (Lemma 68, Ref. (Jin et al., 2018a)). *Suppose we are given a oracle that outputs an gradient estimation $\tilde{\nabla}F(\mathbf{x}_t)$ such that $\left\|\tilde{\nabla}F(\mathbf{x}_t) - \nabla F(\mathbf{x}_t)\right\| \leq \epsilon/20$ and the norm of the perturbation in PGD is bounded by $\|\xi_t\| \leq r$ with $r > \epsilon/20$. If $\|\nabla F(\mathbf{x}_t)\| \leq \epsilon$ and $\lambda_{\min}(\nabla^2 F(\mathbf{x}_t)) \leq -\sqrt{\rho\epsilon}$. By using the PGD update rule $\mathbf{x}_{t+1} = \mathbf{x}_t - \eta(\tilde{\nabla}F(\mathbf{x}_t) + \xi_t)$, we have $F(\mathbf{x}_{t+\mathscr{T}}) - F(\mathbf{x}_t) \leq -\mathscr{F}$ with probability at least $1 - \delta_0$ when running Algorithm 3.*

Now, we are ready to prove Theorem 2.8.

*Proof of Theorem 2.8.* Assume our Algorithm 3 starts at point $\mathbf{x}_0$ and the local minimum of $F$ has value $F^*$. Since $F$ is $B$-bounded, we have $F(\mathbf{x}_0) - F^* \leq 2B$. Set the total number of iterations $T$ to be:

$$T = 3 \max \left\{ \frac{8B}{\eta \epsilon^2}, \frac{2B\mathscr{T}}{\mathscr{F}} \right\}. \tag{30}$$

Assume for some iterations $\mathbf{x}_t$, we have $\nabla F(\mathbf{x}_t) \leq \epsilon$ and $\lambda_{\min}(\nabla^2 F(\mathbf{x}_t)) \leq -\sqrt{\rho \epsilon}$. The error probability of this assumption is given later. Under this assumption, the function value decreases for $\mathscr{F}$ after each $\mathscr{T}$ iterations. The number of such iterations when Lemma 2.11 can be called is bounded by $T/3$ times, for otherwise the function value will decrease greater than $2B \geq F(\mathbf{x}_0) - F^*$, which is impossible. The failure probability is composed of two parts: the failure probability of estimating the gradients in Lemma 2.1 and the failure probability of Lemma 2.11. In each iteration, the probability of failure is bounded by $2\delta_0$ according to the union bound. The overall probability that Algorithm 3 fails to indicate a negative curvature is upper bounded by

$$\frac{T}{3} \cdot 2\delta_0 \leq \frac{\delta}{2} \tag{31}$$

for any $\chi$.

Excluding the iterations in which Lemma 2.11 is applied, we still have $2T/3$ iterations left. We now consider the iterations $\mathbf{x}_t$ with large gradients $\nabla F(\mathbf{x}_t) \geq \epsilon$. According to Lemma 2.9, the function value decreases by at least $\eta \epsilon^2/4$ with probability at least $1 - \delta_0$ in each iteration. Thus there can be at most $T/3$ steps with large gradients, for otherwise, the function value will decrease greater than $2B \geq F(\mathbf{x}_0) - F^*$, which is impossible. The fail probability is bounded by

$$\frac{T}{3} \cdot \delta_0 \leq \frac{\delta}{2}. \tag{32}$$

In summary, we can deduce that with probability at least $1 - \delta$, there are at most $T/3$ iterations within which the neighboring $\mathscr{T}$ iterations have small gradients but large negative curvatures, and at most $T/3$ iterations with large gradients. Therefore, the rest $T/3$ iterations must be $\epsilon$-SOSPs of target function $F$. The number of queries is thus bounded by

$$T \leq \tilde{O}\left( \frac{B\ell}{\epsilon^2} \cdot \log^4 d \right). \tag{33}$$

$\square$

The above Theorem 2.8 indicates that our PGD method with quantum gradient computation still converges and finds an $\epsilon$-SOSP using the same number of iterations (i.e., the same number of queries), even if there exists small noise on the quantum evaluation oracles. We remark that compared to Algorithm 4 in Ref. (Zhang et al., 2021), Algorithm 3 employs a classical perturbation uniformly chosen from the ball $\mathbb{B}(\mathbf{0}, r)$. Therefore, Algorithm 3 requires no access to the quantum evaluation oracle without noise.

It is natural to ask if we can improve the dependence on $\log d$ in the query complexity. We answer this question with an affirmative answer in Appendix C under some additional assumptions. Consider if we have functions $F$ and $f$ that satisfy Assumption 1.1 with $\nu \leq O(\epsilon^6/d^4)$ and we further assume that $f$ is twice differentiable with $\sup_{\mathbf{x}} \|\nabla f - \nabla F\| \leq O(\ell/d^{2+\varsigma})$ and $\sup_{\mathbf{x}} \|\nabla^2 f - \nabla^2 F\| \leq O(\rho/d^{1.5+\varsigma})$ for arbitrary $\varsigma > 0$. We propose a quantum algorithm that can find an $\epsilon$-SOSP for $F$ using $\tilde{O}(\ell B/\epsilon^2 \cdot \log^2 d)$ queries to the quantum evaluation oracle in Eq. (7).

In addition, we can use the techniques above to prove the algorithmic upper bound for function pair $(F, f)$ satisfying Assumption 1.2 with $\tilde{\nu} \leq O(\epsilon/d^{0.5+\varsigma})$ for $\varsigma > 0$ and $\varsigma = \Omega(1/\log(d))$. We provide the following corollary corresponding to the last line in Table 2.

**Corollary 2.12.** *Suppose we have a target function $F$ and a noisy function $f$ satisfying Assumption 1.2 with $\tilde{\nu} \leq O(\epsilon/d^{0.5+\varsigma})$ for $\varsigma > 0$ and $\varsigma = \Omega(1/\log(d))$. Consider the gradient descent*

$\mathbf{x}_{t+1} = \eta(\nabla f + \xi_t)$ *with* $\xi_t$ *uniformly chosen from ball* $\mathbb{B}(\mathbf{0}, r)$. *This rule can output an* $\epsilon$-*SOSP of* $F$ *satisfying Eq. (4), using*

$$\tilde{O}\left(\frac{\ell B}{\epsilon^2} \cdot \log^4 d\right) \tag{34}$$

*queries to* $U_{\mathbf{g}}$ *in (8) with probability* $1 - \delta$, *under the following parameter choices*

$$\eta = \frac{1}{\ell}, \quad \delta_0 = \frac{\delta\epsilon^2}{4\ell B}\chi^{-4}, \quad r = \epsilon\chi^{-3}c^{-6}, \quad \mathscr{T} = \frac{\chi c}{\eta\sqrt{\rho\epsilon}}, \quad \mathscr{F} = \sqrt{\frac{\epsilon^3}{\rho}}\chi^{-3}c^{-5}, \tag{35}$$

*where* $c$ *is some large enough constant and* $\chi = \max\{1, \log(d\ell B/\rho\epsilon\delta_0)\}$.

*Proof.* Without loss of generality, we set $\tilde{\nu} \leq O(\epsilon/d^{0.5+\zeta})$ and $\zeta \geq \log 20/\log(d)$ such that $\tilde{\nu} \geq \epsilon/20$. We then choose $c$ large enough such that $\left\|\tilde{\nabla}F(\mathbf{x}) - \nabla F(\mathbf{x})\right\| \leq \epsilon/20$. As the perturbation $\xi_t$ is chosen from $B_0(r)$, the stochastic part in each iteration $\kappa_t = \tilde{\nabla}F(\mathbf{x}) - \nabla F(\mathbf{x}) + \xi_t$ is bounded by $\|\kappa_t\| = \epsilon/10$. Similar to the proof of Theorem 2.8, we set

$$T = 3\max\left\{\frac{8B}{\eta\epsilon^2}, \frac{2B\mathscr{T}}{\mathscr{F}}\right\}. \tag{36}$$

Suppose for some iterations, the function have small gradients $\nabla F(\mathbf{x}_t) \leq \epsilon$ and large negative curvatures $\lambda_{\min}(\nabla^2 F(\mathbf{x}_t)) \leq -\sqrt{\rho\epsilon}$. Under this assumption, the function value decreases for $\mathscr{F}$ after each $\mathscr{T}$ iterations according to Lemma 2.11. The number of such iterations when Lemma 2.11 can be called is bounded by $T/3$ times, for otherwise, the function value will decrease greater than $2B \geq F(\mathbf{x}_0) - F^*$, which is impossible. The failure probability is bounded above by

$$\frac{T}{3} \cdot \delta_0 \leq \delta/2. \tag{37}$$

Except for the iterations that Lemma 2.11 is applied, we still have $2T/3$ iterations left. We now consider the iterations $\mathbf{x}_t$ with large gradients, $\nabla F(\mathbf{x}_t) \geq \epsilon$. According to Lemma 2.10, the function value decreases by at least $\eta\epsilon^2/4$ with the probability at least $1 - \delta_0$ in each iteration. Thus there can be at most $T/3$ steps with large gradients, for otherwise, the function value will decrease greater than $2B \geq F(\mathbf{x}_0) - F^*$, which is impossible. The failure probability is again bounded above by

$$\frac{T}{3} \cdot \delta_0 \leq \delta/2. \tag{38}$$

Therefore, we can deduce that with probability at least $1 - \delta$, there are at most $T/3$ iterations resulting in points having small gradients but large negative curvature, and at most $T/3$ iterations with large gradients. Therefore, the rest $T/3$ iterations must be $\epsilon$-SOSPs. The number of the queries is bounded by

$$T \leq \tilde{O}\left(\frac{B\ell}{\epsilon^2} \cdot \log^4 d\right). \tag{39}$$

$\square$

Corollary 2.12 indicates that when the gradient $\mathbf{g} = \nabla f$ of the noisy function is close enough to the gradient $\nabla F$ of the target function, the PGD algorithm can converge even if the gradient $\mathbf{g}$ is noisy. As we can directly query the noisy gradient, the quantum algorithms such as quantum mean estimation (Hamoudi, 2021; Cornelissen et al., 2022) or quantum gradient estimation (Jordan, 2005; Gilyén et al., 2019) cannot provide speedup in this case. Moreover, quantum approaches to add perturbation such as quantum simulation (Zhang et al., 2021) require zeroth-order information, which is unavailable under Assumption 1.2. Therefore, there is no quantum speedup compared to the classical gradient descent in the setting of Corollary 2.12.

# 3 QUANTUM SPEEDUP USING MEAN ESTIMATION

When the noise strength further increases, it exceeds the robustness of quantum PGD. To handle this issue, we apply a Gaussian smoothing to the noisy function $f$ inspired by Ref. (Jin et al., 2018a), which can turn a possibly nonsmooth or even non-continuous $f$ into a function $f_\sigma$ with "good" properties such as smoothness and Hessian-Lipschitzness.

## 3.1 ZEROTH-ORDER ALGORITHM AND PERFORMANCE GUARANTEE

In this section, we introduce a quantum algorithm based on Gaussian smoothing for function pairs $(F, f)$ satisfying Assumption 1.1 with $\nu \le O(\epsilon^{1.5}/d)$. We formally define the *Gaussian smoothing* for a function $f$ as follows.

**Definition 3.1.** Given a function $f \colon \mathbb{R}^d \to \mathbb{R}$, we define its Gaussian smoothing $f_\sigma \colon \mathbb{R}^d \to \mathbb{R}$ as

$$f_\sigma := \mathbb{E}_{\mathbf{z} \sim \mathcal{N}(0, \sigma^2 I)}[f(\mathbf{x} + \mathbf{z})], \tag{40}$$

where the parameter $\sigma$ is the smoothing radius.

Given a noisy function $f$ and a target function $F$ satisfying Assumption 1.1, Gaussian smoothing transfers the (probably even non-smooth or not differentiable) noisy $f$ into a smooth function $f_\sigma$ that has close gradient and Hessian with $F$. Formally, $f_\sigma$ has the following properties according to Ref. (Jin et al., 2018a):

**Lemma 3.2** (Lemma 13, Ref. (Jin et al., 2018a)). *Assume the function pair $(F, f)$ satisfies Assumption 1.1, the Gaussian smoothing $f_\sigma$ of $f$ satisfies the following properties.*

- *$f_\sigma(\mathbf{x})$ is $O(\ell + \nu/\sigma^2)$-smooth and $O(\rho + \nu/\sigma^3)$-Hessian Lipshitz.*

- *The distance between the gradient and the Hessian of $f_\sigma$ and $F$ at any $\mathbf{x}$ is bounded by $\|\nabla f_\sigma(\mathbf{x}) - \nabla F(\mathbf{x})\| \le O(\rho d\sigma^2 + \nu/\sigma)$ and $\|\nabla^2 f_\sigma(\mathbf{x}) - \nabla^2 F(\mathbf{x})\| \le O(\rho\sqrt{d}\sigma + \nu/\sigma^2)$.*

The first part of Lemma 3.2 demonstrates that the Gaussian smoothing $f_\sigma$ is a smooth and Hessian Lipschitz function. Thus we can perform standard gradient descent on $f_\sigma$ with a polynomial convergence rate. The second part of Lemma 3.2 indicates that the gradients and Hessians of $f_\sigma$ are similar to those of the target function $F$ up to a term related to the noise rate $\nu$ and the smoothing radius $\sigma$. As the noise rate $\nu$ increases and the noisy function $f$ deviates further from the target function $F$, we have to choose a larger parameter $\sigma$ to bound the terms $\nu/\sigma$, $\nu/\sigma^2$, and $\nu/\sigma^3$. However, choosing a larger smoothing radius $\sigma$ will increase the term $\rho d\sigma^2$ and $\rho\sqrt{d}\sigma$, which erases the information about local geometry of $F$. Hence, the choice of $\sigma$ must balance between the two terms in the bounds in Lemma 3.2.

Suppose we have an $\tilde{\epsilon}$-SOSP $\mathbf{x}_{\text{SOSP}}$ of the Gaussian smoothing $f_\sigma$. One have to guarantee that an $\tilde{\epsilon}$-SOSP of the Gaussian smoothing $f_\sigma$ is also an $\epsilon$-SOSP of $F$. We now search for the value of $\sigma$ and $\tilde{\epsilon}$ such that $\nu$ is maximized. According to Lemma 3.2, we can bound the gradient and the minimal eigenvalue of Hessian for $F(\mathbf{x}_{\text{SOSP}})$ by the following inequalities.

$$\|\nabla F(\mathbf{x}_{\text{SOSP}})\| \le \rho d\sigma^2 + \frac{\nu}{\sigma} + \tilde{\epsilon}, \tag{41}$$

whereas

$$\lambda_{\min}(\nabla^2 F(\mathbf{x}_{\text{SOSP}})) \ge \lambda_{\min}(\nabla^2 f_\sigma(\mathbf{x}_{\text{SOSP}})) + \lambda_{\min}(\nabla^2 F(\mathbf{x}_{\text{SOSP}}) - \nabla^2 f_\sigma(\mathbf{x}_{\text{SOSP}}))$$

$$\ge -\sqrt{\left(\rho + \frac{\nu}{\sigma^3}\right)\tilde{\epsilon}} - \left\|\nabla^2 f_\sigma(\mathbf{x}_{\text{SOSP}}) - \nabla^2 F(\mathbf{x}_{\text{SOSP}})\right\|$$

$$\ge -\sqrt{\left(\rho + \frac{\nu}{\sigma^3}\right)\tilde{\epsilon}} - \left(\rho\sqrt{d}\sigma + \frac{\nu}{\sigma^2}\right). \tag{42}$$

Hence, to guarantee that an $\tilde{\epsilon}$-SOSP of $f_\sigma$ is an $\epsilon$-SOSP of $F$, we only need the following set of inequalities to be satisfied(up to constant factors).

$$\rho\sqrt{d}\sigma + \frac{\nu}{\sigma^2} \leq O(\sqrt{\rho\epsilon}), \tag{43}$$

$$\rho d\sigma^2 + \frac{\nu}{\sigma} \leq O(\epsilon), \tag{44}$$

$$\left(\rho + \frac{\nu}{\sigma^3}\right)\tilde{\epsilon} \leq O(\rho\epsilon). \tag{45}$$

From (43) and (44), we have

$$\sigma \leq O\left(\sqrt{\frac{\epsilon}{\rho d}}\right), \tag{46}$$

$$\nu \leq \sqrt{\rho\epsilon}\sigma^2 = O\left(\sqrt{\frac{\epsilon^3}{\rho}} \cdot \frac{1}{d}\right), \tag{47}$$

$$\tilde{\epsilon} \leq \frac{\rho\epsilon}{\left(\rho + \frac{\nu}{\sigma^3}\right)} = O\left(\frac{\epsilon}{\sqrt{d}}\right). \tag{48}$$

The above results indicate that we can guarantee that an $O(\epsilon/\sqrt{d})$-SOSP for $f_\sigma$ is an $\epsilon$-SOSP of the target function $F$.

The next step is to find an $O(\epsilon/\sqrt{d})$-SOSP of $f_\sigma$ using queries to the noisy oracle in (7). Through Gaussian smoothing, we convert the function evaluations of $f$ into stochastic gradients of $f_\sigma$. According to Ref. (Duchi et al., 2015) the gradients of $f_\sigma$ can be calculated as

$$\nabla f_\sigma = \frac{1}{\sigma^2}\mathbb{E}_{\mathbf{z}\sim\mathcal{N}(0,\sigma^2 I)}[\mathbf{z}(f(\mathbf{x}+\mathbf{z}) - (\mathbf{x}))]. \tag{49}$$

One can thus compute the gradient for $f_\sigma$ by querying the function value of $f$. However, the gradient is unbiasedly computed through averaging over the continuous Gaussian distribution. To approximate the gradient, we employ the zeroth-order quantum oracle in (7) to sample the stochastic gradient estimation $\mathbf{z}[f(\mathbf{x}+\mathbf{z}) - f(\mathbf{x})]/\sigma^2$, where $\mathbf{z} \sim \mathcal{N}(0,\sigma^2 I)$. The stochastic gradient has the following properties.

**Lemma 3.3** (Lemma 14, Ref. (Jin et al., 2018a)). *We denote* $\mathbf{g}(\mathbf{x};\mathbf{z}) = \mathbf{z}[f(\mathbf{x}+\mathbf{z}) - f(\mathbf{x})]/\sigma^2$, *where* $\mathbf{z} \sim \mathcal{N}(0,\sigma^2 I)$. *The following inequalities hold:*

$$\mathbb{E}_{\mathbf{z}}\mathbf{g}(\mathbf{x};\mathbf{z}) = \nabla f_\sigma(\mathbf{x}), \tag{50}$$

$$\Pr\left[\|\mathbf{g}(\mathbf{x};\mathbf{z}) - \nabla f_\sigma(\mathbf{x})\| \geq t\right] \leq \exp\left(-Bt^2/\sigma\right), \qquad \forall t > 0. \tag{51}$$

*The second inequality demonstrates that* $\mathbf{g}(\mathbf{x};\mathbf{z})$ *is a sub-Gaussian random variable with a tail* $B/\sigma$.

Lemma 3.3 guarantees that by sampling a large mini-batch and evaluating the mean of the stochastic gradients, the value converges to the gradient $\nabla f_\sigma(\mathbf{x})$ of the Gaussian smoothing $f_\sigma(\mathbf{x})$. Classically, the batch size required for the sampling can be obtained by the Chernoff bound (say, e.g. Ref. (Lugosi & Mendelson, 2019)).

**Lemma 3.4.** *Given a fixed point* $\mathbf{x}$ *and the mini-batch size* $m$, *for any* $\delta > 0$, *we have:*

$$\left\|\nabla f(\mathbf{x}) - \frac{1}{m}\sum_{i=1}^{m}\mathbf{g}(\mathbf{x};\mathbf{z}^{(i)})\right\| \leq \sqrt{\frac{2\sigma_0^2 d}{m}\log\left(\frac{d}{\delta}\right)} \tag{52}$$

*with probability at least* $1-\delta$, *where* $\sigma_0 = B/\sigma$ *is the standard deviation of the stochastic gradient.*

Lemma 3.4 indicates that it is sufficient to choose a mini-batch of size

$$m \geq \frac{2\sigma_0^2 d}{\epsilon^2}\log\left(\frac{d}{\delta}\right), \tag{53}$$

where $\sigma_0 = B/\sigma$, to estimate the gradient $\nabla f_\sigma(\mathbf{x})$ within $\epsilon$ under Euclidean norm with probability at least $1-\delta$. In addition, this bound is optimal in any classical algorithms (Hopkins, 2020), or

equivalently, any classical multivariate mean estimator with batch size less than this quantity will fail on a certain stochastic gradient function $\mathbf{g}(\mathbf{x}; \mathbf{z})$.

Quantumly, the well-known amplitude estimation algorithm (Brassard et al., 2002) provides a smaller error rate when estimating the mean of Bernoulli random variables. For the multivariate mean estimation problem of a random vector, quantum algorithms can also provide a speedup under certain circumstances. In particular, we consider the mean estimation task of estimating $\mathbf{g}(\mathbf{x}; \mathbf{z})$ in Lemma 3.3 given a *binary oracle* defined as follows.

**Definition 3.5.** Consider the random variable $\mathbf{g}(\mathbf{x}; \mathbf{z}) \in \mathbb{R}^d$ with $\mathbf{z} \sim \mathcal{N}(0, \sigma^2 I)$. Let $\mathcal{H}_{\mathbf{z}}$ and $\mathcal{H}_{\mathbf{g},\mathbf{x}}$ be two Hilbert spaces with basis states $\{|\mathbf{z}\rangle\}_{\mathbf{z}}$ and $\{\mathbf{g}(\mathbf{x}; \mathbf{z})\}_{\mathbf{z}}$, which contains quantum state encoding vectors $\mathbf{z}$ and $\mathbf{g}(\mathbf{x}; \mathbf{z})$, respectively. The binary oracle $B_{\mathbf{z}} : \mathcal{H}_{\mathbf{z}} \otimes \mathcal{H}_{\mathbf{g},\mathbf{x}} \to \mathcal{H}_{\mathbf{z}} \otimes \mathcal{H}_{\mathbf{g},\mathbf{x}}$ is defined as

$$B_{\mathbf{z}} : |\mathbf{z}\rangle |\mathbf{0}\rangle \to |\mathbf{z}\rangle |\mathbf{g}(\mathbf{x}; \mathbf{z})\rangle, \qquad \forall \mathbf{z} \sim \mathcal{N}(0, \sigma^2 I), \tag{54}$$

where we assume $\mathbf{0} \in \{\mathbf{g}(\mathbf{x}; \mathbf{z})\}_{\mathbf{z}}$.

In practice, the above binary oracle can be constructed by employing two quantum evaluation oracles in (7). Using such binary oracle, Ref. (Cornelissen et al., 2022) provides the following performance guarantee.

**Lemma 3.6** (Theorem 3.5, Ref. (Cornelissen et al., 2022)). *Suppose $\mathbf{g}$ is a $d$-dimensional random vector with mean $\mu$ and covariance matrix $\Sigma$ such that $\mathrm{Tr}(\Sigma) = \sigma_0^2$. Given two real values $\delta \in (0, 1)$ and $m \geq \log(d/\delta)$, there exists a quantum algorithm that outputs a mean estimation $\tilde{\mu}$ such that*

$$\|\tilde{\mu} - \mu\| \leq \begin{cases} O\left(\sqrt{\frac{\sigma_0^2}{m}}\right), & n \leq d, \\ O\left(\frac{\sqrt{d\sigma_0^2}\log\left(\frac{d}{\delta}\right)}{m}\right), & n > d, \end{cases} \tag{55}$$

*with probability at least $1 - \delta$. Such an algorithm requires $\tilde{O}(m)$ queries to the binary oracle.*

Lemma 3.6 indicates that it only requires

$$m \geq O\left(\frac{\sqrt{d}\sigma_0}{\epsilon} \log\left(\frac{d}{\delta}\right)\right) \tag{56}$$

samples to estimate the gradient $\nabla f_\sigma(\mathbf{x})$ within error $\epsilon$ with high probability. Compared with the classical mini-batch size in Lemma 3.4, quantum mean estimation provides a quadratic reduction when the classical mini-batch size is $\Omega(d)$. It is worthwhile to mention that the error scaling in Lemma 3.6 is near-optimal up to logarithmic factors (Cornelissen et al., 2022).

We consider using the PGD with stochastic gradient estimation to find an $\epsilon$-SOSP of the target function $F$ (also an $O(\epsilon/\sqrt{d})$-SOSP of $f_\sigma$) using noisy function $f$ in (7). The detailed algorithm is given in Algorithm 4.

---

**Algorithm 4:** Perturbed Stochastic Gradient Descent with Quantum Mean Estimation

---

**Input:** $\mathbf{x}_0$, learning rate $\eta$, noise ratio $r$, mini-batch size $m$
1: **for** $t = 0, 1, ..., T$ **do**
2:     Estimate the gradient $\tilde{\nabla}f_\sigma(\mathbf{x}_t)$ of $f_\sigma(\mathbf{x}_t)$ using quantum mean estimation and $m$ binary queries to (54)
3:     $\mathbf{x}_{t+1} \leftarrow \mathbf{x}_t - \eta(\tilde{\nabla}f_\sigma(\mathbf{x}_t) + \xi_t)$, $\xi_t$ uniformly $\sim B_0(r)$
4: **end for**

---

We now prove the performance guarantee for Algorithm 4, which is the formal version of Theorem 1.5.

**Theorem 3.7** (Formal version of Theorem 1.5). *Suppose we have a target function $F$ and its noisy evaluation $f$ satisfying Assumption 1.1 with $\nu \leq O(\sqrt{\epsilon^3/\rho} \cdot (1/d))$. With probability at least $1 - \delta$, Algorithm 4 finds an $\epsilon$-SOSP of $F$ satisfying (4), using*

$$\tilde{O}\left(\frac{d^{2.5}}{\epsilon^{3.5}} \cdot \mathrm{poly}(\Delta_f, \ell, \rho)\right) \tag{57}$$

*queries to $U_f$ in (7), under the following parameter choices:*

$$\eta = \frac{1}{\ell'}, \quad \delta_0 = \frac{\delta \epsilon'^2}{16\ell'\Delta_f}\chi^{-4}, \quad r = \epsilon'\chi^{-3}c^{-6}, \quad \mathscr{T} = \frac{\chi c}{\eta\sqrt{\rho'\epsilon'}}, \quad \mathscr{F} = \sqrt{\frac{\epsilon'^3}{\rho'}}\chi^{-3}c^{-5}, \quad (58)$$

*where $c$ is some large enough constant, $\Delta_f = f_\sigma(\mathbf{x}_0) - f_\sigma(\mathbf{x}^*)$ is the value between the initial point $\mathbf{x}_0$ and the global minima point $\mathbf{x}^*$, $\chi = \max\{1, \log(d\ell'\Delta_f/\rho'\epsilon'\delta_0)\}$, $\ell' = O(\ell + \sqrt{\epsilon/\rho})$, and $\rho' = O(\rho)$ are the smoothness and Hessian-Lipschitz parameters for $f_\sigma$, and $\epsilon' = O(\epsilon/\sqrt{d})$ such that an $\epsilon'$-SOSP of $f_\sigma$ is an $\epsilon$-SOSP of $F$.*

*Proof.* Notice that an $\epsilon$-SOSP of target function $F$ is an $O(\epsilon/\sqrt{d})$-SOSP of function $f_\sigma$, we only need to prove that Algorithm 4 converges to a $O(\epsilon/\sqrt{d})$-SOSP of $f_\sigma$ using $\tilde{O}(\ell B d^{2.5}/\epsilon^{3.5})$ queries.

In each iteration, Algorithm 4 estimates the gradient $\nabla f_\sigma(\mathbf{x}_t)$ using quantum mean estimation with $O(m)$ queries to the quantum evaluation oracle. According to Lemma 3.3, the stochastic gradient $\mathbf{g}(\mathbf{x}_t; \mathbf{z}^{(i)})$ for $\mathbf{z} \sim \mathcal{N}(0, \sigma^2 I)$ is a random vector with mean $\nabla f_\sigma(\mathbf{x}_t)$ and variance $\sigma_0^2 = B^2/\sigma^2$. As we choose $\sigma = \sqrt{\epsilon/\rho d}$, we require

$$m \geq O\left(\frac{\sqrt{d}(B/\sigma)}{\epsilon/\sqrt{d}}\log\left(\frac{d}{\delta_0}\right)\right) \quad (59)$$

$$= \tilde{O}\left(B\sqrt{\rho} \cdot \sqrt{\frac{d^3}{\epsilon^3}}\right) \quad (60)$$

queries to bound the error $\left\|\tilde{\nabla} f_\sigma(\mathbf{x}_t) - \nabla f_\sigma(\mathbf{x}_t)\right\| \leq \epsilon'/20 \leq O(\epsilon/\sqrt{d} \cdot (1/20))$ with probability at least $1 - \delta_0$ according to Lemma 3.6.

Next, Algorithm 4 employs the estimations of the gradient and the PGD to find an $\epsilon'$-SOSP of $f_\sigma$. Recall that we choose $\sigma = O(\sqrt{\epsilon/\rho d})$ and $\nu = O(\sqrt{\epsilon^3/\rho} \cdot (1/d))$, $f_\sigma$ is thus $\ell'$-smooth and $\rho'$-Hessian Lipschitz, where

$$\ell' = O\left(\ell + \frac{\nu}{\sigma^2}\right) = O\left(\ell + \sqrt{\frac{\epsilon}{\rho}}\right), \quad (61)$$

$$\rho' = O\left(\rho + \frac{\nu}{\sigma^3}\right) = O(\rho). \quad (62)$$

We consider the number of queries required to find an $\epsilon'$-SOSP of $f_\sigma$. We set the total iteration number to be:

$$T = 3\max\left\{\frac{4\Delta_f}{\eta\epsilon'^2}, \frac{\Delta_f\mathscr{T}}{\mathscr{F}}\right\}. \quad (63)$$

Similar to the proof in the previous section, we consider the two cases when a $\mathbf{x}_t$ is not local minima. Suppose for some iterations $\mathbf{x}_t$, we have $\nabla f_\sigma(\mathbf{x}_t) \leq \epsilon'$ and $\lambda_{\min}(\nabla^2 f(\mathbf{x}_t)) \leq -\sqrt{\rho'\epsilon'}$. The error probability of this assumption is given later. Under this assumption, the function value decreases for $\mathscr{F}$ after each $\mathscr{T}$ iterations according to Lemma 2.11. Therefore, the number of such iterations when Lemma 2.11 can be called is bounded by $T/3$ times, for otherwise the function value will decrease greater than $\Delta_f = f_\sigma(\mathbf{x}_0) - f_\sigma(\mathbf{x}^*)$, which is impossible. The failure probability is composed of two parts: the failure probability for estimating the gradient in Lemma 2.1 and the failure probability for Lemma 2.11. In each iteration, the probability of failure is bounded by $2\delta_0$ according to the union bound. The overall probability that Algorithm 3 fails to indicate a negative curvature is upper bounded by

$$\frac{T}{3} \cdot 2\delta_0 \leq \frac{\delta}{2} \quad (64)$$

for any $\chi$.

Excluding the iterations that Lemma 2.11 is applied, we still have $2T/3$ iterations left. We now consider the iterations $\mathbf{x}_t$ with large gradients $\nabla f_\sigma(\mathbf{x}_t) \geq \epsilon'$. According to Lemma 2.10, the function value decreases by at least $\eta\epsilon'^2/4$ with a probability of at least $1 - \delta_0$ in each iteration. Thus there

can be at most $T/3$ steps with large gradients, for otherwise, the function value will decrease greater than $\Delta_f$, which is impossible. The failure probability is bounded again by

$$\frac{T}{3} \cdot \delta_0 \leq \frac{\delta}{2}. \tag{65}$$

In summary, with probability at least $1 - \delta$, there are at most $T/3$ iterations within which the neighboring $\mathscr{T}$ iterations have small gradients but large negative curvatures, and at most $T/3$ iterations with large gradients. Therefore, the rest $T/3$ iterations must be $\epsilon$-SOSPs. The number of the queries is thus bounded by

$$T \leq \tilde{O}\left(\frac{\Delta_f \ell'}{\epsilon'^2} \cdot \chi^4 \cdot m\right) = \tilde{O}\left(\frac{d^{2.5}}{\epsilon^{3.5}} \cdot \mathrm{poly}(\Delta_f, \ell, \rho)\right). \tag{66}$$

$\square$

Theorem 3.7 provides a quantum upper bound $O(d^{2.5}/\epsilon^{3.5})$ in finding $\epsilon$-SOSPs of $F$ using noisy oracle $f$ at $\nu \leq O(\sqrt{\epsilon^3/\rho} \cdot 1/d)$ while the classical upper bound requires $O(d^4/\epsilon^5)$ queries (Jin et al., 2018a). The essence of the speedup lies in the quadratic reduction provided by the quantum mean estimation in the mini-batch size $m$.

## 3.2 FIRST-ORDER ALGORITHM AND PERFORMANCE GUARANTEE

Consider a pair of functions $(F, f)$ satisfying Assumption 1.2 with a relatively large noise strength such that Corollary 2.12 fails to apply. Recall that in the previous subsection we have implemented a Gaussian smoothing for the noisy zeroth-oracle defined in Definition 3.1. Now, we introduce the Gaussian smoothing of the noisy gradient, which is defined as:

$$\nabla f_\sigma(\mathbf{x}) := \mathbb{E}_{\mathbf{z}}\big[\nabla f(\mathbf{x} + \mathbf{z})\big], \quad \mathbf{z} \sim \mathcal{N}(0, \sigma^2 I). \tag{67}$$

After permutating the expectation operator and the gradient operator, we obtain

$$\nabla f_\sigma(\mathbf{x}) = \nabla \cdot \big(\mathbb{E}_{\mathbf{z}}[f(\mathbf{x} + \mathbf{z})]\big), \tag{68}$$

which indicates that $f_\sigma(\mathbf{x})$ is a Gaussian smoothing of $f$. Similar to Lemma 3.2, we deduce the following property of $\nabla f_\sigma(\mathbf{x})$, which originally appeared in Ref. (Jin et al., 2018a).

**Lemma 3.8** (Lemma 48, Ref. (Jin et al., 2018a)). *Assume the function pair $(F, f)$ satisfies Assumption 1.2. The Gaussian smoothing $\nabla f_\sigma$ of the noisy gradient $\mathbf{g} = \nabla f$ satisfies:*

- *$f_\sigma(\mathbf{x})$ is $O(\ell + \tilde{\nu}/\sigma)$-smooth and $O(\rho + \tilde{\nu}/\sigma^2)$-Hessian Lipschitz.*

- *The distances between the gradients and the Hessians of $f_\sigma$ and $F$ are bounded. In particular, we have $\|\nabla f_\sigma(\mathbf{x}) - \nabla F(\mathbf{x})\| \leq O(\rho d \sigma^2 + \tilde{\nu})$ and $\|\nabla^2 f_\sigma(\mathbf{x}) - \nabla^2 F(\mathbf{x})\| \leq O(\rho \sqrt{d}\sigma + \tilde{\nu}/\sigma)$, respectively.*

We can bound the deviation of $\nabla f_\sigma$ from $\nabla F$ and maintain the information about the local geometry of $F$, as well as guaranteeing that any $\tilde{\epsilon}$-SOSP $\mathbf{x}_{\mathrm{SOSP}}$ of $f_\sigma$ is also an $\epsilon$-SOSP of $F$, by choosing a suitable Gaussian smoothing parameter $\sigma$. We optimize $F$ through the Gaussian smooth $f_\sigma$. According to Lemma 3.8, the gradients and the eigenvalues of Hessians of $F$ are bounded by:

$$\|\nabla F(\mathbf{x}_{\mathrm{SOSP}})\| \leq \rho d \sigma^2 + \tilde{\nu} + \tilde{\epsilon}, \tag{69}$$

whereas

$$\lambda_{\min}(\nabla^2 F(\mathbf{x}_{\mathrm{SOSP}})) \geq \lambda_{\min}(\nabla^2 f_\sigma(\mathbf{x}_{\mathrm{SOSP}})) + \lambda_{\min}(\nabla^2 F(\mathbf{x}_{\mathrm{SOSP}}) - \nabla^2 f_\sigma(\mathbf{x}_{\mathrm{SOSP}}))$$

$$\geq -\sqrt{\left(\rho + \frac{\tilde{\nu}}{\sigma^2}\right)\tilde{\epsilon}} - \left\|\nabla^2 f_\sigma(\mathbf{x}_{\mathrm{SOSP}}) - \nabla^2 F(\mathbf{x}_{\mathrm{SOSP}})\right\|$$

$$\geq -\sqrt{\left(\rho + \frac{\tilde{\nu}}{\sigma^2}\right)\tilde{\epsilon}} - \left(\rho \sqrt{d}\sigma + \frac{\tilde{\nu}}{\sigma}\right). \tag{70}$$

To guarantee that any $\tilde{\epsilon}$-SOSP of $f_\sigma$ is an $\epsilon$-SOSP of $F$, we only need the following set of inequalities to be satisfied (up to constant factors).

$$\rho\sqrt{d}\sigma + \frac{\tilde{\nu}}{\sigma} \leq O(\sqrt{\rho\epsilon}), \tag{71}$$

$$\rho d\sigma^2 + \tilde{\nu} \leq O(\epsilon), \tag{72}$$

$$\left(\rho + \frac{\tilde{\nu}}{\sigma^2}\right)\tilde{\epsilon} \leq O(\rho\epsilon). \tag{73}$$

From (71) and (72), we can deduce that

$$\sigma \leq O\left(\sqrt{\frac{\epsilon}{\rho d}}\right), \tag{74}$$

$$\tilde{\nu} \leq \sqrt{\rho\epsilon}\sigma = O\left(\frac{\epsilon}{\sqrt{d}}\right), \tag{75}$$

$$\tilde{\epsilon} \leq \frac{\rho\epsilon}{\left(\rho + \frac{\tilde{\nu}}{\sigma^2}\right)} = O\left(\frac{\epsilon}{\sqrt{d}}\right). \tag{76}$$

Hence, an $O(\epsilon/\sqrt{d})$-SOSP of the Gaussian smoothing $f_\sigma$ is an $\epsilon$-SOSP of the target function $F$. Similar to Assumption 1.1, we now have to find an $O(\epsilon/\sqrt{d})$-SOSP for the Gaussian smoothing $f_\sigma$ through queries to first-order noisy oracle $\nabla f(\mathbf{x})$. To approximate the gradient $\nabla f_\sigma$, we sample from the stochastic gradient estimation $\mathbf{g}_\sigma(\mathbf{x}; \mathbf{z}) = \nabla f(\mathbf{x} + \mathbf{z})$, where $\mathbf{z} \sim \mathcal{N}(0, \sigma^2 I)$. As shown in Ref. (Jin et al., 2018a), the stochastic gradient has the following properties, which is similar to Lemma 3.3.

**Lemma 3.9** (Lemma 53, Ref. (Jin et al., 2018a)). *We denote $\mathbf{g}_\sigma(\mathbf{x}; \mathbf{z}) = \nabla f(\mathbf{x} + \mathbf{z})$ for a sample from the noisy oracle, where $\mathbf{z} \sim \mathcal{N}(0, \sigma^2 I)$. The following inequalities hold:*

$$\mathbb{E}_{\mathbf{z}}[\mathbf{g}_\sigma(\mathbf{x}; \mathbf{z})] = \nabla f_\sigma(\mathbf{x}), \tag{77}$$

$$\Pr\left[\|\mathbf{g}_\sigma(\mathbf{x}; \mathbf{z}) - \nabla f_\sigma(\mathbf{x})\| \geq t\right] \leq \exp\left(-Lt^2\right), \qquad \forall t > 0. \tag{78}$$

*The second inequality indicates that $\mathbf{g}_\sigma(\mathbf{x}; \mathbf{z})$ is a sub-Gaussian random variable with a tail $L$ (Recall that $L$ is the smoothness parameter of $f$ in Assumption 1.2).*

By a similar reduction to Lemma 3.4, we can deduce that the optimal sampling strategy requires

$$m \geq O\left(\frac{\sigma_0^2 d}{\epsilon^2}\log\left(\frac{d}{\delta}\right)\right) \tag{79}$$

queries to approximate $\nabla f_\sigma(\mathbf{x})$ with accuracy $\epsilon$, where $\sigma_0 = L$ according to Lemma 3.9. Quantumly, we can employ Lemma 3.6 and use only

$$m \geq O\left(\frac{\sqrt{d}\sigma_0}{\epsilon}\log\left(\frac{d}{\delta}\right)\right) \tag{80}$$

queries if given access to a quantum binary oracle defined in Eq. (54), which can be constructed by one query to the first-order oracle defined in Eq. (8). We propose a first-order version of PGD with stochastic gradient queries to the smoothed function $f_\sigma$ and quantum mean estimation in Algorithm 5.

---

**Algorithm 5:** First-order Perturbed Stochastic Gradient Descent with Quantum Mean Estimation.

---

**Input:** $\mathbf{x}_0$, learning rate $\eta$, noise ratio $r$, mini-batch size $m$

1: **for** $t = 0, 1, ..., T$ **do**

2:   Estimating the gradient $\tilde{\nabla}f_\sigma(\mathbf{x}_t)$ of $f_\sigma(\mathbf{x}_t)$ using quantum mean estimation and $m$ queries to $\mathbf{g}_\sigma(\mathbf{x}; \mathbf{z}) = \nabla f(\mathbf{x} + \mathbf{z})$ in the binary oracle

3:   $\mathbf{x}_{t+1} \leftarrow \mathbf{x}_t - \eta(\tilde{\nabla}f_\sigma(\mathbf{x}_t) + \xi_t), \xi_t$ uniformly $\sim B_0(r)$

4: **end for**

**Output:** $\mathbf{x}_T$

---

The goal of Algorithm 5 is to find an $O(\epsilon/\sqrt{d})$-SOSP of the Gaussian smoothing $f_\sigma$. The number of queries required can be bounded by the following theorem, which is the formal version of Theorem 1.6.

**Theorem 3.10** (Formal version of Theorem 1.6). *Suppose we have a target function $F$ and its noisy evaluation $f$ satisfying Assumption 1.2 with $\tilde{\nu} \leq O(\epsilon/\sqrt{d})$. With probability at least $1-\delta$, Algorithm 5 finds an $\epsilon$-SOSP of $F$ satisfying (4), using*

$$\tilde{O}\left(\frac{d^2}{\epsilon^3} \cdot \text{poly}(\Delta_f, \ell, \rho)\right) \tag{81}$$

*queries to $U_{\mathbf{g}}$ defined in Eq. (8), under the following choices of parameters:*

$$\eta = \frac{1}{\ell'}, \quad \delta_0 = \frac{\delta\epsilon'^2}{4\ell'\Delta_f}\chi^{-4}, \quad r = \epsilon'\chi^{-3}c^{-6}, \quad \mathscr{T} = \frac{\chi c}{\eta\sqrt{\rho'\epsilon'}}, \quad \mathscr{F} = \sqrt{\frac{\epsilon'^3}{\rho'}}\chi^{-3}c^{-5}, \tag{82}$$

*where $c$ is some large enough constant, $\Delta_f = f_\sigma(\mathbf{x}_0) - f_\sigma(\mathbf{x}^*)$ is the gap between the initial point $\mathbf{x}_0$ and the global minimum $\mathbf{x}^*$, and $\chi = \max\{1, \log(d\ell'\Delta_f/\rho'\epsilon'\delta_0)\}$. Here, $\ell'$, $\rho'$, and $\epsilon'$ have the same definition as in Theorem 3.7.*

Similar to Theorem 3.7, Theorem 3.10 presents a polynomial reduction in the query complexity of oracles using quantum mean estimation. In particular, Algorithm 5 requires only $\tilde{O}(d^2/\epsilon^3)$ queries to the first-order gradient oracles while its classical counterpart (Jin et al., 2018a) requires $\tilde{O}(d^3/\epsilon^4)$.

*Proof of Theorem 3.10.* Since an $\epsilon$-SOSP of the target function $F$ is an $O(\epsilon/\sqrt{d})$-SOSP of function $f_\sigma$, we only need to prove that Algorithm 5 converges to an $O(\epsilon/\sqrt{d})$-SOSP of $f_\sigma$ using $\tilde{O}(d^2/\epsilon^3)$ queries.

In each iteration, Algorithm 5 estimates the gradient $\nabla f_\sigma(\mathbf{x}_t)$ via quantum mean estimation while using $O(m)$ queries to the quantum evaluation oracle in each mini-batch. According to Eq. (80), we require the mini-batch size

$$m \geq O\left(\frac{\sqrt{d}L}{\epsilon/\sqrt{d}}\log\left(\frac{d}{\delta_0}\right)\right) = \tilde{O}\left(\frac{d}{\epsilon}\right) \tag{83}$$

to bound the error $\left\|\tilde{\nabla}f_\sigma(\mathbf{x}_t) - \nabla f_\sigma(\mathbf{x}_t)\right\| \leq \epsilon'/20 \leq O(\epsilon/\sqrt{d} \cdot (1/20))$ with probability at least $1 - \delta_0$.

Next, we consider the number of queries required to find an $\epsilon'$-SOSP of $f_\sigma$. We set the total iteration number to be

$$T = 3\max\left\{\frac{4\Delta_f}{\eta\epsilon'^2}, \frac{\Delta_f \mathscr{T}}{\mathscr{F}}\right\}. \tag{84}$$

We repeat the procedure in the proof of Theorem 3.7. Suppose for some iterations, the function has small gradients $\nabla f_\sigma(\mathbf{x}_t) \leq \epsilon'$ and large negative curvatures $\lambda_{\min}(\nabla^2 f_\sigma(\mathbf{x}_t)) \leq -\sqrt{\rho'\epsilon'}$. Under this assumption, the function value will decrease for $\mathscr{F}$ after each $\mathscr{T}$ iterations according to Lemma 2.11. Therefore, the number of such iterations is bounded by $T/3$, for otherwise the function value will decrease greater than $\Delta_f = f_\sigma(\mathbf{x}_0) - f_\sigma(\mathbf{x}^*)$, which is impossible. The failure probability of the above argument is bounded by

$$\frac{T}{3} \cdot \delta_0 \leq \delta/2. \tag{85}$$

Except for the iterations where Lemma 2.11 is applied, we still have $2T/3$ iterations left. We now consider the iterations $\mathbf{x}_t$ with large gradients, i.e., $\nabla f_\sigma(\mathbf{x}_t) \geq \epsilon'$. According to Lemma 2.10, the function value decreases by at least $\eta\epsilon^2/4$ with a probability of at least $1 - \delta_0$. Therefore, there can be at most $T/3$ steps with large gradients, for otherwise, the function value will decrease greater than $\Delta_f$, which is impossible. The failure probability of the above argument is again bounded by

$$\frac{T}{3} \cdot \delta_0 \leq \delta/2. \tag{86}$$

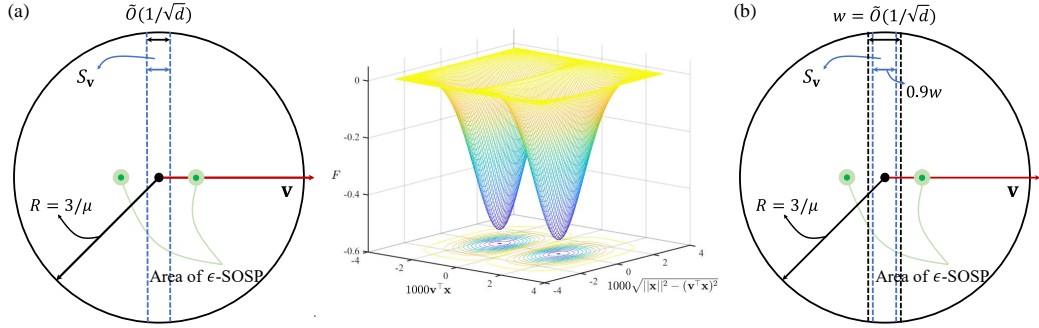

Figure 1: (a) The Sketch of the hard instance function for Theorem 1.7. The left figure illustrates the construction in the domain and the right figure shows a two-dimensional example. (b) The construction of the hard instance function for Theorem 1.10.

Therefore, we can deduce that with probability at least $1 - \delta$, there are at most $T/3$ iterations within which the neighboring $\mathscr{T}$ iterations have small gradients but large negative curvatures, and at most $T/3$ iterations with large gradients. Hence, the rest $T/3$ iterations must be $\epsilon$-SOSPs. The number of queries can be bounded by

$$T \leq \tilde{O}\left(\frac{\Delta_f \ell}{\epsilon'^2} \cdot \chi^4 \cdot m\right) = \tilde{O}\left(\frac{d^2}{\epsilon^3} \cdot \text{poly}(\Delta_f, \ell, \rho)\right). \tag{87}$$

$\square$

## 4 CLASSICAL AND QUANTUM LOWER BOUNDS IN $d$

In this section, we prove the query complexity lower bounds in the dimension $d$ of the input. Intuitively, the lower bound is obtained by constructing a hard instance and calculating its worst-case query complexity.

### 4.1 QUASI-POLYNOMIAL LOWER BOUND FOR QUANTUM ZEROTH-ORDER METHODS

The constructions of our hard instances (as shown in Figure 1 (a)) are inspired by the idea that originally appeared in Ref. (Jin et al., 2018a). We first consider a "scale free" version of function pair $(F, f)$, where we assume $\rho = 1$ and $\epsilon = 1$. Denote $\sin \mathbf{x} := (\sin(x_1), ..., \sin(x_d))$ and $\mathbb{I}(A)$ as the indicator function that has value 1 when $A$ is true and 0 otherwise. We set the constant $\mu = 300$ and define the target function as

$$F(\mathbf{x}) := h(\sin \mathbf{x}) + \|\sin \mathbf{x}\|^2, \tag{88}$$

where $h(\mathbf{x}) := h_1(\mathbf{v}^\top \mathbf{x}) \cdot h_2\left(\sqrt{\|\mathbf{x}\|^2 - (\mathbf{v}^\top \mathbf{x})^2}\right)$, and

$$h_1(x) = g_1(\mu x), \quad g_1(x) = (-16|x|^5 + 48x^4 - 48|x|^3 + 16x^2) \cdot \mathbb{I}\{|x| < 1\}, \tag{89}$$

$$h_2(x) = g_2(\mu x), \quad g_2(x) = (3x^4 + 8|x|^3 + 6x^2 - 1) \cdot \mathbb{I}\{|x| < 1\}. \tag{90}$$

Here, the vector $\mathbf{v}$ is uniformly chosen from the $d$-dimensional unit sphere. In addition, we can split the domain into different regions upon which analysis and constructions are made separately:

- "ball" $\mathbb{B}(0, 3/\mu) = \{\mathbf{x} \in \mathbb{R}^d : \|x\| \leq 3/\mu\}$ is the $d$-dimensional hyperball with radius $3/\mu$.

- "band" $S_{\mathbf{v}} = \{\mathbf{x} \in \mathbb{B}(0, 3/\mu) : \langle \sin \mathbf{x}, \mathbf{v} \rangle \leq \log d/\sqrt{d}\}$.

We provide the landscape of $F$ and the region division in Figure 1 (a). The above construction happens within a hyperball and we cannot fill the entire space $\mathbb{R}^d$ with hyperballs. Therefore, we embed this hyperball into a hypercube and add two regions.

- "hypercube" $H = [-\pi/2, \pi/2]^d$ is the $d$-dimensional hypercube with length $\pi$.
- "padding" $S_2 = H - \mathbb{B}(0, 3/\mu)$.

With the above construction, we can fill the space $\mathbb{R}^d$ using these hypercubes. Meanwhile, the noisy function $f$ is defined as

$$f(\mathbf{x}) = \begin{cases} \|\sin \mathbf{x}\|^2, & \mathbf{x} \in S_{\mathbf{v}}, \\ F(\mathbf{x}), & \mathbf{x} \notin S_{\mathbf{v}}. \end{cases} \tag{91}$$

The "band" region $S_{\mathbf{v}}$ is known as the non-informative region as any query to $f$ in this area will obtain no information regarding $\mathbf{v}$. Intuitively, the metric of the non-informative area approaches 1 as $d$ increases according to the measure of concentration. It is hard for any algorithm (both classical and quantum) to find a point out of this region. In particular, the probability of classically querying a point on $S_{\mathbf{v}}$ is bounded below by

$$\frac{\text{Area}(S_{\mathbf{v}})}{\text{Area}(\mathbb{B}(0, 3/\mu))} \geq 1 - O(d^{-\log d}) \tag{92}$$

according to Lemma A.4 in Appendix A.The following properties hold for function pair according to Ref. (Jin et al., 2018a).

**Lemma 4.1** (Lemma 33, Ref. (Jin et al., 2018a))**.** *The function pair $(F, f)$ defined in (88) and (91) above satisfies:*

- *The value of $f$ in the non-informative region $S_{\mathbf{v}}$ is independent of $\mathbf{v}$.*
- $\sup_{x \in S_{\mathbf{v}}} \|f - F\|_{\infty} \leq \tilde{O}(1/d)$.
- *$F$ has no $\epsilon$-SOSP in the non-informative region $S_{\mathbf{v}}$.*
- *$F$ is $O(d)$-bounded, $O(1)$-Hessian Lipschitz, and $O(1)$-gradient Lipschitz.*

The hard instance in the above Lemma 4.1 has realized the $\tilde{O}(1/d)$ factor for the noise bound in Theorem 1.7 and introduced the non-informative area. The next step is to scale the hard instance to reach the lower bound with correct dependencies on $\rho$ and $\epsilon$. Given $\epsilon > 0$ and $\rho > 0$, we define the scaling functions

$$\tilde{F}(\mathbf{x}) := \epsilon r F\left(\frac{\mathbf{x}}{r}\right), \tag{93}$$

$$\tilde{f}(\mathbf{x}) := \epsilon r f\left(\frac{\mathbf{x}}{r}\right), \tag{94}$$

where $r = \sqrt{\epsilon/\rho}$, and the functions $F, f$ are defined in (88) and (91), respectively. The scaled regions corresponding to $\tilde{F}$ and $\tilde{f}$ are:

- "ball" $\mathbb{B}(0, 3r/\mu) = \{\mathbf{x} \in \mathbb{R}^d : \|x\| \leq 3/\mu\}$ is the $d$-dimensional hyperball with radius $3r/\mu$.
- "band" $\tilde{S}_{\mathbf{v}} = \{\mathbf{x} \in \mathbb{B}(0, 3r/\mu) : \langle \sin(\mathbf{x}/r), \mathbf{v} \rangle \leq \log d / \sqrt{d}\}$.

According to Lemma 4.1, the function pair $(\tilde{F}, \tilde{f})$ satisfies Assumption 1.1 with $\nu = \tilde{\Theta}(\sqrt{\epsilon^3/\rho} \cdot 1/d)$, upon which we prove our quantum lower bound for finding an $\epsilon$-SOSP of the target function $\tilde{F}$ in (93) taking queries to the noisy function $\tilde{f}$ in (94). Formally, we provide the following theorem.

**Theorem 4.2** (Formal version of Theorem 1.7)**.** *For any $B > 0, \ell > 0, \rho > 0$, there exists an $\epsilon_0 = \Theta(\min\{\ell^2/\rho, (B^2\rho/d^2)^{1/3}\})$ such that for any $\epsilon \in (0, \epsilon_0]$, the function pair $(\tilde{F}, \tilde{f})$ defined in (93) and (94) satisfies Assumption 1.1 with $\nu = \tilde{\Theta}(\sqrt{\epsilon^3/\rho} \cdot 1/d)$, and any quantum algorithm that only queries a quasi-polynomial $O(d^{\log(d)})$ times to the zeroth-order quantum oracle $U_{\tilde{f}}$ will fail with high probability to find an $\epsilon$-SOSP of $\tilde{F}$.*

To prove Theorem 4.2, we introduce some lemmas to construct a reduction of the problem. In particular, our goal is to transform the quantum lower bound on the unstructured search problem (Bennett

et al., 1997; Nayak & Wu, 1999) into a lower bound for the problem of finding an $\epsilon$-SOSP of $\tilde{F}$ considered in Theorem 4.2. We discretize the problem via the following results on distributing exponentially many points on $\mathbb{S}^{d-1}$ in a uniform way such that the distances between each pair of points are at least $\delta$.

**Lemma 4.3** (Lemma D.1, Ref. (Liu et al., 2023)). *For any constant $\delta \in \left(0, \frac{\log d}{2\sqrt{d}}\right)$, there exists a set* $\Gamma = \{\mathbf{y}_1, \mathbf{y}_2, \ldots, \mathbf{y}_N\}$ *of $N$ unit vectors in $\mathbb{R}^d$ such that*

- $\forall \mathbf{y}_i \neq \mathbf{y}_j \in \Gamma$, $\|\mathbf{y}_i - \mathbf{y}_j\| \geq \delta$;

- $\forall \mathbf{z} \in \mathbb{S}^{d-1}$, *there exists an $\mathbf{y}_i \in \Gamma$ such that $\|\mathbf{z} - \mathbf{y}_i\| \leq \delta$;*

- $N \geq \left(\frac{1}{2\delta} + \frac{1}{2}\right)^d - \left(\frac{1}{2\delta} - \frac{1}{2}\right)^d$.

Inspired by Ref. (Liu et al., 2023), we consider the following unstructured search problem which can be reduced to finding an $\epsilon$-SOSP of $\tilde{F}$ with polynomial overhead.

*Problem* 1. Consider a set $\Gamma$ of $N$ unit vectors in $\mathbb{R}^d$ satisfying Lemma 4.3, for an unknown unit vector $\mathbf{v} \in \Gamma$, we define $q \colon \Gamma \to \mathbb{R}^d$ as follows:

$$q(\mathbf{x}) := \begin{cases} \mathbf{v}, & \langle \sin \mathbf{x}, \mathbf{v} \rangle > \frac{\log d}{2\sqrt{d}}, \\ \mathbf{0}, & \text{otherwise.} \end{cases} \tag{95}$$

The goal is to find $\mathbf{v}$ only with access to values of $q$.

We now present the reduction from Problem 1 to the problem of finding an $\epsilon$-SOSP of $\tilde{F}$ under the setting of Theorem 4.2. To make the reduction more straightforward, we additionally introduce an intermediate function $\hat{q}(\mathbf{x}) \colon \mathbb{B}(\mathbf{0}, 3/\mu) \to \mathbb{R}$ between $q$ and $f$. In particular, for any $\mathbf{x} \in \mathbb{B}(\mathbf{0}, 3/\mu)$, we use $\hat{\mathbf{y}}(\mathbf{x})$ to denote the vector $\mathbf{y}_i$ in $\Gamma$ such that the distance $\|\mathbf{x}/\|\mathbf{x}\| - \mathbf{y}_i\|$ is minimized. If more than one of such vectors exists, we choose the one with the smallest lower index. We define $\hat{q}(\mathbf{x})$ as

$$\hat{q}(\mathbf{x}) := \begin{cases} \|\sin \mathbf{x}\|^2, & q(\hat{\mathbf{y}}(\mathbf{x})) = \mathbf{0}, \\ F(\mathbf{x}), & \text{otherwise.} \end{cases} \tag{96}$$

Similar to $f$, $\hat{q}$ also has a large "non-informative" region $\hat{S}_{\mathbf{v}}$ where the function value equals $\|\sin \mathbf{x}\|^2$ and reveals no information about $\mathbf{v}$. Quantitatively, we can observe that $\hat{S}_{\mathbf{v}} = \{\mathbf{y} \in \mathbb{B}(\mathbf{0}, 3/\mu) : g(\hat{\mathbf{y}}(\mathbf{x})) = \mathbf{0}\}$, and $\hat{q}$ has the following properties.

**Lemma 4.4.** *The function $\hat{q}(\mathbf{x})$ defined in (96) has the following properties:*

- *One query to $\hat{q}$ can be implemented using one query to $q$.*

- *Its non-informative region $\hat{S}_{\mathbf{v}} = \{\mathbf{y} \in \mathbb{B}(\mathbf{0}, 3/\mu) : q(\hat{\mathbf{y}}(\mathbf{x})) = \mathbf{0}\}$ is a subset of $S_{\mathbf{v}}$, which is the non-informative region of $f$ defined in (91).*

- *For any $\mathbf{x} \in \mathbb{B}(\mathbf{0}, 3/\mu) - S_{\mathbf{v}}$, we have $\hat{q}(\mathbf{x}) = f(\mathbf{x})$.*

*Proof.* For the first property, one can observe that for any $\mathbf{x} \in \mathbb{B}(\mathbf{0}, 3/\mu)$, $\hat{q}(\mathbf{x})$ can be expressed as

$$\hat{q}(\mathbf{x}) = \|\sin \mathbf{x}\|^2 + h(\sin \mathbf{x}) \tag{97}$$

$$= \|\sin \mathbf{x}\|^2 + h_1(\langle q(\hat{\mathbf{y}}(\mathbf{x})), \mathbf{x} \rangle) \cdot h_2\left(\sqrt{\|\mathbf{x}\|^2 - \langle q(\hat{\mathbf{y}}(\mathbf{x})), \mathbf{x} \rangle^2}\right), \tag{98}$$

which can be implemented using one query to $q(\mathbf{x})$.

For the second property, $\forall \mathbf{x} \in \hat{S}_{\mathbf{v}}$, the corresponding $\hat{\mathbf{y}}(\mathbf{x})$ satisfies $\langle \hat{\mathbf{y}}(\mathbf{x}), \mathbf{v} \rangle \leq \frac{\log d}{2\sqrt{d}}$. Since $\|\mathbf{x}/\|\mathbf{x}\| - \hat{y}(\mathbf{x})\| \leq \delta = \frac{\log d}{2\sqrt{d}}$ by Lemma 4.3, we deduce that $\langle \mathbf{x}, \mathbf{v} \rangle \leq \langle \mathbf{x}/\|\mathbf{x}\|, \mathbf{v} \rangle \leq \log d/\sqrt{d}$, indicating $\mathbf{x} \in S_{\mathbf{v}}$.

The third property can be directly obtained from the second property. $\qquad \square$

Next, we present the reduction from Problem 1 to the problem of finding an $\epsilon$-SOSP of $F$ under the setting of Theorem 4.2 through the following lemma.

**Lemma 4.5.** *Under the setting of Theorem 4.2, with polynomial overhead Problem 1 can be reduced to the problem of finding an $\epsilon$-SOSP of $F$ defined in (88) for any $\epsilon \in (0, \epsilon_0]$, using access to values of $f$ defined in (91).*

*Proof.* Since one query to $\hat{q}$ can be implemented using one query to $q$ by Lemma 4.4, Problem 1 can be reduced to the problem of finding an $\mathbf{x} \in \mathbb{B}(0, 3/\mu)$ satisfying $\hat{q}(\mathbf{x}) \neq \|\sin \mathbf{x}\|^2$, or equivalently, $\mathbf{x} \in \mathbb{B}(0, 3/\mu) - \hat{S}_{\mathbf{v}}$, with only access to values of $\hat{q}$.

By Lemma 4.4, any $\epsilon$-SOSP of $F$, denoted $\mathbf{x}_{\text{SOSP}}^F$, satisfies

$$\mathbf{x}_{\text{SOSP}}^F \in \mathbb{B}(0, 3/\mu) - S_{\mathbf{v}} \subseteq \mathbb{B}(0, 3/\mu) - \hat{S}_{\mathbf{v}}. \tag{99}$$

Therefore, we have reduced Problem 1 to the nonconvex optimization task. $\qquad\square$

We can scale the "scale-free" hard instance $(F, f)$ to the hard instance $(\tilde{F}, \tilde{f})$ satisfying Assumption 1.1 using (93) and (94). In particular, we introduce the following lemma, which originally appeared in Ref. (Jin et al., 2018a).

**Lemma 4.6** (Appendix C.2, Ref. (Jin et al., 2018a)). *For any $B > 0, \ell > 0, \rho > 0$, there exists an $\epsilon_0 = \Theta(\min\{\ell^2/\rho, (B^2\rho/d^2)^{1/3}\})$ such that for any $\epsilon \in (0, \epsilon_0]$, there exists a function pair $(\tilde{F}, \tilde{f})$ satisfying the assumptions in Assumption 1.1 with $\nu = \tilde{\Theta}(\sqrt{\epsilon^3/\rho} \cdot (1/d))$, so that with constant overhead the problem of finding an $\epsilon$-SOSP of $F$ defined in (88) using only access to values of $f$ defined in (91) can be reduced to the problem of finding an $\epsilon$-SOSP of $\tilde{F}$ using only access to values of $\tilde{f}$.*

Equipped with Lemma 4.6,, we prove Theorem 4.2

*Proof.* According to Lemma 4.5 and Lemma 4.6, there exists a function pair $(\tilde{F}, \tilde{f})$ satisfying Assumption 1.1 with $\nu = \tilde{\Theta}(\sqrt{\epsilon^3/\rho} \cdot (1/d))$, such that with polynomial overhead Problem 1 can be reduced to the problem of finding an $\epsilon$-SOSP of $\tilde{F}$ using only access to values of $\tilde{f}$.

We divide the $N$ unit vectors in $\Gamma$ into two parts.

$$\Gamma_1 = \{\mathbf{x} \in \Gamma : q(\mathbf{x}) = \mathbf{v}\} \tag{100}$$
$$\Gamma_0 = \{\mathbf{x} \in \Gamma : q(\mathbf{x}) = \mathbf{0}\}. \tag{101}$$

We denote the size of the two parts as $N_1 = |\Gamma_1|$ and $N_0 = |\Gamma_2|$. Our goal is to find any $\mathbf{x}$ in the set $\Gamma_1$. Intuitively, under limitation $\delta \to 0$ and $\delta < \frac{\log d}{2\sqrt{d}}$, we can deduce that

$$\frac{N_1}{N} \sim \frac{2\text{Area}(\mathbb{B}(0, 3/\mu) - S_{\mathbf{v}})}{\text{Area}(\mathbb{B}(0, 3/\mu))} = O(d^{-\log d}). \tag{102}$$

We bound the deviation of $N_1/N$ from $2\text{Area}(\mathbb{B}(0, 3/\mu) - S_{\mathbf{v}})/\text{Area}(\mathbb{B}(0, 3/\mu))$ when $\delta \neq 0$. For $S_{\mathbf{v}}$, we consider $S'_{\mathbf{v}}$ the area that is the "band" area along $\mathbf{v}$ within $\log d/2\sqrt{d} - \delta$ from $0$. The border area of $S_{\mathbf{v}}$ out of $S'_{\mathbf{v}}$ contains ignorable $O(\exp(-d))$ directions compared to $S_{\mathbf{v}}$ when $\delta \ll 1/\sqrt{d}$.

Even if we consider the boundary above, we can still derive the upper bound for $N_1/N$

$$\frac{N_1}{N} \leq \frac{2\text{Area}(\mathbb{B}(0, 3/\mu) - S'_{\mathbf{v}})}{\text{Area}(\mathbb{B}(0, 3/\mu))} = O(d^{-\log d}). \tag{103}$$

The inequality comes from the fact that $\delta \ll \text{poly}(1/d)$ and the boundary area can only bring exponential deviation from the expectation value $2\text{Area}(\mathbb{B}(0, 3/\mu) - S_{\mathbf{v}})/\text{Area}(\mathbb{B}(0, 3/\mu))$.

With the fact that $N_1/N \leq O(d^{-\log d})$ is quasi-polynomially small, any quantum algorithm that solves Problem 1 with high probability requires query complexity at least $\Omega(\sqrt{N/N_1}) = \Omega(d^{\log d})$ (Nayak & Wu, 1999; Nielsen & Chuang, 2010). $\qquad\square$

## 4.2 INFORMATION-THEORETIC LIMITATION OF QUANTUM ZEROTH-ORDER METHODS

When the noise between $F$ and $f$ keeps increasing under Assumption 1.1, it can erase the landscape of target function $F$ in the worst case. As a result, when the noise rate is larger than a certain threshold, for any quantum algorithm we can find a hard instance on which it will fail with a large probability. We consider the same target function defined in Eq. (88) with a different noisy function $f$. We apply the scaling in (93) and (94) as

$$F(\mathbf{x}) = h(\sin \mathbf{x}) + \|\sin \mathbf{x}\|^2, \tag{104}$$

$$f(\mathbf{x}) = \|\sin \mathbf{x}\|^2, \tag{105}$$

$$\tilde{F}(\mathbf{x}) = \epsilon r F\left(\frac{\mathbf{x}}{r}\right), \qquad \tilde{f}(\mathbf{x}) = \epsilon r f\left(\frac{\mathbf{x}}{r}\right). \tag{106}$$

Similar to Lemma 4.1, the following properties hold for the above hard instance $(\tilde{F}, \tilde{f})$.

**Lemma 4.7** (Appendix D.2, Ref. (Jin et al., 2018a)). *The function pair $(F, f)$ defined in (106) above satisfies:*

- *The values of $\tilde{f}$ in $\mathbb{B}(0, 3/\mu)$ are independent of $\mathbf{v}$.*

- $\sup_{x \in \mathbb{B}(0, 3/\mu)} \left\| \tilde{f} - \tilde{F} \right\|_\infty \leq \tilde{O}(\epsilon^{1.5}/\sqrt{\rho})$.

- *$F$ is $B$-bounded, $O(\rho)$-Hessian Lipschitz, and $O(\ell)$-gradient Lipschitz.*

We derive the following result concerning the hard instance $(\tilde{F}, \tilde{f})$ in Eq. (106), which is the formal version of Theorem 1.8.

**Theorem 4.8** (Formal version of Theorem 1.8). *For any $B > 0, \ell > 0, \rho > 0$, any $\epsilon \in (0, \epsilon_0]$ for some $\epsilon_0 = \Theta(\min\{\ell^2/\rho, (B^2\rho/d^2)^{1/3}\})$, and any possible quantum algorithm, we can choose a function pair $(\tilde{F}, \tilde{f})$ with $\mathbf{v}$ defined in (106) satisfying Assumption 1.1 with $\nu = \tilde{\Theta}(\sqrt{\epsilon^3/\rho})$ such that the quantum algorithm will fail with high probability to find an $\epsilon$-SOSP of $\tilde{F}$ given only access to $U_{\tilde{f}}$.*

*Proof.* As $\tilde{f}$ is independent of $\mathbf{v}$, neither quantum nor classical query can reveal any information on $\mathbf{v}$ and the $\epsilon$-SOSP of $\tilde{F}$. Any solutions output by any algorithm will be independent of $\mathbf{v}$ with probability 1. Therefore, the probability of success must be independent of the number of iterations, which indicates that any algorithm cannot output an $\epsilon$-SOSP with probability more than a constant. Specifically, no algorithm can do better than random guessing in $\mathbb{B}(0, 3/\mu)$ within this construction. $\square$

We remark that the noise bound and its underlying intuition in the quantum case is the same as the classical case (Jin et al., 2018a). However, Theorem 4.8 only indicates the classical and quantum algorithms have the same worst-case lower bound for some level of noise strength, and there is still a possible quantum speedup for solving specific instances $(F, f)$. In Appendix D, we show a concrete example in which quantum tunneling walk (Liu et al., 2023) can find an $\epsilon$-SOSP of $F$ using polynomial queries and proper initial states containing information of the landscape, while any classical algorithm requires exponential queries even given access to such information.

## 4.3 INFORMATION-THEORETIC LIMITATION OF CLASSICAL ZEROTH-ORDER METHODS

Here, we use an information-theoretic approach to prove Theorem 1.9, which indicates that if a classical algorithm can find an $\epsilon$-SOSP of $F$ for any function pair $(F, f)$ satisfying Assumption 1.1 with $\nu = \Omega(1/\operatorname{poly}(d))$, the query complexity is bounded by $\Omega(d/\log d)$.

In particular, we consider the target function $\tilde{F}$ defined by (88) and (93). As $\tilde{F}$ is $\rho$-Hessian Lipschitz and $\ell$-smooth, we can estimate the vector $\mathbf{v}$ within $\operatorname{poly}(\epsilon)$ distance under infinity norm[3], which requires $d \log(1/\epsilon)$ bits of information. Furthermore, as the noisy zeroth-order oracle $\tilde{f}$ contains

---

[3]Here, we have ignored the dependence on $\ell$ and $\rho$ and regarded them as constants

noise of strength $\nu = \Omega(1/\operatorname{poly}(d))$, each classical query reveals at most $O(\log(1/\delta))$ bits of information (Chakrabarti et al., 2020). Therefore, any classical algorithm has to take at least

$$\Omega\left(\frac{d\log(1/\epsilon)}{\log(1/\delta)}\right) = \Omega\left(\frac{d}{\log d}\right) \tag{107}$$

queries to the noisy oracle.

Moreover, Ref. (Chakrabarti et al., 2020) shows even estimating a sub-gradient for a Lipschitz convex function within infinity norm $\epsilon = O(1/\operatorname{poly}(d))$ using zeroth-order oracle $f$ with noise rate $\nu = \Omega(1/\operatorname{poly}(d))$ requires $\Omega(d/\log d)$ classical queries. However, Jordan's algorithm enables simultaneous queries to different points using a single oracle query. Thus, only $O(1)$ query to quantum oracle is required to estimate the sub-gradient, which provides the exponential speedup for quantum algorithms.

### 4.4 LOWER BOUND FOR FIRST-ORDER METHODS

We now derive the lower bound for classical and quantum algorithms in finding $\epsilon$-SOSPs of target function $F$ through noisy function satisfying Assumption 1.2. We propose the following two theorems as two parts for the formal version of Theorem 1.10. In the first part, we consider the case of adding noise such that for any quantum or classical algorithm we can find a hard instance that will make the algorithm fail with high probability, which is an analog of Theorem 4.8 under Assumption 1.2. Formally, we have the following theorem

**Theorem 4.9** (Formal version of Theorem 1.10, Part I). *For any $B > 0, \ell > 0, \rho > 0$, any $\epsilon \in (0, \epsilon_0]$ for some $\epsilon_0 = \Theta(\min\{\ell^2/\rho, (B^2\rho/d^2)^{1/3}\})$, and any quantum or classical algorithm, we choose function pair $(\tilde{F}, \tilde{f})$ with $\mathbf{v}$ defined in (106) satisfies Assumption 1.2 with $\tilde{\nu} = \tilde{\Theta}(\epsilon/\sqrt{d})$ such that it will fail with high probability to find any $\epsilon$-SOSP of $\tilde{F}$ given only access to $U_{\mathbf{g}}$.*

*Proof.* We consider the same target function $\tilde{F}$ and noisy function $\tilde{f}$ in (106). Except from the properties in Lemma 4.7, the noisy function $\tilde{f}$ is also smooth and $\left\|\nabla\tilde{f} - \nabla\tilde{F}\right\| \leq \tilde{\Theta}(\epsilon/\sqrt{d})$. Therefore, the hard instance in (106) also satisfies Assumption 1.2 with $\tilde{\nu} = \tilde{\Theta}(\epsilon/\sqrt{d})$. According to Theorem 4.8, for any *quantum* and *classical* algorithm, we choose function pair $(\tilde{F}, \tilde{f})$ with $\mathbf{v}$ such that it will success with probability no more than a constant to find any $\epsilon$-SOSP of $\tilde{F}$ given only access to $U_{\tilde{\mathbf{g}}}$. $\qquad\square$

Next, we consider the quasi-polynomial lower bound under Assumption 1.2. Unlike Theorem 4.2, we cannot directly apply the hard instance $(F, f)$ defined in (93) and (94) because $f$ is not differentiable (or more strictly, not continuous). To address this problem, we construct a different noisy function $f$ (as shown in Figure 1 (b)). We start with the "scale free" version. We still set the $\mu = 300$ and define the target function $F(\mathbf{x}) = h(\sin\mathbf{x}) + \|\sin\mathbf{x}\|^2$, which is the same with (88). We uniformly choose $\mathbf{v}$ and divide the "hypercube" into different regions as

- "hypercube" $H = [-\pi/2, \pi/2]^d$ is the $d$-dimensional hypercube with length $\pi$.
- "ball" $\mathbb{B}(0, 3/\mu) = \{\mathbf{x} \in \mathbb{R}^d : \|x\| \leq 3/\mu\}$ is the $d$-dimensional ball with radius $3/\mu$.
- "band" $S = \{\mathbf{x} \in \mathbb{B}(0, 3/\mu) : \langle\sin\mathbf{x}, \mathbf{v}\rangle \leq w\}$ with $w = O(\log d/\sqrt{d})$.
- "non-informative band" $S_{\mathbf{v}} = \{\mathbf{x} \in \mathbb{B}(0, 3/\mu) : \langle\sin\mathbf{x}, \mathbf{v}\rangle \leq 0.9w\}$.
- "padding" $S_2 = H - \mathbb{B}(0, 3/\mu)$.

Meanwhile, the noisy function $f$ is defined as

$$f(\mathbf{x}) = \begin{cases} \|\sin\mathbf{x}\|^2, & \mathbf{x} \in S_{\mathbf{v}}, \\ \|\sin\mathbf{x}\|^2 + h_3(\mathbf{x}) \cdot h_2(\sqrt{\|\sin\mathbf{x}\|^2 - (\mathbf{v}^\top\sin\mathbf{x})^2}), & \mathbf{x} \in S - S_{\mathbf{v}} \\ F(\mathbf{x}), & \mathbf{x} \notin S, \end{cases} \tag{108}$$

where

$$h_3(\mathbf{x}) = h_1(\mathbf{v}^\top\sin(10\mathbf{x} - 9w\mathbf{v}/2)). \tag{109}$$

By the chain rule of gradients we deduce the following lemma:

**Lemma 4.10.** *The function pair* $(F, f)$ *defined in (88) and (108) satisfies:*

- *The value of $f$ in the non-informative region $S_{\mathbf{v}}$ is independent of $\mathbf{v}$. $f$ is differentiable and satisfies the Lipshitz condition.*

- $\sup_{x \in S} \|\nabla f - \nabla F\|_{\infty} \leq \tilde{O}(1/\sqrt{d}).$

- *$F$ has no $\epsilon$-SOSP in the non-informative region $S_{\mathbf{v}}$.*

- *$F$ is $O(d)$-bounded, $O(1)$-Hessian Lipschitz, and $O(1)$-gradient Lipschitz.*

We apply the scaled version of $f$ as

$$\tilde{f} = \epsilon r f \left(\frac{\mathbf{x}}{r}\right). \tag{110}$$

Based on the hard instance $(\tilde{F}, \tilde{f})$ defined in (93), (108) and (110). We propose the following theorem.

**Theorem 4.11** (Formal version of Theorem 1.10, Part II). *For any $B > 0, \ell > 0, \rho > 0$, there exists $\epsilon_0 = \Theta(\min\{\ell^2/\rho, (B^2\rho/d^2)^{1/3}\})$ such that for any $\epsilon \in (0, \epsilon_0]$, the function pair $(\tilde{F}, \tilde{f})$ defined in (93), (108) and (110) satisfies Assumption 1.2 with $\tilde{\nu} = \tilde{\Theta}(\epsilon/\sqrt{d})$, and any quantum or classical algorithm that only requires a quasi-polynomial number $\Omega(d^{\log(d)})$ queries to function values of $\tilde{f}$ will fail with high probability, to find an $\epsilon$-SOSP of $\tilde{F}$.*

*Proof.* According to Lemma 4.10 and Lemma 4.6, the width of non-informative band is $0.9w = \tilde{\Theta}(\epsilon/\sqrt{d})$. Then we can directly apply the similar procedure when we prove Theorem 1.7 as the non-informative band has the same width scaling. $\square$

## 5 CLASSICAL AND QUANTUM LOWER BOUNDS IN $\epsilon$

In this section, we prove classical randomized lower bounds and quantum lower bounds in $\epsilon$ for finding an $\epsilon$-SOSP of an objective function $F$ given access to noisy classical or quantum zeroth- or first-order oracles, where $F \colon \mathbb{R}^d \to \mathbb{R}$ is $\ell$-smooth and $\rho$-Hessian Lipschitz, and satisfies

$$F(\mathbf{0}) - \inf_{\mathbf{x}} F(\mathbf{x}) \leq \Delta, \tag{111}$$

for some constant $\Delta$.

### 5.1 HARD INSTANCE FOR DETERMINISTIC CLASSICAL ALGORITHMS

We first discuss the construction and intuition of hard instances upon which we can obtain lower bound results for deterministic classical algorithms. Consider the toy example proposed by Nesterov (Nesterov, 2003),

$$F(\mathbf{x}) \coloneqq \frac{1}{2}(x_1 - 1)^2 + \frac{1}{2}\sum_{i=1}^{T-1}(x_i - x_{i+1})^2, \tag{112}$$

whose gradient satisfies that

$$\forall 1 < i < T, \quad \nabla_i F(\mathbf{x}) = \mathbf{0} \Leftrightarrow x_{i-1} = x_i = x_{i+1}. \tag{113}$$

Then, if we query the gradient of $F$ at a point with only its first $t$ entries being nonzero, the derivative can only reveal information about the $(t + 1)$th direction, if one does not have knowledge about the directions of the coordinate axes. Formally, such properties can be summarized to consist of the concept of *zero-chain*, which is defined as follows.

**Definition 5.1** (Definition 3, Ref. (Carmon et al., 2020)). A function $F \colon \mathbb{R}^d \to \mathbb{R}$ is called a zero-chain if for every $\mathbf{x} \in \mathbb{R}^d$,

$$\operatorname{supp}\{\mathbf{x}\} \subseteq \{1, \ldots, i-1\} \Rightarrow \operatorname{supp}\{\nabla f(\mathbf{x})\} \subseteq \{1, \ldots, i\}, \tag{114}$$

where the support of a vector $\mathbf{y} \in \mathbb{R}^d$ is defined as

$$\operatorname{supp}\{\mathbf{y}\} \coloneqq \{i \in [d] \mid y_i \neq 0\}. \tag{115}$$

From an algorithmic perspective, if we encode $F(\mathbf{x})$ or any other $T$-dimensional zero chain into a $d$-dimensional space with $d \gg T$ and apply a random rotation $U$, any deterministic algorithm making fewer than $T$ queries will fail on certain instances to find the directions of all the $T$ axes. Hence, from an algorithmic perspective, if we can construct a $T$-dimensional zero chain with all SOSPs or even FOSPs overlapped with all the $T$ axes, we can establish an $\Omega(T)$ lower bound for all deterministic classical algorithms.

Following this intuition, Ref. (Carmon et al., 2021) provided a concrete hard instance construction to obtain an $\Omega(1/\epsilon^2)$ lower bound for deterministic classical algorithms. In particular, Ref. (Carmon et al., 2021) first defined the following zero-chain $\bar{F}_{T;\mu}(\mathbf{x})\colon \mathbb{R}^{T+1} \to \mathbb{R}$:

$$\bar{F}_{T;\mu}(\mathbf{x}) = \frac{\sqrt{\mu}}{2}(x_1 - 1)^2 + \frac{1}{2}\sum_{i=1}^{T}(x_{i+1} - x_i)^2 + \mu\sum_{i=1}^{T}\Gamma(x_i), \tag{116}$$

where the non-convex function $\Gamma\colon \mathbb{R} \to \mathbb{R}$ is defined as

$$\Gamma(x) = 120\int_1^x \frac{t^2(t-1)}{1+t^2}\mathrm{d}t. \tag{117}$$

According to Lemma A.1 in Appendix A, finding an FOSP requires knowledge about the directions of all the $T$ axes. We further apply a unitary rotation $U \in \mathbb{R}^{(T+1)\times d}$ and certain appropriate scaling to obtain the formal hard instance

$$\tilde{F}_{T;U}(\mathbf{x}) := \lambda\sigma^2\bar{F}_T\big(\langle\mathbf{x}/\sigma, \mathbf{u}^{(1)}\rangle, \langle\mathbf{x}/\sigma, \mathbf{u}^{(2)}\rangle, \dots, \langle\mathbf{x}/\sigma, \mathbf{u}^{(T+1)}\rangle\big), \tag{118}$$

where $\mathbf{u}^{(i)}$ stands for the $i$-th column of the rotation matrix $U$, and all its columns $\{\mathbf{u}^{(1)}, \dots, \mathbf{u}^{(T+1)}\}$ forms a set of orthonormal vectors. We use $\tilde{\mathcal{F}}$ to denote the set of functions that can be presented in the form of (118) for some suitable parameters $T, U, \lambda$, and $\sigma$ whose function value at point $\mathbf{0}$ is not far from its minimum value, i.e.,

$$F(\mathbf{0}) - \inf_{\mathbf{x}} F(\mathbf{x}) \le \Delta, \quad \forall F \in \tilde{\mathcal{F}}. \tag{119}$$

Based on $\tilde{\mathcal{F}}$, we have the following classical lower bound result.

**Lemma 5.2** (Theorem 2, Ref. (Carmon et al., 2021)). *There exist numerical constants $c, C \in \mathbb{R}_+$ and $\ell_q \le e^{\frac{3q}{2}\log q + Cq}$ for every $q \in \mathbb{N}$ such that, for any deterministic classical algorithm making at most*

$$c \cdot \Delta\Big(\frac{L_1}{\ell_1}\Big)^{\frac{3}{7}}\Big(\frac{L_2}{\ell_2}\Big)^{\frac{2}{7}}\epsilon^{-12/7} \tag{120}$$

*gradient queries, there exists a function $\tilde{F} \in \tilde{\mathcal{F}}$ such that the output of this algorithm on $\tilde{F}$ is not an $\epsilon$-FOSP of $\tilde{F}$.*

This lower bound regarding deterministic classical algorithms is, however, hard to be extended to randomized classical algorithms straightforwardly. Intuitively, the concept of zero-chain in Definition 5.1 can be extended to higher-order derivatives, and the hard instance $\tilde{F}_{T;U}$ in (118) is no longer a zero chain for derivatives of second- or higher-orders. Hence, the algorithm may benefit from adding random perturbations and may not need to discover all the $T + 1$ components $\{\mathbf{u}^{(1)}, \dots, \mathbf{u}^{(T+1)}\}$ one by one. To the best of our knowledge, it remains unclear whether the same lower bound result holds for randomized classical algorithms.

In the remaining part of this section, we will demonstrate that the presence of noise can drastically increase the hardness of finding an $\epsilon$-FOSP in the worst case. Specifically, we first derive the lower bound result for general noise models parameterized by the concept of noise radius $r_0$. Then, we discuss the values of $r_0$ in different settings with noisy zeroth-order oracle (Assumption 1.1) or noisy first-order oracle (Assumption 1.2), respectively.

## 5.2 Noisy Quantum Lower Bound with Bounded Input Domain

In this subsection, we first introduce the quantum lower bound on functions with bounded input domains. The intuition is that the noise can create a non-informative region around $\mathbf{0}$, which is

a hyperball $\mathbb{B}(0, r_0)$ whose certain radius $r_0$ depends on the noise rate. Then, if the dimension of $\tilde{F}_{T;U}$ defined in Eq. (118) is large enough, any random perturbation with bounded norm will fall in $\mathbb{B}(0, r_0)$ with an overwhelming probability, which leads to the fact that the lower bound in Lemma 5.2 additionally holds for not only randomized classical algorithms but also quantum algorithms.

We adopt the quantum query model introduced in (Garg et al., 2021a). For a $d$-dimensional objective function $F$, assume we have access to its noisy evaluation $f \colon \mathbb{R}^d \to \mathbb{R}$ via the following quantum oracle $O_f$,

$$O_f \ket{\mathbf{x}} \ket{\mathbf{y}} \to \ket{\mathbf{x}} \ket{\mathbf{y} \oplus \big(f(\mathbf{x}), \nabla f(\mathbf{x})\big)}. \tag{121}$$

We remark that the oracle $O_f$ here is even stronger than the zeroth- or the first-order oracles in (7) and (8). Then, any quantum algorithm $A_{\text{quantum}}$ making $q$ queries to $O_f$ can be described by the following sequence of unitaries

$$V_q O_f V_{q-1} O_f \cdots V_1 O_f V_0 \tag{122}$$

applied to some initial state, say $\ket{0}$ without loss of generality. In the special case where the objective function $\tilde{F}_{T;U} \in \tilde{F}$, for the convenience of notation we denote $O_{f;T;U}$ to be the quantum oracle encoding its noisy evaluation $f_{r_0;T;U}$ in the form of (121). To obtain our quantum lower bound, we set the noisy function to be in the form

$$f_{r_0;T;U}(\mathbf{x}) := \lambda \sigma^2 \bar{F}_T\big(\langle \mathbf{x}/\sigma, \mathbf{u}^{(1)}\rangle, \ldots, \langle \mathbf{x}/\sigma, \mathbf{u}^{(\text{prog}_{\sigma r_0}(\mathbf{x})+1)}\rangle, 0, \ldots, 0\big), \tag{123}$$

where $\text{prog}_{\sigma r_0}(\mathbf{x})$ is defined as the largest index $j$ between 1 and $T$ satisfying $|\langle \mathbf{x}, \mathbf{u}^{(j)}\rangle| \geq \sigma r_0$. Moreover, we define the following indicator function

$$\delta_{r_0}(y) := \mathbb{1}\{|y| \geq r_0\} \cdot y. \tag{124}$$

Intuitively, in the noisy function $f_{r_0;T;U}$ we eliminate the influence of the $i$th component on the function when the overlap between $\mathbf{x}$ and $\mathbf{u}^{(i)}$ is smaller than certain threshold $r_0$. The detailed values of $r_0$ under different noise assumptions will be specified later. Hence, when the dimension $d$ is large enough, any random perturbation with a bounded norm will make no observable difference with an overwhelming probability. Moreover, we can note that $\epsilon$-FOSPs of $f_{r_0;T;U}(\mathbf{x})$ and $\tilde{F}_{T;U}(\mathbf{x})$ are the same. Hence, one needs to identify all the $T$ components $\{\mathbf{u}^{(1)}, \ldots, \mathbf{u}^{(T)}\}$ to find an $\epsilon$-FOSP, which we demonstrate later that can only be done sequentially even by a quantum algorithm.

For any possible quantum algorithm $A_{\text{quantum}}$ making $k < T$ queries in total, adopting a similar technique introduced in (Garg et al., 2021a;b), we define a sequence of unitaries starting with $A_0 = A_{\text{quantum}}$ as follows:

$$\begin{aligned}
A_0 &:= V_k O_{f;T;U} V_{k-1} O_{f;T;U} \cdots O_{f;T;U} V_1 O_{f;T;U} V_0 \tag{125}\\
A_1 &:= V_k O_{f;T;U} V_{k-1} O_{f;T;U} \cdots O_{f;T;U} V_1 O_{f;1;U_1} V_0 \\
A_2 &:= V_k O_{f;T;U} V_{k-1} O_{f;T;U} \cdots O_{f;2;U_2} V_1 O_{f;1;U_1} V_0 \\
&\ \ \vdots \\
A_k &:= V_k O_{f;k;U_k} V_{k-1} O_{f;k-1;U_{k-1}} \cdots O_{f;2;U_2} V_1 O_{f;1;U_1} V_0,
\end{aligned}$$

where $U_t \in \mathbb{R}^{d \times t}$ is defined as the orthogonal matrix with columns $\mathbf{u}^{(1)}, \ldots, \mathbf{u}^{(t)}$, and the function $f_{r_0;t;U_t}(\mathbf{x})$ encoded in $O_{f;t;U_t}$ is defined as

$$f_{r_0;t;U_t} := \lambda \sigma^2 \bar{F}_T\big(\langle \mathbf{x}/\sigma, \mathbf{u}^{(1)}\rangle, \ldots, \langle \mathbf{x}/\sigma, \mathbf{u}^{(\text{prog}_{\sigma r_0}^t(\mathbf{x})+1)}\rangle, 0, \ldots, 0\big), \tag{126}$$

where $\text{prog}_{\sigma r_0}^t(\mathbf{x})$ is defined as the largest index $j$ between 1 and $t-1$ satisfying $|\langle \mathbf{x}, \mathbf{u}^{(j)}\rangle| \geq \sigma r_0$. Our goal is to demonstrate that $A_0$ will fail to find an $\epsilon$-FOSP with high probability. To do so, we employ a hybrid argument showing that the outputs of $A_i$ and $A_{i+1}$ defined in the sequence (125) are close for every $i < k$, so does the outputs of $A_0$ and $A_k$, which cannot solve the problem with high probability since it contains no information of the $T$-th component, which is necessary for finding an $\epsilon$-FOSP with high success probability.

**Lemma 5.3** ($A_t$ and $A_{t-1}$ have similar outputs). *Consider the hard instance $\tilde{F}_{T;U}(\mathbf{x}) \colon \mathbb{R}^d \to \mathbb{R}$ defined in (116) with domain $\mathbb{B}(0, 2\sigma\sqrt{T})$ and its noisy evaluation $f_{r_0;T;U}$ defined in (123) with $d \geq 4T$, let $A_t$ for $t \in [k-1]$ be the unitaries defined in Eq. (125). Then*

$$\mathbb{E}_U\big(\|A_t \ket{\mathbf{0}} - A_{t-1} \ket{\mathbf{0}}\|^2\big) \leq 8T e^{-dr_0^2/(4T)}. \tag{127}$$

*Proof.* From the definition of the unitaries in Eq. (125) and the unitary invariance of the spectral norm, we have

$$\|A_t |\mathbf{0}\rangle - A_{t-1} |\mathbf{0}\rangle\| = \left\|(O_{f;t;U_t} - O_{f;T;U})V_{t-1}O_{f;t-1;U_{t-1}}\cdots O_{f;1;U_1}V_0 |\mathbf{0}\rangle\right\|. \quad (128)$$

We will prove the claim for any fixed choice of vectors $\{\mathbf{u}^{(1)},\ldots,\mathbf{u}^{(t-1)}\}$, which will imply the claim for any distribution over those vectors. Let us prove the claim for any fixed choice of vectors $\{\mathbf{u}^{(1)},\ldots,\mathbf{u}^{(t-1)}\}$, which will imply the claim for any distribution over those vectors. Once we have fixed these vectors, the state $V_{t-1}O_{f;t-1;U_{t-1}}\cdots O_{f;1;U_1}V_0 |\mathbf{0}\rangle$ is a fixed state, which can be referred to as $|\psi\rangle$. Thus our problem reduces to showing for all quantum states $|\psi\rangle$,

$$\mathbb{E}_{\{\mathbf{u}^{(t)},\ldots,\mathbf{u}^{(T)}\}}\left(\|(O_{f;t;U_t} - O_{f;T;U}) |\psi\rangle\|^2\right) \leq 8Te^{-dr_0^2/(4T)}. \quad (129)$$

We write the state $|\psi\rangle$ as $|\psi\rangle = \sum_{\mathbf{x}} \alpha_{\mathbf{x}} |\mathbf{x}\rangle |\phi_{\mathbf{x}}\rangle$, where $\mathbf{x}$ is the query made to the oracle, and $\sum_{\mathbf{x}} |\alpha_{\mathbf{x}}|^2 = 1$. Hence, the left-hand side of Eq. (129) equals

$$\mathbb{E}_{\{\mathbf{u}^{(t)},\ldots,\mathbf{u}^{(T)}\}}\left(\left\|\sum_{\mathbf{x}} \alpha_{\mathbf{x}}(O_{f;t;U_t} - O_{f;T;U}) |\mathbf{x}\rangle |\phi_{\mathbf{x}}\rangle\right\|^2\right) \quad (130)$$

$$\leq \sum_{\mathbf{x}} |\alpha_{\mathbf{x}}|^2 \cdot \mathbb{E}_{\{\mathbf{u}^{(t)},\ldots,\mathbf{u}^{(T)}\}}\left(\|(O_{f;t;U_t} - O_{f;T;U}) |\mathbf{x}\rangle\|^2\right). \quad (131)$$

Since $|\alpha_{\mathbf{x}}|^2$ defines a probability distribution over $\mathbf{x}$, we can again upper bound the right-hand side for any $\mathbf{x}$ instead. Note that $O_{f;t;U_t}$ and $O_{f;T;U}$ behave identically for some inputs x, the only nonzero terms are those where the oracles respond differently, which can only happen if

$$\left(f_{r_0;t;U_t}(\mathbf{x}), \nabla f_{r_0;t;U_t}(\mathbf{x})\right) \neq \left(f_{r_0;T;U}(\mathbf{x}), \nabla f_{r_0;T;U}(\mathbf{x})\right). \quad (132)$$

When the response is different, we can upper bound $\|(O_{f;t;U_t} - O_{f;T;U}) |\mathbf{x}\rangle\|^2$ by 4 using the triangle inequality. Thus for any $\mathbf{x} \in \mathbb{B}(\mathbf{0}, 2\sigma\sqrt{T})$, we have

$$\mathbb{E}_{\{\mathbf{u}^{(t)},\ldots,\mathbf{u}^{(T)}\}}\left[\|(O_{f;t;U_t} - O_{f;T;U}) |\mathbf{x}\rangle\|^2\right] \quad (133)$$

$$\leq 4 \Pr_{\{\mathbf{u}^{(t)},\ldots,\mathbf{u}^{(T)}\}}\left[\left(f_{r_0;t;U_t}(\mathbf{x}), \nabla f_{r_0;t;U_t}(\mathbf{x})\right) \neq \left(f_{r_0;T;U}(\mathbf{x}), \nabla f_{r_0;T;U}(\mathbf{x})\right)\right]. \quad (134)$$

We use $\mathbf{x}_\perp$ to denote the projection of x to the span $\{\mathbf{u}^{(t)},\ldots,\mathbf{u}^{(T)}\}$. Intuitively, as long as each component of $\mathbf{x}_\perp$ has absolute value smaller than $\sigma r_0$, the components $\{\mathbf{u}^{(t)},\ldots,\mathbf{u}^{(T)}\}$ will have no observable impact. Quantitatively,

$$\Pr\left[\left(f_{r_0;t;U_t}(\mathbf{x}), \nabla f_{r_0;t;U_t}(\mathbf{x})\right) \neq \left(f_{r_0;T;U}(\mathbf{x}), \nabla f_{r_0;T;U}(\mathbf{x})\right)\right] \quad (135)$$

$$\leq 1 - \Pr\left[|\langle\mathbf{u}^{(t)},\mathbf{x}\rangle|,\ldots,|\langle\mathbf{u}^{(T)},\mathbf{x}\rangle| \leq \delta r_0\right]. \quad (136)$$

Since $\{\mathbf{u}^{(t)},\ldots,\mathbf{u}^{(T)}\}$ are chosen uniformly at random in the $(d-t+1)$-dimensional orthogonal complement of span $\{\mathbf{u}^{(1)},\ldots,\mathbf{u}^{(t-1)}\}$, for any $t \leq i \leq T$, by Lemma A.4 we can further derive that

$$\Pr\left[|\langle\mathbf{u}^{(i)},\mathbf{x}\rangle| > r_0\right] \leq 2e^{-dr_0^2/(4T)}, \quad (137)$$

which leads to

$$\Pr\left[|\langle\mathbf{u}^{(t)},\mathbf{x}\rangle|,\ldots,|\langle\mathbf{u}^{(T)},\mathbf{x}\rangle| \leq r_0\right] \geq \left(1 - 2e^{-dr_0^2/(4T)}\right)^T \geq 1 - 2Te^{-dr_0^2/(4T)}, \quad (138)$$

indicating

$$\mathbb{E}_{\{\mathbf{u}^{(t)},\ldots,\mathbf{u}^{(T)}\}}\left[\|(O_{f;t;U_t} - O_{f;T;U}) |\mathbf{x}\rangle\|^2\right] \leq 8Te^{-dr_0^2/(4T)}, \quad (139)$$

and

$$\mathbb{E}_U\left(\|A_t |\mathbf{0}\rangle - A_{t-1} |\mathbf{0}\rangle\|^2\right) \leq 8Te^{-dr_0^2/(4T)}. \quad (140)$$

$\square$

**Proposition 5.4.** *Consider the $d$-dimensional function $\tilde{f}_{T;U}(\mathbf{x}): \mathbb{B}(\mathbf{0}, 2\sigma\sqrt{T}) \to \mathbb{R}$ defined in (118) with the rotation matrix $U$ being chosen arbitrarily. Consider any quantum algorithm $A_{\text{quantum}}$ containing $t < T$ queries to the noisy oracle $O_f$ defined in Eq. (121), let $p_U$ be the probability distribution over $\mathbf{x} \in \mathbb{B}(\mathbf{0}, 2\sigma\sqrt{T})$ obtained by measuring the state $A_{\text{quantum}} |0\rangle$, which is related to the rotation matrix $U$. Then,*

$$\Pr_{U,\mathbf{x}_{out}\sim p_U}\left[\|\nabla\tilde{F}_{T;U}(\mathbf{x}_{out})\| \leq \lambda\sigma\mu^{3/4}/8\right] \leq 16Te^{-dr_0^2/(8T)}. \quad (141)$$

*Proof.* Consider the sequence of unitaries $\{A_0, \ldots, A_t\}$ associated with $A_{\text{quantum}}$ defined in (125), we first demonstrate that $A_t$ cannot find a point with small gradient with high probability. In particular, let $p_U^{(t)}$ be the probability distribution over $\mathbf{x} \in \mathbb{B}(\mathbf{0}, 2\sigma\sqrt{T})$ obtained by measuring the output state $A_t \ket{0}$. Then we have

$$\Pr_{U_t, \mathbf{x}_{\text{out}} \sim p_{U_t}^{(t)}} \left[ \|\nabla \tilde{F}_{T;U}(\mathbf{x}_{\text{out}})\| \leq \lambda \sigma \mu^{3/4}/8 \right] \tag{142}$$

$$\leq \max_{\mathbf{x} \in \mathbb{B}(\mathbf{0}, 2\sigma\sqrt{T})} \Pr_{\{\mathbf{u}^{(t+1)}, \ldots, \mathbf{u}^{(T+1)}\}} \left[ \|\nabla \tilde{F}_{T;U}(\mathbf{x})\| \leq \lambda \sigma \mu^{3/4}/8 \right], \tag{143}$$

whereby Lemma B.1 we have

$$\Pr_{\{\mathbf{u}^{(t+1)}, \ldots, \mathbf{u}^{(T+1)}\}} \left[ \|\nabla \tilde{F}_{T;U}(\mathbf{x})\| \leq \lambda \sigma \mu^{3/4}/8 \right] \leq 8T e^{-dr_0^2/(8T)} \tag{144}$$

for any $\mathbf{x} \in \mathbb{B}(\mathbf{0}, 2\sigma\sqrt{T})$, which leads to

$$\Pr_{U_t, \mathbf{x}_{\text{out}} \sim p_{U_t}^{(t)}} \left[ \|\nabla \tilde{F}_{T;U}(\mathbf{x}_{\text{out}})\| \leq \lambda \sigma \mu^{3/4}/8 \right] \leq 8T^2 e^{-dr_0^2/(4T)}. \tag{145}$$

Moreover, by Lemma 5.3 and Cauchy-Schwartz inequality, we have

$$\mathbb{E}_U \left[ \|A_t \ket{0} - A_0 \ket{0}\|^2 \right] \leq t \cdot \mathbb{E}_U \left[ \sum_{k=1}^{t-1} \|A_{k+1} \ket{0} - A_k \ket{0}\|^2 \right] \leq 8T^2 e^{-dr_0^2/(4T)}. \tag{146}$$

Then by Markov's inequality,

$$\Pr_U \left[ \|A_{t-1} \ket{0} - A_0 \ket{0}\|^2 \geq 4T e^{-dr_0^2/(8T)} \right] \leq 4T e^{-dr_0^2/(8T)}, \tag{147}$$

since both norms are at most 1. Hence, we can deduce that the total variance distance between $p_U$ n and $p_U^{(t)}$ can be bounded by

$$4T e^{-dr_0^2/(8T)} + 4T e^{-dr_0^2/(8T)} \leq 8T e^{-dr_0^2/(8T)}, \tag{148}$$

which further leads to

$$\Pr_{U, \mathbf{x}_{\text{out}} \sim p_U} \left[ \|\nabla \tilde{F}_{T;U}(\mathbf{x}_{\text{out}})\| \leq \lambda \sigma \mu^{3/4}/8 \right] \tag{149}$$

$$\leq \Pr_{U_t, \mathbf{x}_{\text{out}} \sim p_{U_t}^{(t)}} \left[ \|\nabla \tilde{F}_{T;U}(\mathbf{x}_{\text{out}})\| \leq \lambda \sigma \mu^{3/4}/8 \right] + 8T e^{-dr_0^2/(8T)} \leq 16T e^{-dr_0^2/(8T)}. \tag{150}$$

$$\square$$

**Proposition 5.5.** *Let $r_0, \Delta, L_1, L_2, \epsilon$ be positive and $\epsilon \leq L_1^2/L_2$. Then there exist positive numerical constants $c, C \in \mathbb{R}$ and $\ell_q \leq e^{\frac{3q}{2} \log q + Cq}$ for every $q \in \mathbb{N}$, and a set $\Omega$ consisting of function pairs $(\tilde{F}_{T;U}, f_{r_0;T;U})$ with $\tilde{F}_{T;U}$ and $f_{r_0;T;U}$ defined in (118) and (123) respectively upon the input domain $\mathbb{B}(\mathbf{0}, \mathcal{R})$ with*

$$T = \frac{\Delta}{20} \left( \frac{L_2}{\ell_2} \right)^{\frac{5}{4}} \left( \frac{L_1}{2\ell_1} \right)^{\frac{3}{7}} (8\epsilon)^{-12/7}, \qquad \mathcal{R} = 2\sqrt{T} \cdot \left( \frac{L_2}{\ell_2} \right)^{-3/4} \left( \frac{L_1}{\ell_1} \right)^{-1/7} (8\epsilon)^{4/7}, \tag{151}$$

*such that, for any quantum algorithm $A_{\text{quantum}}$ making fewer than $T$ queries to the oracle $O_f$ in the form of (121) encoding the function values and gradients of $f_{r_0;T;U}$ in (123), there exists an orthogonal matrix $U$ such that, $A_{\text{quantum}}$ cannot find an $\epsilon$-FOSP of the corresponding $\tilde{F}_{T;U}$ with probability larger than, where $\tilde{F}_{T;U}$ is $L_1$-smooth and $L_2$-Hessian Lipschitz and satisfies*

$$\tilde{F}_{T;U}(\mathbf{0}) - \inf_{\mathbf{x}} \tilde{F}_{T;U}(\mathbf{x}) \leq \Delta. \tag{152}$$

*Proof.* We set the scaling parameters $\lambda, \sigma$ in $\tilde{F}_{T;U}$ and $f_{r_0;T;U}$ to be

$$\lambda = \frac{L_1}{2\ell_1}, \qquad \mu = \frac{L_2 \sigma}{\lambda \ell_2}, \qquad \sigma = \left( \frac{L_2}{\ell_2} \right)^{-3/4} \lambda^{-1/7} (8\epsilon)^{4/7}, \tag{153}$$

which satisfy $\mu \leq 1$ since $\epsilon \leq L_1^2/L_2$. By Proposition 5.4, for any possible quantum algorithm $A_{\text{quantum}}$ making $t < T$ queries to the oracle $O_f$ defined in (121) encoding $(\tilde{f}_{r_0;U;T}, \nabla \tilde{f}_{r_0;U;T})$, we have

$$\Pr_{U,\mathbf{x}_{\text{out}} \sim p_U} \left[ \|\nabla \tilde{f}_{T;U}(\mathbf{x}_{\text{out}})\| \leq \lambda \sigma \mu^{3/4}/8 \right] = \Pr_{U,\mathbf{x}_{\text{out}} \sim p_U} \left[ \|\nabla \tilde{f}_{T;U}(\mathbf{x}_{\text{out}})\| \leq \epsilon \right] \leq 16 T e^{-dr_0^2/(8T)}, \tag{154}$$

where $p_U$ is the probability distribution over $\mathbf{x} \in \mathbb{B}(\mathbf{0}, 2\sigma\sqrt{T})$ obtained by measuring the state $A_{\text{quantum}} |0\rangle$, indicating that the success probability of $A_{\text{quantum}}$ finding an $\epsilon$-FOSP of $\tilde{F}_{T;U}$ is at most $16 T e^{-dr_0^2/(8T)}$. Moreover, by Lemma A.2 we can derive that, for any $T \times d$ orthogonal matrix $U$, the function $\tilde{F}_{T;U}$ is $(1+\mu)\ell_1 = L_1$-smooth and $\lambda\mu\ell_2/\sigma = L_2$-Hessian Lipschitz, with

$$\tilde{F}_{T;U}(\mathbf{0}) - \inf_{\mathbf{x}} \tilde{F}_{T;U}(\mathbf{x}) \leq \lambda\sigma^2 \left( \frac{\sqrt{\mu}}{2} + 10\mu T \right) \leq \Delta. \tag{155}$$

$\square$

Proposition 5.5 shows that, if we restrict the input domain of the function pair $(F, f)$ to a hyperball with radius $\mathcal{R}$, in the worst case every quantum algorithm has to make at least $\Omega(\epsilon^{-12/7})$ queries to the noisy evaluation $f$ to find an $\epsilon$-FOSP of $F$ with high probability. Moreover, the dimension $d$ of the hard instance achieving this lower bound is of order $\Omega(\epsilon^{-12/7} \log(1/\epsilon)/r_0^2)$, where the noise radius $r_0$ is determined by the noise rate with different relations under different noise assumptions, on which a detailed discussion is given in Section 5.4 after we extend this lower bound to unbounded input domain in Section 5.3.

## 5.3 NOISY QUANTUM LOWER BOUND WITH UNBOUNDED INPUT DOMAIN

In this subsection, we extend the quantum lower bound proved in Proposition 5.5 to functions with an unbounded input domains. In particular, Ref. (Carmon et al., 2020) introduced a method for extending lower bound to unbounded input domain by adding a scaling term on the input vector and additionally introducing a quadratic term. The intuition is that, if the input vector has a large norm, the corresponding function value is almost solely determined by the quadratic term and it cannot be an approximate stationary point. Hence, it is not beneficial for any classical algorithm to explore any point outside a certain bounded region, indicating that the lower bound with an unbounded input domain is the same as the one with a bounded input domain. The same argument also holds for quantum algorithms, as shown in Ref. (Zhang & Li, 2023a).

Quantitatively, we consider the following $T + 1$ dimensional kernel function defined on $\mathbb{R}^{T+1}$,

$$\bar{\mathfrak{F}}_{T;\mu}(\mathbf{x}) := \bar{F}_{T;\mu}(\gamma_\alpha(\mathbf{x})) + \beta \|\sin \mathbf{x}\|^2, \tag{156}$$

where $\gamma_\alpha(\mathbf{x})$ is defined as

$$\gamma_\alpha(\mathbf{x}) := \begin{cases} \left(1 - \frac{\|\mathbf{x}\|}{\alpha\sqrt{T}}\right)^3 \cdot \mathbf{x}, & \|\mathbf{x}\| \leq \alpha\sqrt{T}, \\ \mathbf{0}, & \|\mathbf{x}\| > \alpha\sqrt{T}. \end{cases} \tag{157}$$

By Lemma A.3, finding an $\epsilon$-SOSP or even an $\epsilon$-FOSP of $\bar{\mathfrak{F}}_{T;\mu}$ requires knowledge of all the $T + 1$ coordinate directions, if it is projected to a $d$-dimensional space via an arbitrary orthogonal matrix $U \in \mathbb{R}^{d \times (T+1)}$. Moreover, to guarantee that the hard instance satisfies the $B$-boundedness condition required in the empirical risk setting considered in this paper, we additionally add a sine function to the quadratic term and obtain the following hard instance defined on the hypercube $[-\pi\mathcal{L}/2, \pi\mathcal{L}/2]^d$ with $\mathcal{L} = \zeta\alpha\sqrt{T}$ for some constant $\zeta \geq 2$,

$$\hat{\mathfrak{F}}_{T;U}(\mathbf{x}) := \bar{F}_{T;\mu}(\gamma_\alpha(U^T\mathbf{x})) + \beta\mathcal{L}^2 \|\sin(\mathbf{x}/\mathcal{L})\|^2, \tag{158}$$

where the constants $\alpha, \beta$ are chosen according to Lemma A.3, and for any $\mathbf{y} \in \mathbb{R}^d$, $\sin \mathbf{y}$ is defined as

$$\sin \mathbf{y} := (\sin y_1, \ldots, \sin y_d)^T. \tag{159}$$

**Lemma 5.6.** *Consider the function $\hat{\mathfrak{F}}_{T;U} \colon [-\pi\mathcal{L}/2, \pi\mathcal{L}/2]^d \to \mathbb{R}$ defined in Eq. (158), suppose that the parameter $\mu$ satisfies $\mu \leq 1$. Then, there exist positive constants $\alpha, \beta, \zeta$ such that*

1. *For any $\mathbf{x} \in [-2\zeta\alpha\sqrt{T}, 2\zeta\alpha\sqrt{T}]^d$ such that*

$$|\langle \mathbf{x}, \mathbf{u}^T \rangle|, |\langle \mathbf{x}, \mathbf{u}^{T+1} \rangle| \leq \frac{0.05}{T + 1/\sqrt{\mu}}, \tag{160}$$

   *its gradient satisfies*

$$\|\nabla \bar{\hat{\mathfrak{F}}}_{T;U}(\mathbf{x})\| \geq \mu^{3/4}/32; \tag{161}$$

2. $\bar{\hat{\mathfrak{F}}}_{T;U}(\mathbf{0}) - \inf_{\mathbf{x}} \bar{\hat{\mathfrak{F}}}_{T;U}(\mathbf{x}) \leq \frac{\sqrt{\mu}}{2} + 10\mu T$;

3. *For $p = 1, 2$, the $p$-th order derivatives of $\bar{\hat{\mathfrak{F}}}_{T;U}$ are $(2\mathbb{I}\{p = 1\} + \mu)\ell_p$-Lipschitz continuous in the hyperball $\mathbb{B}(\mathbf{0}, \alpha\sqrt{T})$, where $\ell_p \leq \exp\left(\frac{3p}{2}\log p + cp\right)$ for a numerical constant $c < \infty$.*

*Proof.* We set the constants $\alpha, \beta$ according to Lemma A.3. Note that for any vector $\mathbf{y}$ with $\|\mathbf{y}\| \leq 1/(2\zeta)$, the values as well as first- and second-order derivatives of $\|\mathbf{y}\|^2$ and $\|\sin\mathbf{y}\|^2$ are close to each other given that $\zeta$ reaches a large enough value that is independent from $d$. Quantitatively, we have

$$\|\mathbf{y}\|^2 - \|\sin\mathbf{y}\|^2 \leq \sum_{i=1}^{d} y_i^2 - \left(y_i - y_i^3/6\right)^2 \leq \sum_{i=1}^{d} \frac{y_i^4}{3} \leq \frac{1}{48\zeta^4}, \tag{162}$$

and

$$\left\|\nabla \cdot (\|\mathbf{y}\|^2 - \|\sin\mathbf{y}\|^2)\right\| = \|2\mathbf{y} - \sin(2\mathbf{y})\| \leq \frac{1}{6} \cdot \frac{1}{\zeta^3} = \frac{1}{6\zeta^3}. \tag{163}$$

Moreover, we notice that

$$\nabla^2(\|\mathbf{y}\|^2 - \|\sin\mathbf{y}\|^2) = I - \begin{bmatrix} \cos 2y_1, & \cdots, & 0 \\ \vdots, & \ddots, & \vdots \\ 0, & \cdots, & \cos 2y_d \end{bmatrix}, \tag{164}$$

which leads to

$$\left\|\nabla^2(\|\mathbf{y}\|^2 - \|\sin\mathbf{y}\|^2)\right\| \leq \frac{1}{2} \cdot (2/\zeta)^2 = \frac{2}{\zeta^2}. \tag{165}$$

Hence, there exists a large enough $\zeta = O(1/\mu)$ independent from $d$ such that, $\hat{\mathfrak{F}}_{T;U}$ is close enough to the pure rotation of $\bar{\mathfrak{F}}_{T;\mu}$ in the hyperball $\mathbb{B}(\mathbf{0}, \alpha\sqrt{T})$ up to the second-order derivatives, and the above three conditions can be satisfied. $\square$

Note that if we replicate the hypercube $[-\pi\mathcal{L}/2, \pi\mathcal{L}/2]^d$ in $\mathbb{R}^d$ consecutively and have the function value in each hypercube being $\hat{\mathfrak{F}}_{T;U}$ respectively, the new function defined on $\mathbb{R}^d$ is still infinitely differentiable. Moreover, we can notice that finding an $\epsilon$-SOSP in $\mathbb{R}^d$ is equivalent to finding an $\epsilon$-SOSP in one specific hypercube $[-\pi\mathcal{L}/2, \pi\mathcal{L}/2]^d$, since for any $\mathbf{x}$ on the boundary of $[-\pi\mathcal{L}/2, \pi\mathcal{L}/2]^d$, the Hessian matrix

$$\nabla^2 \hat{\mathfrak{F}}_{T;U}(\mathbf{x}) = 2\beta \cdot \begin{bmatrix} \cos(2x_1/\mathcal{L}) & \cdots & 0 \\ \vdots & \ddots & \vdots \\ 0 & \cdots & \cos(2x_d/\mathcal{L}) \end{bmatrix}, \tag{166}$$

is positive definite with the matrix norm being $2\beta$, indicating that $\mathbf{x}$ cannot be an $\epsilon$-SOSP. Similar to Section 5.2, we add scaling parameters $\lambda$ and $\sigma$ to $\hat{\mathfrak{F}}_{T;U}$ and obtain the formal hard function

$$\tilde{\hat{\mathfrak{F}}}_{T;U}(\mathbf{x}) := \lambda\sigma^2 \hat{\mathfrak{F}}_{T;U}(\mathbf{x}/\sigma). \tag{167}$$

Moreover, we assume access to the following noisy evaluation $\mathfrak{f}_{r_0;T;U}$ of $\hat{\mathfrak{F}}_{T;U}$,

$$\mathfrak{f}_{r_0;T;U} := \lambda\sigma^2 \big[\bar{F}_{T;\mu}\big(\gamma_\alpha(\langle \mathbf{x}/\sigma, \mathbf{u}^{(1)} \rangle), \ldots, \gamma_\alpha(\langle \mathbf{x}/\sigma, \mathbf{u}^{(\mathrm{prog}_{r_0}(\gamma_\alpha(\mathbf{x}/\mu))+1)} \rangle), 0, \ldots, 0\big) \tag{168}$$
$$+ \beta\mathcal{L}\|\sin(\mathbf{x}/(\sigma\mathcal{L}^2))\|^2\big], \tag{169}$$

which is encoded in the quantum oracle $O_{\mathfrak{f}}$ with form (121). Then, we can prove the following quantum lower bound via the function pair $(\tilde{\hat{\mathfrak{F}}}_{T;U}, \mathfrak{f}_{r_0;T;U})$.

**Theorem 5.7.** *Let $\Delta, \epsilon, r_0$ be positive and $\epsilon \leq 1/2$, where the noise radius $r_0$ is a parameter related to the noise rate. Then there exist positive numerical constants $c, C, \alpha, \beta \in \mathbb{R}$ and $\ell_q \leq e^{\frac{3q}{2} \log q + Cq}$ for every $q \in \mathbb{N}$, and a set $\Omega$ consisting of function pairs $(\tilde{\mathfrak{F}}_{T;U}, \mathfrak{f}_{r_0;T;U})$ with $\tilde{\mathfrak{F}}_{T;U}$ and $\mathfrak{f}_{r_0;T;U}$ defined in (167) and (168) respectively with*

$$T = \frac{\Delta}{20}\left(\frac{1}{\ell_2}\right)^{\frac{5}{4}}\left(\frac{1}{2\ell_1}\right)^{\frac{3}{7}}(16\epsilon)^{-12/7}, \tag{170}$$

*such that, for any quantum algorithm $A_{\text{quantum}}$ making fewer than $T$ queries to the oracle $O_{\mathfrak{f}}$ in the form of (121) encoding the function value and gradient of $\mathfrak{f}_{r_0;T;U}$ in (168), there exists an orthogonal matrix $U$ such that, $A_{\text{quantum}}$ cannot find an $\epsilon$-SOSP of the corresponding $\tilde{\mathfrak{F}}_{T;U}$ with probability larger than $e^{-dr_0^2/(2\alpha^2 T)}$, where $\tilde{\mathfrak{F}}_{T;U}$ is $B$ bounded, $L_1$-smooth, and $L_2$-Hessian Lipschitz with*

$$B = O\left(\Delta + \epsilon^{-12/7}\right), \quad L_1 = O(1), \quad L_2 = O(1), \tag{171}$$

*and satisfies*

$$\tilde{\mathfrak{F}}_{T;U}(\mathbf{0}) - \inf_{\mathbf{x}} \tilde{\mathfrak{F}}_{T;U}(\mathbf{x}) \leq \Delta. \tag{172}$$

*Proof.* Since the functions and noisy evaluations in each hypercube are the same, without loss of generality we assume all queries happen in the hypercube $[-\pi\sigma\mathcal{L}/2, \pi\sigma\mathcal{L}/2]^d$ centered at $\mathbf{0}$. Similar to the setting of Proposition 5.5, we set the scaling parameters $\lambda, \mu$ and $\mu$ in $\tilde{\mathfrak{F}}_{T;U}$ and $\mathfrak{f}_{r_0;T;U}$ to be

$$\lambda = \frac{1}{2\ell_1}, \quad \mu = \frac{\sigma}{\lambda\ell_2}, \quad \sigma = \ell_2^{3/4}\lambda^{-2/7}(16\epsilon)^{4/7}, \tag{173}$$

which satisfies $\mu \leq 1$ since $\epsilon \leq 1$. By Lemma 5.6, finding an $\lambda\sigma\mu^{3/4}/32 = \epsilon$-SOSP of $\tilde{\mathfrak{F}}$ with high probability requires complete knowledge of all the $T + 1$ columns of the matrix $U$. Equivalently, we can find an $2\epsilon$-SOSP of the function $\tilde{F}_{T;U}$ by finding an $\frac{\epsilon}{2}$-SOSP of $\tilde{\mathfrak{F}}_{T;U}$ with the same $U$ and same settings of parameters, which by Proposition 5.5 requires at least $T$ queries to the quantum oracle $O_f$ encoding the noisy evaluation $f_{r_0;T;U}$ of $\tilde{F}_{T;U}$ to guarantee a success probability at least $e^{-dr_0^2/(2\alpha^2 T)}$.

In addition, we notice that one query to the quantum oracle $O_f$ can be implemented via one query to the quantum oracle $O_{\mathfrak{f}}$ encoding the noisy evaluation $\mathfrak{f}_{r_0;U;T}$. Hence, by Proposition 5.5 we can claim that to find an $\epsilon$-SOSP of $\tilde{\mathfrak{F}}_{T;U}$ with success probability at least $e^{-dr_0^2/(2\alpha^2 T)}$, it takes at least

$$T = \frac{\Delta}{20}\left(\frac{1}{\ell_2}\right)^{\frac{5}{4}}\left(\frac{1}{2\ell_1}\right)^{\frac{3}{7}}(16\epsilon)^{-12/7} \tag{174}$$

queries to the oracle $O_{\mathfrak{f}}$.

Moreover, by the second entry of Lemma 5.6, we know that

$$\tilde{\mathfrak{F}}_{T;U}(\mathbf{0}) - \inf_{\mathbf{x}} \tilde{\mathfrak{F}}_{T;U}(\mathbf{x}) \leq \lambda\sigma^2\left(\frac{\mu}{2} + 10\mu T\right) \leq \Delta. \tag{175}$$

Further, we can observe that

$$\sup_{\mathbf{x}} \tilde{\mathfrak{F}}_{T;U}(\mathbf{x}) - \tilde{\mathfrak{F}}_{T;U}(\mathbf{0}) \leq \lambda\sigma^2 \sup_{\|\mathbf{x}\| \leq \alpha\sqrt{T}} \bar{F}_{T;\mu}(\mathbf{x}) + \lambda\sigma^2\beta\mathcal{L}^2 \sup_{\mathbf{x}} \|\sin(\mathbf{x}/(\sigma\mathcal{L}))\|^2 \tag{176}$$

$$\leq \lambda\sigma^2\left(2\alpha^2 T + 60\alpha^2 T + \beta\zeta^2\alpha^2 T\right) \tag{177}$$

$$= O(\lambda\sigma^2\zeta^2 T) = O(\lambda\sigma^2\mu^{-2}T) \tag{178}$$

$$= O\left(\epsilon^{-12/7}\right), \tag{179}$$

indicating that $\tilde{\mathfrak{F}}_{T;U}(\mathbf{x})$ is $B$-bounded for $B = O(\Delta + \epsilon^{-12/7})$.

By the third entry of Lemma 5.6, $\tilde{F}_{T;U}$ is $\lambda(2 + \mu)\ell_1 = O(1)$-smooth and $\mu\ell_2/\sigma = O(1)$-Hessian Lipschitz in the region $\mathbb{B}(\mathbf{0}, \alpha\sigma\sqrt{T})$. For any point $\mathbf{x} \in [-\pi\mathcal{L}\sigma/2, \pi\mathcal{L}\sigma/2]^d - \mathbb{B}(\mathbf{0}, \alpha\sigma\sqrt{T})$, we have

$$\|\nabla^2\tilde{\mathfrak{F}}_{T;U}(\mathbf{x})\| = \beta\mathcal{L}^2\sigma^2\left\|\nabla^2\sin^2\left(\frac{\mathbf{x}}{\sigma\mathcal{L}}\right)\right\| \leq 4\beta = O(1), \tag{180}$$

and

$$\|\nabla^3 \tilde{F}_{T;U}(\mathbf{x})\| = \beta \mathcal{L}^2 \sigma^2 \left\| \nabla^3 \sin^2 \left( \frac{\mathbf{x}}{\sigma \mathcal{L}} \right) \right\| \leq \frac{8\beta}{\sigma \mathcal{L}} = O(1). \tag{181}$$

Hence, we can conclude that $\tilde{\mathfrak{F}}_{T;U}$ is $O(1)$-smooth and $O(1)$-Hessian Lipschitz in the entire space $\mathbb{R}^d$. $\qquad \square$

## 5.4 Lower Bound for Quantum Algorithms with Noisy Zeroth- and First-order Oracles

In this subsections, we specify the value of noise radius $r_0$ appearing in Theorem 5.7 when we are given noisy zeroth-order oracle or noisy first-order oracle satisfying Assumption 1.1 or Assumption 1.2, respectively, and further discuss the requirement on dimension $d$ to obtain our lower bound in $\epsilon$.

We first discuss the setting with zeroth-order oracle access.

**Corollary 5.8** (Formal version of Theorem 1.11, Part 1). *Let $\Delta, \epsilon > 0$ and $\epsilon \leq 1/2$. Then there exist positive numerical constants $c, C, \alpha, \beta \in \mathbb{R}$ and $\ell_q \leq e^{\frac{3q}{2} \log q + Cq}$ for every $q \in \mathbb{N}$, and a set $\Omega$ consisting of function pairs $(F, f)$ satisfying Assumption 1.1 with some $\nu$ satisfying*

$$\nu = \Omega\left(\epsilon^{-16/7}/d\right), \tag{182}$$

*such that, for any quantum algorithm $A_{\mathrm{quantum}}$ making fewer than $\Theta(\epsilon^{-12/7})$ queries to the oracle $O_f$ defined in (121) encoding the function value and gradient of $f$, there exists a function pair $(F, f)$ such that $A_{\mathrm{quantum}}$ cannot find an $\epsilon$-SOSP of $F$ with probability larger than $1/3$, where $F$ is $B$ bounded, $L_1$-smooth, and $L_2$-Hessian Lipschitz with*

$$B = O\left(\Delta + \epsilon^{-12/7}\right), \quad L_1 = O(1), \quad L_2 = O(1), \tag{183}$$

*and satisfies*

$$F(\mathbf{0}) - \inf_{\mathbf{x}} F_{T;U}(\mathbf{x}) \leq \Delta. \tag{184}$$

*Proof.* We adopt the settings of functions and parameters in Theorem 5.7 and set $\Omega$ to be

$$\Omega = \{(\tilde{\mathfrak{F}}_{T;U}, \mathfrak{f}_{r_0;T;U}) \mid U \in \mathbb{R}^{d \times (T+1)} \text{ s.t. } U^\top U = I\}, \tag{185}$$

where

$$T = \frac{\Delta}{20} \left( \frac{1}{2\ell_1} \right)^{\frac{3}{7}} \left( \frac{1}{\ell_2} \right)^{\frac{5}{4}} (16\epsilon)^{-12/7}. \tag{186}$$

By Lemma B.2, the parameter $r_0$ satisfies

$$\lambda \sigma^2 (50 r_0^2 T + 2\alpha r_0 \sqrt{T}) \leq \nu. \tag{187}$$

Moreover, by Theorem 5.7, if the dimension $d$ satisfies

$$d \geq \frac{4\alpha^2 T}{r_0^2}, \tag{188}$$

then for any quantum algorithm $A_{\mathrm{quantum}}$ making $T$ queries to $O_f$, there exists a function pair $(F, f) = (\tilde{\mathfrak{F}}_{T;U}, \mathfrak{f}_{r_0,T;U}) \in \Omega$ such that the success probability of $A_{\mathrm{quantum}}$ finding an $\epsilon$-SOSP of $\tilde{\mathfrak{F}}_{T;U}$ is at most

$$\exp\left( -d r_0^2 / (2\alpha^2 T) \right) \leq \frac{1}{3}. \tag{189}$$

In order to guarantee inequality (188), we can require $\nu$ to satisfy

$$\nu \geq \frac{4\alpha^2 (50 + 2\alpha) \lambda \sigma^2 T^2}{d} \geq \Omega(\epsilon^{-16/7}/d). \tag{190}$$

Moreover, by Theorem 5.7 we can conclude that $F = \tilde{\mathfrak{F}}_{T;U}$ is $O(\Delta + \epsilon^{-12/7})$-bounded, $O(1)$-smooth and $O(1)$ Hessian Lipschitz with

$$F(\mathbf{0}) - \inf_{\mathbf{x}} F_{T;U}(\mathbf{x}) \leq \Delta. \tag{191}$$

$\qquad \square$

A similar conclusion can be obtained concerning the setting with first-order oracle access.

**Corollary 5.9** (Formal version of Theorem 1.11, Part 2 )**.** *Let* $\Delta, \epsilon > 0$ *and* $\epsilon \leq 1/2$. *Then there exist positive numerical constants* $c, C, \alpha, \beta \in \mathbb{R}$ *and* $\ell_q \leq e^{\frac{3q}{2}\log q + Cq}$ *for every* $q \in \mathbb{N}$, *and a set* $\Omega$ *consisting of function pairs* $(F, f)$ *satisfying Assumption 1.2 except the smoothness condition of* $f$ *with some* $\tilde{\nu}$ *satisfying*

$$\tilde{\nu} = \Omega(\epsilon^{-8/7}/\sqrt{d}) \tag{192}$$

*such that, for any quantum algorithm* $A_{\text{quantum}}$ *making fewer than* $\Theta(\epsilon^{-12/7})$ *queries to the oracle* $O_f$ *defined in (121) encoding the function value and gradient of* $f$, *there exists a function pair* $(F, f)$ *such that* $A_{\text{quantum}}$ *cannot find an* $\epsilon$-*SOSP of* $F$ *with probability larger than* $1/3$, *where* $F$ *is* $B$ *bounded,* $L_1$-*smooth, and* $L_2$-*Hessian Lipschitz with*

$$B = O\big(\Delta + \epsilon^{-12/7}\big), \quad L_1 = O(1), \quad L_2 = O(1), \tag{193}$$

*and satisfies*

$$F(\mathbf{0}) - \inf_{\mathbf{x}} F_{T;U}(\mathbf{x}) \leq \Delta. \tag{194}$$

*Remark* 5.10. One may notice that in the statement of Corollary 5.9, the hard instance $(F, f)$ we consider only satisfies part of Assumption 1.2 except the smoothness condition of $f$. Nevertheless, adopting a similar smoothing technique presented in Section 4.4, we can modify the hard instance to further satisfy the smoothness condition of $f$ without affecting the asymptotic lower bound.

*Proof.* We adopt the settings of functions and parameters in Theorem 5.7 and set $\Omega$ to be

$$\Omega = \big\{(\tilde{\mathfrak{F}}_{T;U}, \mathfrak{f}_{r_0;T;U}) \,|\, U \in \mathbb{R}^{d \times (T+1)} \text{ s.t. } U^\top U = I\big\}, \tag{195}$$

where

$$T = \frac{\Delta}{20}\Big(\frac{1}{2\ell_1}\Big)^{\frac{3}{7}}\Big(\frac{1}{\ell_2}\Big)^{\frac{5}{4}}(16\epsilon)^{-12/7}. \tag{196}$$

By , the parameter $r_0$ satisfies

$$6\lambda\sigma r_0\sqrt{T} \leq \tilde{\nu}. \tag{197}$$

Moreover, by Theorem 5.7, if the dimension $d$ satisfies

$$d \geq \frac{4\alpha^2 T}{r_0^2}, \tag{198}$$

then for any quantum algorithm $A_{\text{quantum}}$ making $T$ queries to $O_f$, there exists a function pair $(F, f) = (\tilde{\mathfrak{F}}_{T;U}, \mathfrak{f}_{r_0,T;U}) \in \Omega$ such that the success probability of $A_{\text{quantum}}$ finding an $\epsilon$-SOSP of $\tilde{\mathfrak{F}}_{T;U}$ is at most

$$\exp\big(-dr_0^2/(2\alpha^2 T)\big) \leq \frac{1}{3}. \tag{199}$$

In order to guarantee inequality (199), we can require $\tilde{\nu}u$ to satisfy

$$\tilde{\nu} \geq 6\lambda\sigma \cdot \sqrt{4\alpha^2 T^2/d} = \Omega(\epsilon^{-8/7}/\sqrt{d}). \tag{200}$$

Moreover, by Theorem 5.7 we can conclude that $F = \tilde{\mathfrak{F}}_{T;U}$ is $O(\Delta + \epsilon^{-12/7})$-bounded, $O(1)$-smooth and $O(1)$ Hessian Lipschitz with

$$F(\mathbf{0}) - \inf_{\mathbf{x}} F_{T;U}(\mathbf{x}) \leq \Delta. \tag{201}$$

$\square$

## ACKNOWLEDGEMENT

We thank Yizhou Liu for helpful discussions about the quantum tunneling walk in Liu et al. (2023). TL was supported by the National Natural Science Foundation of China (Grant Numbers 92365117 and 62372006), and also the Tianyan Quantum Computing Program.

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

## A  AUXILIARY LEMMAS

**Lemma A.1** (Lemma 3, Ref. (Carmon et al., 2021)). *Consider the function $\bar{F}_{T;\mu}\colon \mathbb{R}^{T+1} \to \mathbb{R}$ defined in Eq. (116), suppose the parameter $\mu$ satisfies $\mu \leq 1$. Then for any $\mathbf{x} \in \mathbb{R}^{T+1}$ such that* [4]

$$|x_T|, |x_{T+1}| \leq \frac{0.1}{T + 1/\sqrt{\mu}}, \tag{202}$$

*we have*

$$\left\|\nabla \bar{F}_{T;\mu}(\mathbf{x})\right\| \geq \mu^{3/4}/8. \tag{203}$$

**Lemma A.2** (Lemma 4, Ref. (Carmon et al., 2021)). *The function $\bar{F}_{T;\mu}(\mathbf{x})$ defined in Eq. (116) satisfies the following.*

1. *$\bar{F}_{T;\mu}(\mathbf{0}) - \inf_{\mathbf{x}} \bar{F}_{T;\mu}(\mathbf{x}) \leq \frac{\sqrt{\mu}}{2} + 10\mu T$;*

2. *For $\mu \leq 1$ and every $p \geq 1$, the $p$-th order derivatives of $\bar{F}_{T;\mu}$ are $(\mathbb{I}\{p = 1\} + \mu)\ell_p$-Lipschitz continuous, where $\ell_p \leq \exp\left(\frac{3p}{2}\log p + cp\right)$ for a numerical constant $c < \infty$.*

**Lemma A.3** (Lemma 3 and Lemma 4, Ref. (Carmon et al., 2021)). *Consider the function $\bar{\mathfrak{F}}_{T;\mu}\colon \mathbb{R}^{T+1} \to \mathbb{R}$ defined in Eq. (156), suppose the parameter $\mu$ satisfies $\mu \leq 1$. Then, there exist positive constants $\alpha, \beta$ such that* [5]

1. *For any $\mathbf{x} \in \mathbb{R}^{T+1}$ satisfying*

$$|x_T|, |x_{T+1}| \leq \frac{0.05}{T + 1/\sqrt{\mu}}, \tag{204}$$

   *its gradient satisfies*

$$\left\|\nabla \bar{\mathfrak{F}}_{T;\mu}(\mathbf{x})\right\| \geq \mu^{3/4}/16; \tag{205}$$

2. *$\bar{\mathfrak{F}}_{T;\mu}(\mathbf{0}) - \inf_{\mathbf{x}} \bar{\mathfrak{F}}_{T;\mu}(\mathbf{x}) \leq \frac{\sqrt{\mu}}{2} + 10\mu T$;*

3. *For every $p > 1$, the $p$-th order derivatives of $\bar{\mathfrak{F}}_{T;\mu}$ are $(2\mathbb{I}\{p = 1\} + \mu)\ell_p$-Lipschitz continuous, where $\ell_p \leq \exp\left(\frac{3p}{2}\log p + cp\right)$ for a numerical constant $c < \infty$.*

**Lemma A.4** (Proposition 14, Ref. (Garg et al., 2021a)). *Let $\mathbf{x} \in \mathbb{B}(\mathbf{0}, 1)$. Then for a $d$-dimensional random unit vector $\mathbf{u}$ and all $c > 0$,*

$$\Pr_{\mathbf{u}}(|\langle \mathbf{x}, \mathbf{u} \rangle| \geq c) \leq 2e^{-dc^2/2}. \tag{206}$$

**Lemma A.5** (Lemma 9, Ref. (Zhang et al., 2021)). *Let $H_1$ and $H_2$ be two Hermitian operators and $H$ be the sum of two operators. For any $t > 0$ and state vector $|\varphi\rangle$, we have*

$$\left\|e^{-iH_1 t}e^{-iH_2 t}|\varphi\rangle - e^{-iHt}|\varphi\rangle\right\| \leq \frac{t^2}{2} \sup_{\tau_1, \tau_2 \in [0,t]} \left\|[H_1, H_2]e^{-iH_2\tau_2}e^{-iH_1\tau_1}|\varphi\rangle\right\|. \tag{207}$$

---

[4]The condition below is a bit different from the original condition in Lemma 3 of (Carmon et al., 2021), which is $x_T = x_{T+1} = 0$. Nevertheless, the following stricter conditions can be achieved with only minor modifications to the original proof.

[5]The formula of the function $\bar{\mathfrak{F}}_{T;\mu}$ is a bit different from the original function considered in Lemma 3 and Lemma 4 of Ref. (Carmon et al., 2021). Nevertheless, this lemma can be proved via only minor modifications to the original proof.

**Lemma A.6** (Corollary 1, Ref. (Zhang et al., 2021)). *Consider a quadratic function of form $F_q = (\mathbf{x} - \mathbf{x}_s)H(\mathbf{x} - \mathbf{x}_s) + f_0$ for Hermitian function $H$ and constant $f_0$, and the Shröödinger equation*

$$i\frac{\partial}{\partial t}\Phi = \left[-\frac{r_0^2}{2}\Delta + \frac{1}{r_0^2}F_q\right]\Phi, \tag{208}$$

*with periodic boundary conditions and initial state in (238). We have*

$$\|\nabla\Phi(t)\| \leq C\sqrt{\frac{d}{r_0}}(\log t)^\alpha \tag{209}$$

*for some constant $\alpha$ and $C$.*

# B TECHNICAL LEMMAS

**Lemma B.1** (Cannot guess stationary point). *Let $k < T$ be a positive in $\{\mathbf{u}^{(1)}, \ldots, \mathbf{u}^{(k)}\}$ be a set of orthonormal vectors. Let $\{\mathbf{u}^{k+1}, \ldots, \mathbf{u}^T\}$ be chosen uniformly at random from $\operatorname{span}(\mathbf{u}^{(1)}, \ldots, \mathbf{u}^{(k)})^\perp$ such that all columns of the matrix $U = [\mathbf{u}^{(1)}, \ldots, \mathbf{u}^{(T+1)}]$ forms a set of orthonormal vectors. Then,*

$$\forall \mathbf{x} \in \mathbb{B}(\mathbf{0}, 2\sigma\sqrt{T}), \quad \Pr_{\{\mathbf{u}^{k+1}, \ldots, \mathbf{u}^{(T+1)}\}}\left[\|\nabla\tilde{F}_{T;U}(\mathbf{x})\| \leq \lambda\sigma\mu^{3/4}/8\right] \leq 8Te^{-dr_0^2/(8T)}, \tag{210}$$

*for the function $\tilde{F}_{T;U}\colon \mathbb{R}^d \to \mathbb{R}$ defined in Eq. (118), given that the parameters $r > 1$ and*

$$r_0 \leq \frac{0.2\sqrt{T}}{T + 1/\sqrt{\mu}}. \tag{211}$$

*Proof.* By Lemma A.4, with probability at least $1 - 8Te^{-dr_0^2/(8T)}$, we have

$$|\langle\mathbf{x}, \mathbf{u}^{(T)}/\sigma\rangle|, |\langle\mathbf{x}, \mathbf{u}^{(T+1)}/\sigma\rangle| \leq \frac{r_0}{2\sqrt{T}} \leq \frac{0.1}{T + 1/\sqrt{\mu}} \tag{212}$$

at the same time. Then by Lemma A.1, we have

$$\|\nabla\tilde{F}_{T;U}(\mathbf{x})\| \leq \lambda\sigma\|\nabla\bar{F}_{T;\mu;r}(U^\top\mathbf{x}/\sigma)\| \leq \lambda\sigma\mu^{3/4}/8. \tag{213}$$

$\square$

**Lemma B.2.** *Consider the functions $\tilde{\mathfrak{F}}_{T;U}\colon \mathbb{R}^d \to \mathbb{R}$ and $\mathfrak{f}_{r_0;T;U}\colon \mathbb{R}^d \to \mathbb{R}$ defined in Eq. (167) and Eq. (168), respectively. We have*

$$\max_\mathbf{x} |\tilde{\mathfrak{F}}_{T;U}(\mathbf{x}) - \mathfrak{f}_{r_0;T;U}(\mathbf{x})| \leq \lambda\sigma^2(50r_0^2 T + 2\alpha r_0\sqrt{T}), \tag{214}$$

*and*

$$\max_\mathbf{x} \|\nabla\tilde{\mathfrak{F}}_{T;U} - \nabla\mathfrak{f}_{r_0;T;U}(\mathbf{x})\| \leq 6\lambda\sigma r_0\sqrt{T}. \tag{215}$$

*where we regard $\nabla\mathfrak{f}_{r_0;T;U}$ to be $\mathbf{0}$ on the sphere $\mathbb{S}(\mathbf{0}, \alpha\sigma\sqrt{T})$ of each hypercube in $\mathbb{R}^d$, given that $\mu, \epsilon \leq 1$ and $r_0 < \sigma$.*

*Proof.* Without loss of generality, we prove this lemma with the input domain being the hypercube $[-\pi\sigma\mathcal{L}/2, \pi\sigma\mathcal{L}/2]^d$ centered at $\mathbf{0}$. Denote the vector $\mathbf{y} \in \mathbb{R}^{T+1}$ to be

$$\mathbf{y} := \left(\gamma_\alpha(\langle\mathbf{u}^{(1)}, \mathbf{x}/\sigma\rangle), \ldots, \gamma_\alpha(\langle\mathbf{u}^{(T+1)}, \mathbf{x}/\sigma\rangle)\right)^\top, \tag{216}$$

which satisfies $\|\mathbf{y}\| \leq \alpha\sqrt{T}$. For the convenience of notations, we denote $y_0 := 0$. Then,

$$\max_\mathbf{x} |\tilde{\mathfrak{F}}_{T;U}(\mathbf{x}) - \mathfrak{f}_{r_0;T;U}(\mathbf{x})| = \lambda\sigma^2 \max_{\mathbf{y}\in\mathbb{B}(\mathbf{0}, \alpha\sqrt{T})} |\bar{F}_{T;\mu}(\mathbf{y}^{\text{prog}}) - \bar{F}_{T;\mu}(\mathbf{y})|, \tag{217}$$

where

$$\mathbf{y}^{\text{prog}} := \left(\gamma_\alpha(\langle\mathbf{x}/\sigma, \mathbf{u}^{(1)}\rangle), \ldots, \gamma_\alpha(\langle\mathbf{x}/\sigma, \mathbf{u}^{(\text{prog}_{r_0}(\gamma_\alpha(\mathbf{x}/\mu))+1)}\rangle), 0, \ldots, 0\right)^\top. \tag{218}$$

Note that

$$\max_{\mathbf{y} \in \mathbb{B}(\mathbf{0}, \alpha\sqrt{T})} \left| \bar{F}_{T;\mu}(\mathbf{y}^{\mathrm{prog}}) - \bar{F}_{T;\mu}(\mathbf{y}) \right| \tag{219}$$

$$\leq \frac{1}{2} \max_{\mathbf{y} \in \mathbb{B}(\mathbf{0}, \alpha\sqrt{T})} \left| \sum_{i=0}^{T} (y_i - y_{i+1})^2 - (y_i^{\mathrm{prog}} - y_{i+1}^{\mathrm{prog}})^2 \right| \tag{220}$$

$$+ \mu \max_{\mathbf{y} \in \mathbb{B}(\mathbf{0}, 2\alpha\sqrt{T})} \left| \sum_{i=1}^{T+1} \left( \Gamma(y_i) - \Gamma(y_i^{\mathrm{prog}}) \right) \right|, \tag{221}$$

where

$$\max_{\mathbf{y} \in \mathbb{B}(\mathbf{0}, \alpha\sqrt{T})} \left| \sum_{i=1}^{T+1} \left( \Gamma(y_i) - \Gamma(y_i^{\mathrm{prog}}) \right) \right| \leq (T+1)(\Gamma(0) - \Gamma(r_0)) \leq 40 T r_0^3, \tag{222}$$

and

$$(y_1 - 1)^2 - (y_1^{\mathrm{prog}} - 1)^2 = 0, \qquad \forall \mathbf{y} \in \mathbb{B}(\mathbf{0}, \alpha\sqrt{T}), \tag{223}$$

since $\mathrm{prog}_{r_0}(\gamma_\alpha(\mathbf{x}/\mu)) + 1 \geq 1$ for all possible $\mathbf{x}$ and corresponding $\mathbf{y}$. As for the term in (220), we note that

$$\max_{\mathbf{y} \in \mathbb{B}(\mathbf{0}, \alpha\sqrt{T})} \left| \sum_{i=1}^{T} (y_i - y_{i+1})^2 - (y_i^{\mathrm{prog}} - y_{i+1}^{\mathrm{prog}})^2 \right| \tag{224}$$

$$\leq 4 T r_0^2 + 2\alpha r_0 \sqrt{T}. \tag{225}$$

Thus we can conclude that

$$\max_{\mathbf{y} \in \mathbb{B}(\mathbf{0}, \alpha\sigma\sqrt{T})} \left| \bar{F}_{T;\mu}(\delta_{r_0}(\mathbf{y})) - \bar{F}_{T;\mu}(\mathbf{y}) \right| \leq 50 r_0^2 T + 2\alpha r_0 \sqrt{T}, \tag{226}$$

and

$$\max_{\mathbf{x}} \left| \tilde{\mathfrak{F}}_{T;U}(\mathbf{x}) - \mathfrak{f}_{r_0;T;U} \right| \leq \lambda \sigma^2 (50 r_0^2 T + 2\alpha r_0 \sqrt{T}). \tag{227}$$

Similarly, we can observe that

$$\max_{\mathbf{x}} \left\| \nabla \tilde{\mathfrak{F}}_{T;U}(\mathbf{x}) - \nabla \mathfrak{f}_{r_0;T;U}(\mathbf{x}) \right\| \leq \lambda \sigma \max_{\mathbf{y} \in \mathbb{B}(\mathbf{0}, \alpha\sqrt{T})} \left\| \nabla \bar{F}_{T;\mu}(\delta_{r_0}(\mathbf{y})) - \nabla \bar{F}_{T;\mu}(\mathbf{y}) \right\|, \tag{228}$$

where

$$\max_{\mathbf{y} \in \mathbb{B}(\mathbf{0}, \alpha\sqrt{T})} \left\| \nabla \bar{F}_{T;\mu}(\delta_{r_0}(\mathbf{y})) - \nabla \bar{F}_{T;\mu}(\mathbf{y}) \right\| \tag{229}$$

$$\leq \frac{1}{2} \max_{\mathbf{y} \in \mathbb{B}(\mathbf{0}, \alpha\sqrt{T})} \left\| \nabla \cdot \sum_{i=0}^{T} \left[ (y_i - y_{i+1})^2 - (y_i^{\mathrm{prog}} - y_{i+1}^{\mathrm{prog}})^2 \right] \right\| \tag{230}$$

$$+ \mu \max_{\mathbf{y} \in \mathbb{B}(\mathbf{0}, \alpha\sqrt{T})} \left\| \nabla \cdot \sum_{i=1}^{T+1} \left[ \Gamma(y_i) - \Gamma(y_i^{\mathrm{prog}}) \right] \right\|, \tag{231}$$

where the term in (231) satisfies

$$\max_{\mathbf{y} \in \mathbb{B}(\mathbf{0}, \alpha\sqrt{T})} \left\| \nabla \cdot \sum_{i=1}^{T+1} \left[ \Gamma(y_i) - \Gamma(y_i^{\mathrm{prog}}) \right] \right\| \leq 2\sqrt{T} \cdot |\Gamma'(r_0) - \Gamma'(0)| \leq 2 r_0 \sqrt{T} \tag{232}$$

As for the term in (230), we first note that for any $0 \leq i \leq T$ and any $\mathbf{y} \in \mathbb{B}(\mathbf{0}, \alpha\sqrt{T})$, we have

$$\frac{1}{2} \left\| \nabla \cdot \left[ (y_i - y_{i+1})^2 - (y_i^{\mathrm{prog}} - y_{i+1}^{\mathrm{prog}})^2 \right] \right\| \leq 2\sqrt{2} r_0, \tag{233}$$

which leads to

$$\max_{\mathbf{y} \in \mathbb{B}(\mathbf{0}, \alpha\sqrt{T})} \left\| \nabla \cdot \sum_{i=1}^{T} \left[ (y_i - y_{i+1})^2 - (y_i^{\mathrm{prog}} - y_{i+1}^{\mathrm{prog}})^2 \right] \right\| \leq 4 r_0 \sqrt{T}. \tag{234}$$

Hence,

$$\max_{\mathbf{y} \in \mathbb{B}(\mathbf{0}, \alpha\sqrt{T})} \left\| \nabla \bar{F}_{T;\mu}(\delta_{r_0}(\mathbf{y})) - \nabla \bar{F}_{T;\mu}(\mathbf{y}) \right\| \le 6 r_0 \sqrt{T}, \tag{235}$$

by which we can conclude that

$$\max_{\mathbf{x}} \left\| \nabla \tilde{\mathfrak{F}}_{T;U}(\mathbf{x}) - \nabla \mathfrak{f}_{r_0;T;U}(\mathbf{x}) \right\| \le 6 \lambda \sigma r_0 \sqrt{T}. \tag{236}$$

$\square$

## C  PERTURBED GRADIENT DESCENT WITH QUANTUM SIMULATION AND GRADIENT ESTIMATION

In this section, we consider an alternative version of Algorithm 3 that has a faster convergence rate in some cases. Inspired by Ref. (Zhang et al., 2021), we replace the uniform perturbation in Algorithm 3 with quantum simulation. We consider the scaled evolution under Schrödinger equation

$$i \frac{\partial}{\partial t} \Phi = \left[ -\frac{r_0^2}{2} \Delta + \frac{1}{r_0^2} f \right] \Phi, \tag{237}$$

where $\Phi$ is a wave function in $\mathbb{R}^d$, $\Delta$ is the Laplacian operator, $r_0$ is the scaling parameter, and $f$ is the potential of the evolution. To construct a quantum algorithm using this evolution, quantum simulations are required. There is rich literature on the cost of quantum simulations (Berry et al., 2007; 2015; Childs, 2017; Lloyd, 1996; Low & Chuang, 2017; 2019). Here, we introduce the following theorem concerning the cost of simulating (237) using zeroth-order oracle $F$, which was originally proposed in Ref. (Zhang et al., 2021).

**Lemma C.1** (Lemma 2, Ref. (Zhang et al., 2021)). *Let $F(\mathbf{x}) \colon \mathbb{R}^d \to \mathbb{R}$ be a real-valued function that has a saddle point at $\mathbf{x} = 0$ such that $f(\mathbf{0}) = 0$. Consider the (scaled) Schrödinger equation in (237) defined on the domain $\Omega = \{\mathbf{x} \in \mathbb{R}^d \colon \|\mathbf{x}\| \le M\}$ with periodic boundary condition, where $M > 0$ is the diameter specified later. Given the noiseless zeroth-order oracle $U_f(|\mathbf{x}\rangle \otimes |0\rangle) = |\mathbf{x}\rangle \otimes |f(\mathbf{x})\rangle$ and an arbitrary initial state. The evolution for time $t > 0$ can be simulated using $\tilde{O}(t \log d \log^2(t/\epsilon))$ queries to $U_f$, where $\epsilon$ is the simulation precision.*

Notice that $F$ is assumed to be Hessian Lipschitz in both Assumption 1.1 and Assumption 1.2, we can approximate the function value near a saddle point. The approximation is more accurate on a ball with radius $r_0$ centered at this saddle point. We scale the initial distribution and the Schrödinger equation to be localized in term of $r_0$ and results in Algorithm 6, which is originally proposed in Ref. (Zhang et al., 2021).

---

**Algorithm 6:** QuantumSimulation($\tilde{\mathbf{x}}, r_0, t_e, f(\cdot)$).

1  Evolve a Gaussian wave packet in the potential field $f$, with its initial state being:

$$\Phi_0(\mathbf{x}) = \left(\frac{1}{2\pi}\right)^{n/4} \frac{1}{r_0^{n/2}} \exp\left(-(\mathbf{x} - \tilde{\mathbf{x}})^2 / 4 r_0^2\right); \tag{238}$$

Simulate such evolution in potential field $f$ under the Schrödinger equation for time $t_e$;
2  Measure the position of the wave packet and output the outcome.

---

Algorithm 6 is the main building block of the quantum implementation of perturbation in PGD. It can effectively reduce the iteration number compared to the classical perturbations in Algorithm 3 (Zhang et al., 2021) for some functions. To achieve a better performance than Theorem 2.8, we have to add some constraints on the target and the noisy function $(F, f)$. Specifically, we consider the following setting.

**Assumption C.2.** The underlying target function $F$ is $B$-bounded, $\ell$-smooth, and $\rho$-Hessian Lipschitz. We can query a noisy function $f$ that is twice differentiable. We assume

$$\nu = \sup_{\mathbf{x}} \|F(\mathbf{x}) - f(\mathbf{x})\|_\infty = \tilde{O}\left(\frac{\epsilon^6}{d^4}\right), \tag{239}$$

$$\tilde{\nu} = \sup_{\mathbf{x}} \|\nabla F(\mathbf{x}) - \nabla f(\mathbf{x})\|_\infty = \ell_f M = O\left(d^{-3}\right), \tag{240}$$

$$\hat{\nu} = \sup_{\mathbf{x}} \|\nabla^2 F(\mathbf{x}) - \nabla^2 f(\mathbf{x})\|_\infty = \rho_f M = O\left(d^{-3}\right), \tag{241}$$

where $\ell_f$ and $\rho_f$ are arbitrary constants, and $M = O(d^{-3})$ is some value to be fixed later.

By using quantum simulations to implement perturbations, we propose the following Algorithm 7 that can effectively find an $\epsilon$-SOSP of $F$ using queries to noisy $f$ for function pair $(F, f)$ in Assumption C.2 with high probability.

---

**Algorithm 7:** Perturbed Gradient Descent with Quantum Gradient Computation.

---

**Input:** $\mathbf{x}_0$, learning rate $\eta$, noise ratio $r$

1: **for** $t = 0, 1, \ldots, T$ **do**
2:      Apply Lemma 2.1 to compute an estimate $\tilde{\nabla}F(\mathbf{x})$ of $\nabla F(\mathbf{x})$
3:      **if** $\left\|\tilde{\nabla}F(\mathbf{x}_t)\right\| \le \epsilon$ **then**
4:          $\xi \sim \text{QuantumSimulation}\left(\mathbf{x}_t, r_0, \mathscr{T}, f(\mathbf{x}) - \langle\tilde{\nabla}F(\mathbf{x}_t), \mathbf{x} - \mathbf{x}_t\rangle\right)$
5:          $\Delta_t \leftarrow \frac{2\xi}{3\|\xi\|}\sqrt{\frac{\rho^*}{\epsilon}}$
6:          $\mathbf{x}_t \leftarrow \arg\min_{\zeta \in \{\mathbf{x}_t + \Delta_t, \mathbf{x}_t - \Delta_t\}} f(\zeta)$
7:      **end if**
8:      $\mathbf{x}_{t+1} \leftarrow \mathbf{x}_t - \eta\tilde{\nabla}F(\mathbf{x})$
9: **end for**

---

Algorithm 7 has the following performance guarantee:

**Theorem C.3.** *Suppose we have a target function $F$ and its noisy evaluation $f$ satisfying Assumption C.2 with $\nu \le \tilde{O}(\delta^2\epsilon^6/d^4)$. Then with probability at least $1 - \delta$, Algorithm 7 can find an $\epsilon$-SOSP of $F$ satisfying (4), using*

$$\tilde{O}\left(\frac{\ell B}{\epsilon^2} \cdot \log^2 d\right) \tag{242}$$

*queries to $U_f$, under the following parameter choices:*

$$\ell' = \ell + \frac{1}{2}\ell_f, \quad \rho' = \rho + \rho_f, \quad \eta = \frac{1}{\ell'}, \quad \delta_0 = \min\left\{\frac{\delta}{162B}\frac{\epsilon^3}{\rho'}, \frac{\eta\epsilon^2\delta}{16B}\right\}, \quad \mathscr{F} = \frac{2}{81}\sqrt{\frac{\epsilon^3}{\rho'}}, \tag{243}$$

$$\mathscr{T} = \frac{8}{(\rho'\epsilon)^{1/4}}\log\left(\frac{\ell'}{\delta_0\sqrt{\rho'\epsilon}}\left(d + 2\log\left(\frac{3}{\delta_0}\right)\right)\right), \quad r_0 = \frac{4c_r^3}{9\mathscr{T}^4}\left(\frac{\delta_0}{3}\cdot\frac{1}{d^{3/2} + 2c_0 d\ell'(\log\mathscr{T})^\alpha}\right)^2, \tag{244}$$

*where $c_0$, $\alpha$, and $c_r$ are absolute constants specified in the proof.*

Before proving Theorem C.3, we first consider the effectiveness of quantum simulations for adding perturbations. We focus on the scenarios with $\epsilon \le \ell^2/\rho$, which is the standard assumption adopted in Ref. (Jin et al., 2018b). The local landscape in this case "flat" and the Hessian has only a small spectral radius. The classical gradient descent will move slowly while the variance of the probability distribution corresponding to the Gaussian wavepacket still has a large increasing rate. If we evolve the Gaussian wavepacket for a long enough time period and measure its position, we will obtain a vector that indicates a negative curvature direction with high probability.

However, Algorithm 6 using quantum simulation and noisy oracle in Assumption C.2 suffers two deviation terms from the ideal Gaussian evolution: the deviation of $F$ from quadratic potential and

the noise of $f$ from $F$. We have to bound the resulting deviation on the distribution from the perfect Gaussian wavepacket. We specify the constant $c_r$ as the ratio between the wavepacket variance and the radius of the simulation region. By choosing a small enough $c_r$, the simulation region is much larger than the range of the wavepacket. As the function $F$ is $\ell$-smooth, the spectral norm of the Hessian matrix is upper bounded by constant $\ell$. The radius $M$ of the simulation region is chosen as

$$M = \frac{r_0}{C_r} = \frac{4c_r^2}{9\mathscr{T}^4}\left(\frac{\delta_0}{3}\cdot\frac{1}{d^{3/2}+2c_0d\ell(\log\mathscr{T})^\alpha}\right)^2 \le 1. \tag{245}$$

By choosing the above $M$, we can reach the following lemma.

**Lemma C.4.** *Under the setting of Assumption C.2 and Theorem 1.4, let $H$ be the Hessian matrix of $F$ at the saddle point $\mathbf{x}_s$, and define $F_q(\mathbf{x}) = f(\mathbf{x}_s) + (\mathbf{x}-\mathbf{x}_s)^\top H(\mathbf{x}-\mathbf{x}_s)$ to be the quadratic approximation of $F$ near $\mathbf{x}_s$. We denote the measurement outcome from Algorithm 6 with noisy function $f$ and evolution $t_e$ as a random variable $\xi$, and the measurement outcome from the ideal potential $F_q$ and the same evolution time $t_e$ as another random variable $\xi'$. We define $\mathbb{P}_\xi$ and $\mathbb{P}_{\xi'}$ to be the distribution of $\xi$ and $\xi'$. If the quantum wavepacket is confined to a hypercube with regions length $M$, then*

$$TV(\mathbb{P}_\xi, \mathbb{P}_{\xi'}) \le \left(\frac{\sqrt{d}\rho'}{2} + \frac{2c_f\ell'}{\sqrt{r_0}}(\log t_e)^\alpha\right)\frac{dMt_e^2}{2}, \tag{246}$$

*where $TV(\cdot, \cdot)$ denotes the total variation distance, $\alpha$ is an absolute constant, and $c_f$ is an $F$-related constant.*

*Proof.* We first define the following notations:

$$A = -\frac{r_0^2}{2}\Delta, \quad B = \frac{1}{r_0^2}f, \quad B' = \frac{1}{r_0^2}F_q, \tag{247}$$

$$H = A + B, \quad H' = A + B', \quad E = H - H' = \frac{1}{r_0^2}(f - F_q). \tag{248}$$

We denote $|\Phi(t)\rangle = e^{-iHt}|\Phi_0\rangle$ and $|\Phi'(t)\rangle = e^{-iH't}|\Phi_0\rangle$ be the wave functions at time $t$ for two different Hamiltonians $H$ and $H'$. By Lemma A.5, we have

$$\left\||e^{iEt_e}|\Phi'(t_e)\rangle - |\Phi(t_e)\rangle\right\| \le \frac{t_e^2}{2}\sup_{\tau_1,\tau_2\in[0,t_e]}\left\|[H',E]e^{-iE\tau_2}e^{-iH'\tau_1}|\Phi_0\rangle\right\| \tag{249}$$

$$= \frac{t_e^2}{2}\sup_{\tau_1\in[0,t_e]}\left\|[H',E]e^{-iH'\tau_1}|\Phi_0\rangle\right\|. \tag{250}$$

Denoting $|\Psi(\tau_1)\rangle = e^{-iH'\tau_1}|\Phi_0\rangle$, we have

$$\sup_{\tau_1\in[0,t_e]}\|[H',E]\Psi(\tau_1)\| = \frac{1}{2}\sup_{\tau_1\in[0,t]}\|[-\Delta, f - F_q]\Psi(\tau_1)\| \tag{251}$$

$$= \frac{1}{2}\sup_{\tau_1\in[0,t_e]}\|-\Delta(f - F_q)\Psi(\tau_1) - 2\nabla(f - F_q)\cdot\nabla\Psi(\tau_1)\| \tag{252}$$

$$\le \frac{1}{2}\|\Delta(f - F_q)\|_\infty + \|\nabla(f - F_q)\|_\infty\|\nabla\Psi(\tau_1)\|. \tag{253}$$

The first equality follows from $[H',E] = [A + B', E]$ and $B'$ commutes with $E$. The second equality follows from $[-\Delta, g]\varphi = -(\Delta g)\varphi - 2\nabla g \cdot \nabla\varphi$ for any smooth function $\varphi$ and $g$. As we assume $F$ is $\rho$-Hessian Lipschitz, we can deduce that

$$|\Delta(f(\mathbf{x}) - F_q(\mathbf{x}))| = \left|\mathrm{tr}(\nabla^2 f(\mathbf{x}) - \nabla^2 F(\mathbf{x}))\right| + \left|\mathrm{tr}(\nabla^2 F(\mathbf{x}) - \nabla^2 F_q(\mathbf{x}))\right| \tag{254}$$

$$= \left|\mathrm{tr}(\nabla^2 f(\mathbf{x}) - \nabla^2 F(\mathbf{x}))\right| + \left|\mathrm{tr}(\nabla^2 F(\mathbf{x}) - \nabla^2 F(\mathbf{x}_s))\right| \tag{255}$$

$$\le d\left\|\nabla^2 f(\mathbf{x}) - \nabla^2 F(\mathbf{x})\right\| + d\left\|\nabla^2 F(\mathbf{x}) - \nabla^2 F(\mathbf{x}_s)\right\| \tag{256}$$

$$\le d^{3/2}(\rho + \rho_f)M \tag{257}$$

$$= d^{3/2}\rho'M. \tag{258}$$

Next, we bound the term on the gradient of $f - F_q$:

$$\|\nabla(f - F_q)\|_\infty \leq \sup_{\mathbf{x}} \|\nabla f(\mathbf{x}) - \nabla F_q(\mathbf{x})\| \tag{259}$$

$$= \sup_{\mathbf{x}} \|\nabla f(\mathbf{x}) - \nabla F(\mathbf{x})\| + \sup_{\mathbf{x}} \|\nabla F(\mathbf{x}) - H(\mathbf{x} - \mathbf{x}_s)\| \tag{260}$$

$$\leq \sup_{\mathbf{x}} \|\nabla f(\mathbf{x}) - \nabla F(\mathbf{x})\| + \sup_{\mathbf{x}} \|\nabla F(\mathbf{x})\| + \sup_{\mathbf{x}} \|H(\mathbf{x} - \mathbf{x}_s)\| \tag{261}$$

$$\leq (2\ell + \ell_f) M d^{-1/2} \tag{262}$$

$$= 2\ell' M d^{-1/2} \tag{263}$$

The upper bound for $\sup_{\tau_1 \in [0, t_e]} \|\nabla \Psi(\tau_1)\|$ is given by Lemma A.6. Combining the above bounds, we obtain

$$\left\| e^{iEt_e} \left|\Phi'(t_e)\right\rangle - \left|\Phi(t_e)\right\rangle \right\| \leq \left( \frac{\sqrt{d}\rho'}{2} + \frac{2c_f \ell'}{\sqrt{r_0}} (\log t_e)^\alpha \right) \frac{dM t_e^2}{2}. \tag{264}$$

In the following part, we denote $\Psi'$ for $\Psi'(t_e)$ and $|\Psi''\rangle = e^{-iEt_e} |\Psi'\rangle$. We observe that $|\Psi'|^2 = |\Psi''|^2$ as $e^{-iEt_e}$ is a scalar function with modulus 1. Thus

$$TV(\mathbb{P}_\xi, \mathbb{P}_{\xi'}) = TV(|\Psi|^2, |\Psi''|^2) \tag{265}$$

$$= \frac{1}{2} \int_{\mathbf{x}} \left| \Psi\Psi^\dagger - \Psi''\Psi''^\dagger \right| d\mathbf{x} \tag{266}$$

$$\leq \frac{1}{2} \int_{\mathbf{x}} \left| (\Psi - \Psi'')\Psi^\dagger \right| d\mathbf{x} + \frac{1}{2} \int_{\mathbf{x}} \left| \Psi''(\Psi - \Psi'')^\dagger \right| d\mathbf{x} \tag{267}$$

$$\leq \left( \frac{1}{2} \int_{\mathbf{x}} |\Psi - \Psi''|^2 d\mathbf{x} \right)^{1/2} \tag{268}$$

$$\leq \left( \frac{\sqrt{d}\rho'}{2} + \frac{2c_f \ell'}{\sqrt{r_0}} (\log t_e)^\alpha \right) \frac{dM t_e^2}{2}. \tag{269}$$

$\square$

Lemma C.4 indicates that the actual perturbation given by quantum simulation $\xi \sim \mathbb{P}_\xi$ deviates from the ideal Gaussian case $\xi' \sim \mathbb{P}_{\xi'}$ for at most $\tilde{O}(Md^{3/2}t_e^2)$. In Algorithm 7 with $t_e = \mathscr{T} = O(\log d)$, such deviation can be bounded for the choice of $M$ in (245). Based on Lemma C.4, we reach the following lemma.

**Lemma C.5** (Adaptive version of Proposition 1, Ref. (Zhang et al., 2021))**.** *Suppose $(F, f)$ satisfies Assumption C.2. For arbitrary $\delta_0$, we choose the following parameters:*

$$\eta = \frac{1}{\ell'}, \quad \mathscr{T} = \frac{8}{(\rho'\epsilon)^{1/4}} \log\left( \frac{\ell'}{\delta_0\sqrt{\rho'\epsilon}} \left(d + 2\log\left(\frac{3}{\delta_0}\right)\right) \right), \tag{270}$$

$$\mathscr{F} = \frac{2}{81}\sqrt{\frac{\epsilon^3}{\rho'}}, \quad r_0 = \frac{4c_r^3}{9\mathscr{T}^4}\left( \frac{\delta_0}{3} \cdot \frac{1}{d^{3/2} + 2c_0 d\ell'(\log \mathscr{T})^\alpha} \right)^2, \tag{271}$$

*where $\rho'$, $\ell'$, $c_r$, $c_0$, and $\alpha$ are the same with Theorem C.3. Then, for an saddle point $\mathbf{x}_s$ with $\|F(\mathbf{x}_s)\| \leq \epsilon$ and $\lambda_{\min}(\nabla^2 F(\mathbf{x}_s)) \leq -\sqrt{\rho\epsilon}$, Algorithm 7 provides a perturbation that decreases the function value for at least $\mathscr{F}$ with probability at least $1 - \delta_0$.*

We now prove Theorem C.3.

*Proof.* We set the zeroth-order noise bound $\nu \leq c_\nu \delta^2 \epsilon^6 / d^4$ for small enough $c_\nu$ and let the total total iteration number to be

$$T = 4\max\left\{ \frac{2B}{\mathscr{F}}, \frac{4B}{\eta\epsilon^2} \right\} = \tilde{O}\left( \frac{B}{\epsilon^2} \cdot \log d \right). \tag{272}$$

We first consider the iteration number at saddle points $\mathbf{x}_t$ with $\|F(\mathbf{x}_t)\| \leq \epsilon$ and $\lambda_{\min}(\nabla^2 F(\mathbf{x}_t)) \leq -\sqrt{\rho\epsilon}$. According to Lemma C.5, each iteration of Algorithm 7 in this case will decrease the function value for at least $\mathscr{T}$. Under this assumption, Algorithm 6 can be called for at most $T/2$ times, for otherwise the function value decreases greater than $2B \geq F(\mathbf{x}^*) - F(\mathbf{x}_0)$. The failure probability is bounded by

$$81\sqrt{\frac{\rho}{\epsilon^3}} \cdot \delta_0 = \frac{\delta}{2}. \tag{273}$$

Except for these iterations that quantum simulation is implemented to add perturbations, we still have $T/2$ iterations. We consider the iterations with large gradients $\|F(\mathbf{x}_t)\| \geq \epsilon$. In each iteration, the function value will decrease at least $\eta\epsilon^2/4$. There can be at most $T/4$ iterations, for otherwise the function value decreases greater than $2B \geq F(\mathbf{x}^*) - F(\mathbf{x}_0)$. The failure probability is bounded by

$$\frac{4B}{\eta\epsilon^2} \cdot \delta_0 = \delta/2. \tag{274}$$

In summary, with probability at least $1 - \delta$, there are at most $T/2$ iterations when the quantum simulation is called and at most $T/4$ iterations when the gradient is large. There are thus at least $T/4$ iterations resulting in $\epsilon$-SOSP of $F$.

The number of queries can be decomposed into two parts, the number of queries required for gradient estimations, denoted by $T_1$, and the number of queries required for quantum simulations, denoted by $T_2$. For the first part, we have

$$T_1 = O(T) = \tilde{O}\left(\frac{B}{\epsilon^2} \cdot \log d\right). \tag{275}$$

For $T_2$, the number of queries is given by Lemma C.1 as

$$T_2 = \tilde{O}\left(\frac{B}{\epsilon^2} \cdot \log^2 d\right). \tag{276}$$

The total query complexity $T_1 + T_2$ is bounded by

$$\tilde{O}\left(\frac{B}{\epsilon^2} \cdot \log^2 d\right). \tag{277}$$

$\square$

## D    EXISTENCE OF EXAMPLE WITH QUANTUM ADVANTAGE USING QUANTUM TUNNELLING WALK

In Theorem 1.8, we have proved that when the noise under Assumption 1.1 increases to $\nu \geq \Theta(\epsilon^{1.5})$, we can find a hard instance for any classical or quantum algorithm even using exponentially many queries. Although the noise bound for the quantum algorithms is the same, there might be quantum speedup for a specific instance. In this section, we provide a candidate for this argument under some proper additional assumptions.

We set the constant $\mu = 300$. For the target function $F$, we still consider the following function defined in (88):

$$F(\mathbf{x}) := h(\sin \mathbf{x}) + \|\sin \mathbf{x}\|^2, \tag{278}$$

where $h(\mathbf{x}) := h_1(\mathbf{v}^\top \mathbf{x}) \cdot h_2\left(\sqrt{\|\mathbf{x}\|^2 - (\mathbf{v}^\top \mathbf{x})^2}\right)$, and

$$h_1(x) = g_1(\mu x), \quad g_1(x) = (-16|x|^5 + 48x^4 - 48|x|^3 + 16x^2) \cdot \mathbb{I}\{|x| < 1\}, \tag{279}$$

$$h_2(x) = g_2(\mu x), \quad g_2(x) = (3x^4 + 8|x|^3 + 6x^2 - 1) \cdot \mathbb{I}\{|x| < 1\}. \tag{280}$$

We adopt the construction in Ref. (Liu et al., 2023) for the construction of noisy function $f$. In the following, we denote $R$ to be the radius of the hyperball where the main construction is. We also choose $\mathbf{v}$ uniformly in the unit sphere. We define two regions $W_- = \mathbb{B}(0, a)$ and $W_+ = \nvDash \mathbf{v}, \eth$ with $b \geq a$. We choose $a$ and $b$ such that $W_-$ and $W_+$ are in $\mathbb{B}(0, R)$. We denote the region $S_{\mathbf{v}} := \{\mathbf{x} | \mathbf{x} \in \mathbb{B}(0, R), |\mathbf{x} \cdot \mathbf{v}| \leq w\}$, where $w$ is chosen in $[0, \sqrt{3}w/2)$. We define

$$B_{\mathbf{v}} := \{\mathbf{x} | w < \mathbf{x} \cdot \mathbf{v} < 2b - w, \sqrt{\|\mathbf{x}\|^2 - (\mathbf{x} \cdot \mathbf{v})^2} < \sqrt{a^2 - w^2}, \mathbf{x} \notin W_- \cup W_+\}. \tag{281}$$

The construction of $f$ is given by

$$f = \begin{cases} \frac{1}{2}\omega^2\|\mathbf{x}\|^2, & \mathbf{x} \in W_-, \\ \frac{1}{2}\omega^2\|\mathbf{x} - 2b\mathbf{v}\|^2, & \mathbf{x} \in W_+, \\ H_1, & \mathbf{x} \in B_{\mathbf{v}}, \\ H_2, & \text{otherwise.} \end{cases} \tag{282}$$

We choose $0 < \omega^2 a^2/2 \sim H_1 \ll H_2$. There are two local minima for $f$ in (282), $\mathbf{0}$ and $2b\mathbf{v}$. We can verify that the function pair $(F, f)$ satisfies the following properties.

- $\sup_x \|f - F\|_\infty \leq \tilde{O}(1)$.
- $F$ is $O(d)$-bounded, $O(1)$-Hessian Lipschitz, and $O(1)$-gradient Lipschitz.

We apply the same scaling in the main text as

$$\tilde{F}(\mathbf{x}) := \epsilon r F\left(\frac{\mathbf{x}}{r}\right), \tag{283}$$

$$\tilde{f}(\mathbf{x}) := \epsilon r f\left(\frac{\mathbf{x}}{r}\right), \tag{284}$$

where $r = \sqrt{\epsilon/\rho}$. According to Ref. (Liu et al., 2023), quantum tunneling walk can provide a speedup for finding SOSP of $f$ using ground states containing information of $\mathbf{v}$ and $W_+$.

**Lemma D.1** (Proposition 4.2 and Theorem 4.1, Ref. (Liu et al., 2023)). *Assume we start from point $\mathbf{0}$ and we are provided with knowledge that $\mathbf{0}$ is a local minimum. We know local ground states associated with $W_-$ and $W_+$. By properly choosing the parameter $a$, $b$, $H_1$, and $H_2$, quantum tunneling walk can find the another local minima with high probability using $O(\text{poly}(d))$ queries while any classical algorithm requires $\Omega(e^{dB})$ queries to zeroth-order oracle $f$.*

Under the setting of this paper, we consider choosing $R$ such that $2b\mathbf{v}$ is also a local minimum of $F$. Notice that $\mathbf{0}$ is not a local minimum of $F$, our goal is to find the $\epsilon$-SOSP near the local minima $2b\mathbf{v}$ of $F$ taking queries to noisy oracle $f$ in (7). We can reach the following corollary using Lemma D.1.

**Corollary D.2.** *Consider the hard instance $(\tilde{F}, \tilde{f})$ defined by (283), (284), and a proper chosen $R$ such that $2b\mathbf{v}$ is also a local minimum of $\tilde{F}$. There exists a choice of parameters $a$, $b$, $H_1$, and $H_2$ such that a quantum algorithm starting at $\mathbf{0}$ can find an $\epsilon$-SOSP of $\tilde{F}$ with high probability using $O(\text{poly}(d))$ queries to the noisy $\tilde{f}$ and proper initial ground state. However, any classical algorithm with proper initial ground states requires $\Omega(e^{dB})$ queries.*

The above corollary demonstrates that if we assume that we have some ground states revealing information above $\mathbf{v}$, the quantum algorithm can provide an exponential speedup in solving a special hard instance $(F, f)$ that satisfies Assumption 1.1. It is worthwhile to mention that the additional assumption on the local ground state is essential for this speedup and the quantum algorithm also requires query complexity that is exponential in $d$ without such assumption (Liu et al., 2023).