# OpenReview forum: "Robustness of Quantum Algorithms for Nonconvex Optimization"
_ICLR.cc/2025/Conference — ICLR 2025 Poster_

### Official Review · Reviewer_W1b5 · 2024-10-29

**Soundness:** 2
**Presentation:** 1
**Contribution:** 3
**Rating:** 6
**Confidence:** 3

**Summary:**

This paper studies how to apply quantum algorithms for finding the $\epsilon$-approximate second-order stationary point ($\epsilon$-SOSP) of a $d$-dimensional non-convex function, given noisy zeroth- (i.e., noisy evaluation query) or first-order(i.e., noisy gradient query) oracles as inputs. The study focuses on settings where only noisy zeroth-order (evaluation queries) or noisy first-order (gradient queries) oracles are accessible. Specifically, the authors analyze scenarios under various levels of noise, parameterized by poly $(\epsilon, 1/d)$,  and demonstrate that their approach significantly outperforms state-of-the-art (SoTA) methods regarding query complexity. Furthermore, they establish several lower bounds for alternative scenarios where efficient query complexity is not achievable with the current methods. Tables 1, 2 and 3 concisely summarize the main results, offering a comparative view of query complexity improvements relative to existing methods.

**Strengths:**

- Applying quantum oracles to speedup existing results is a timely and meaningful interesting problem.
- This work gives a dense collection of results.
- The summarization of the results in Tables 1, 2 and 3 is informative and clear.

**Weaknesses:**

- The presentation of technical results in this paper feels somewhat vague; for instance, there is no pseudocode or intuitive explanation of the underlying algorithms. This lack of detail impacts the paper's clarity and readability.

- The current presentation makes it challenging to assess the paper's novelty. For instance, when referencing Jordan's algorithm, the paper points to [2] for details on Jordan's gradient estimation (line 223) and states that it "replaces the gradient queries in PAGD with this oracle" (line 232). However, there is insufficient discussion on the complexity of replacing the oracle, which obscures an evaluation of the paper's originality.

- There are no related discussions on when the noise levels are practical, for example, what problem setting would generate a noise level of $\tilde{\Omega} (\epsilon^{1.5} / d)$?

- The definition of $\epsilon$-SOSP doesn't align with the one in [1]. Instead of letting $\|F(x)\| \leq \epsilon$, the saddle point should satisfy $\|\nabla F(x)\| \leq \epsilon$?


- Several notations are used for the same purpose in this work, for example, [line 119-130], $\||$ , | |, and | are used to denote similar norms. The reviewer would recommend using one notation to denote all the norms.

Minor Comments:
- Some $O(\cdot)$ sign probably should be $\Theta(\cdot)$ in the column "noise strength" in Table 1?

[1] Jin, C., Ge, R., Netrapalli, P., Kakade, S. M., & Jordan, M. I. (2017, July). How to escape saddle points efficiently. In International conference on machine learning (pp. 1724-1732). PMLR.

[2]Chakrabarti, S., Childs, A. M., Li, T., & Wu, X. (2020). Quantum algorithms and lower bounds for convex optimization. Quantum, 4, 221.

**Questions:**

- It appears that $\nabla f(x+z) - \nabla f(x)$ is intended as an estimate of the Hessian rather than the gradient, as indicated in line 312?

-The most recent work that this paper compares against in Table 1 is from 2018. Are there been any more recent results in this direction?

---

> ### Author Response · Authors · 2024-11-21
>
> We appreciate the reviewer for the great effort and valuable suggestions.
>
> We thank the reviewer for proposing the comment of ambiguity in the presentation of our results and pseudocodes. However, we would like to point out that every algorithmic result in our manuscript is accompanied by a corresponding pseudocode in the supplementary material. We agree with the reviewer that the lack of pointers in the main body to the relevant pseudocodes affects readability. We have addressed this issue in the revised version by adding appropriate references in the main body.
>
> Regarding the complexity of Jordan’s gradient estimation, we added the precise bound shown in Chakrabarti et al. (2020) (currently Lemma 2.1 at the beginning of Section 2), and we also polished the discussion at the beginning of Section 2.2 on our contributions.
>
> We thank the reviewer for raising the question of whether the noise levels considered in this work are practical. Although a noise level like $O(\epsilon^{1.5}/d)$ seems to be small, we would like to point out that: (i) If we interpret a noisy query of the value (gradient) of the loss function as a result of empirical estimations of actual values (gradients) of the loss function, then a small number of polylog$(1/\epsilon,d)$ samples are sufficient to estimate the loss function within this noise regime. In practice, it is reasonable to assume that we have enough data samples to control the empirical risk within such a regime. (ii) If we interpret a noisy query of the value (gradient) of the loss function as a result of quantum noise, then our results indicate that, although providing potential exponential speedups, quantum algorithms are not robust against small noise in the worst case.
>
> The reviewer also raised the question of the complexity of replacing the gradient oracle in PAGD with Jordan’s algorithm. We would like to kindly ask the reviewer to refer to Section 2.1 of our supplementary material for a more detailed discussion. We agree with the author that the current presentation in the short version of the manuscript omits many details on complexity analyses of algorithms and proofs of lower bounds. In the revised version, we have addressed this issue by adding more descriptions at the beginning of Section 2.
>
> We thank the reviewer for pointing out the typo in the definition of $\epsilon$-SOSP and the notation for norms. We have fixed them in our revised version.
>
> The reviewer also suggested that some $O(\cdot)$ should be $\Theta(\cdot)$ in Table 1. However, we would like to point out that here we use $O(\cdot)$ and $\Omega(\cdot)$ to provide the upper bound and lower bound of the noise regime, respectively.
>
> We thank the reviewer for pointing out a mistake regarding how to provide a stochastic gradient function of the Gaussian smoothing of $f$. Instead of $\nabla f(x+z)-\nabla f(x)$, it should be $\nabla f(x+z)$ that provides a stochastic gradient function of the Gaussian smoothing of $f$. We have fixed this point in the revised version of our manuscript and provided further explanations.
>
> Finally, regarding the question on recent classical results in Table 1, to the best of our knowledge, we are not aware of any new relevant literature after 2018.

---

> > ### Comment · Reviewer_W1b5 · 2024-12-03
> > **Feedback and Clarifications**
> >
> > Thanks the authors for the rebuttal! I would like to clarify a few points in my initial review:
> >
> > - **Noise Levels and Practicality**
> >   I appreciate the detailed results on various noise levels presented in the paper. However, I would like to clarify my earlier comment: *"There are no related discussions on when the noise levels are practical, for example, what problem setting would generate a noise level of \(\tilde{\Omega} (\epsilon^{1.5} / d)\)?"*
> >   My intent was to ask whether **there exist problem settings where these specific noise levels are practically relevant**. For example, in the smooth analysis regime, it is typical to assume that natural processes introduce Gaussian noise with specific levels. Since this paper addresses various noise levels, it would strengthen the work to include an explanation of the intuition behind studying these levels. Additionally, **highlighting any optimization scenarios that naturally produce such noise levels would provide valuable context** and make the results even more compelling.
> >
> > - **Presentation Quality**
> >   While I found the core ideas of the paper intriguing, I believe the presentation could benefit from further refinement. Some errors I noted in the manuscript appear to be genuine mistakes, and I encourage the authors to conduct a thorough review to ensure there are no **issues with definitions or foundational properties**. Addressing these would enhance the clarity and reliability of the paper.
> >
> > - **Relevance of the Topic**
> >   Regarding the statement, *"To the best of our knowledge, we are not aware of any new relevant literature after 2018,"* I would encourage **the authors to clarify the significance of this work in light of the apparent decline in attention to this problem area**. It would be helpful to emphasize why this research is still meaningful and how it contributes to advancing the field.
> >
> > I also noticed that Reviewer **kwBc** raised similar concerns regarding the novelty of this paper. I will discuss this further with the AC and other reviewers before finalizing my score.

---

> ### Author Response · Authors · 2024-12-03
>
> Dear Reviewers,
>
> We would like to express our gratitude for the time and effort you have dedicated to reviewing our paper. We have carefully considered your feedback and provided a detailed response in rebuttal. We are eager to hear your opinions and to address any further concerns you may have.
>
> Thank you once again for your dedication, and we look forward to your response.

---

> ### Author Response · Authors · 2024-12-04
>
> We appreciate the reviewer for the follow-up questions and suggestions.
>
> **Noise Levels and Practicality:** We thank the reviewer for clarifying the previous comment. In the following, we would like to provide two meaningful contexts for the noise levels considered in our work from classical optimization and quantum computing perspectives, respectively. We would also add these explanations in the revised version of our manuscript.
>
> From the perspective of classical optimization, nonconvex optimization arises naturally when we study the landscape of the loss functions for training neural networks with a bounded number of data samples. In practice, loss functions are usually nonconvex and nonsmooth. However, according to the PAC learning framework, the loss function is approximately smooth and its distance (i.e. the empirical noise level) from a smooth function is proportional to the inverse of the number of data samples. In particular, roughly $O(d/\epsilon^{1.5})$ samples can guarantee empirical risk of amplitude $O(\epsilon^{1.5}/d)$ for dimension $d$ and accuracy parameter $\epsilon$. By varying the dependence of the sample size on dimension $d$ and accuracy parameter $\epsilon$, we can generate empirical noise of different levels of $\text{poly}(\epsilon,1/d)$. Understanding the complexity of optimizing approximately smooth nonconvex functions of different noise rates of $\text{poly}(\epsilon,1/d)$ offers insights into the necessary data sample number for training neural networks.
>
> From the perspective of quantum computing, the realization of future quantum computers heavily depends on achieving fault-tolerant quantum computation. A standard approach involves concatenating multiple levels of quantum error-correcting codes. The number of concatenation levels typically scales poly-logarithmically with the target noise rate, leading to an inverse polynomial overhead in the noise rate. For a noise rate of order $\text{poly}(\epsilon, 1/d)$, achieving fault tolerance requires $\text{polylog}(\epsilon, 1/d)$ levels of concatenation. This overhead is generally considered practical. However, it can vary significantly depending on the specific noise level within this regime. Understanding the effects of different noise levels is crucial for optimizing the number of concatenation levels and reducing experimental resource requirements.
>
> **Presentation Quality:** We thank the reviewer for the valuable comment. We sincerely apologize for any confusion caused earlier and want to assure you that we are committed to thoroughly reviewing our definitions and foundational properties before finalizing our paper. We are also happy to make any further changes/clarifications regarding any such points - feel free to point out if you feel you encounter any.

---

> ### Author Response · Authors · 2024-12-04
>
> **Relevance of the Topic:** We thank the reviewer for suggesting us to clarify the significance of our work. We would like to point out that although no papers directly work on nonconvex optimization with empirical risk under exactly the same setting as this paper after 2018, this does not mean the decline of this problem area. In addition, the result of this paper is widely exploited in a vast number of relevant works in both theory and application. We would provide several directions that are related:
>
> 1. Classical algorithms for nonconvex optimization in recent years mainly extend to more advanced settings, and noise in classical oracles is less of an issue due to the power of classical computing these days. For instance, recent works have covered finding second-order stationary points of a nonconvex function with a zeroth-order (evaluation) oracle [1, 2, 3] or a first-order (gradient) oracle with high-order smoothness [4], and finding an approximate Goldstein point of a nonsmooth nonconvex function [5, 6].
>
> 2. Nonconvex optimization in the existence of noise due to empirical risk or other reasons arises naturally in training neural networks in practice. From the theoretical perspective, nonconvex optimization with noise under different settings has been investigated after this work [7, 8]. The result of this work has been exploited to obtain guarantees for training neural networks with empirical risk under suitable assumptions (e.g. [9] and following-up works).
>
> 3. Quantum algorithms for optimization constitute a relatively new topic, for instance, the paper by Chakrabarti et al. [10] and van Apeldoorn et al. [11] on convex optimization was formally published in 2020, these were followed by a series of works [12, 13, 14, 15] on nonconvex optimization in recent two years. It is worth noting, however, that these works generally assume a perfect quantum evaluation oracle. The robustness of quantum optimization algorithms remains a largely open question, especially given that quantum systems are inherently more prone to noise compared to their classical counterparts.
>
> Therefore, from the perspective of quantum computing, we deem our result as a natural extension of research on quantum algorithms for optimization. In any case, we are happy to add more relevant discussions when finalizing our paper.
>
> [1]. Hualin Zhang, Huan Xiong, and Bin Gu. Zeroth-order negative curvature finding: Escaping saddle points without gradients. NeurIPS 2022.
>
> [2]. Hualin Zhang, and Bin Gu. Faster gradient-free methods for escaping saddle points. ICLR 2022.
>
> [3]. Zhaolin Ren, Yujie Tang, and Na Li. Escaping saddle points in zeroth-order optimization: the power of two-point estimators. ICML 2023.
>
> [4]. Nuozhou Wang, Junyu Zhang, and Shuzhong Zhang. Efficient first order method for saddle point problems with higher order smoothness. SIAM Journal on Optimization 2024.
>
> [5]. Tianyi Lin, Zeyu Zheng, and Michael Jordan. Gradient-free methods for deterministic and stochastic nonsmooth nonconvex optimization. NeurIPS 2022.
>
> [6]. Lesi Chen, Jing Xu, and Luo Luo. Faster gradient-free algorithms for nonsmooth nonconvex stochastic optimization. ICML 2023.
>
> [7]. Mo Zhou, Tianyi Liu, Yan Li, Dachao Lin, Enlu Zhou, and Tuo Zhao. Toward Understanding the Importance of Noise in Training Neural Networks. ICML 2019.
>
> [8]. Yin Tat Lee, Zhao Song, and Qiuyi Zhang. Solving Empirical Risk Minimization in the Current Matrix Multiplication Time. COLT 2019.
>
> [9]. Samet Oymak and Mahdi Soltanolkotabi. Toward Moderate Overparameterization: Global Convergence Guarantees for Training Shallow Neural Networks. IEEE Journal on Selected Areas in Information Theory 2020.
>
> [10]. Shouvanik Chakrabarti, Andrew M. Childs, Tongyang Li, and Xiaodi Wu. Quantum algorithms and lower bounds for convex optimization. Quantum 2020.
>
> [11]. Joran van Apeldoorn, András Gilyén, Sander Gribling, and Ronald de Wolf. Convex optimization using quantum oracles. Quantum 2020.
>
> [12]. Chenyi Zhang, and Tongyang Li. Quantum lower bounds for finding stationary points of nonconvex functions. ICML 2023.
>
> [13]. Yizhou Liu, Weijie J. Su, and Tongyang Li. On quantum speedups for nonconvex optimization via quantum tunneling walks. Quantum 2023.
>
> [14]. Jiaqi Leng, Ethan Hickman, Joseph Li, and Xiaodi Wu. Quantum Hamiltonian Descent. Bulletin of the American Physical Society 2023.
>
> [15]. Aaron Sidford, and Chenyi Zhang. Quantum speedups for stochastic optimization. NeurIPS 2024.

---

### Official Review · Reviewer_kwBc · 2024-11-02

**Soundness:** 4
**Presentation:** 4
**Contribution:** 2
**Rating:** 6
**Confidence:** 3

**Summary:**

The work studies the task of finding $\epsilon$-approximate second-order stationary points of $d$-dimensional non-convex functions with zeroth- or first-order noisy quantum oracles. Under various different noise regimes (as functions of $\epsilon$ and $d$), the work presents upper and lower bounds for the classical and the quantum query complexities of the task.
With zeroth order oracle, it has been shown that the quantum algorithm proposed gives an exponential speedup (against classic lower bound) when the noise is at a small scale of $\tilde O( \epsilon^6 / d^4 )$, and a polynomial speedup (compared to the state-of-the-art classical algorithm) when the noise is of scale $[ \tilde \Omega(\epsilon^6 / d^4), O(\epsilon^{1.5} / d) ] $.
With first-order oracle, there is a polynomial quantum speedup (compared to the state-of-the-art classical algorithm) when the noise is of scale $\Theta(\epsilon / d^{0.5})$.

**Strengths:**

The paper provides a comprehensive set of results that cover a wide range of noise scale regimes, and oracle assumptions. The comparison with the classical results are meaningful and well presented.

**Weaknesses:**

The quantum query complexity bounds established are far from being tight.
Most of the algorithms are almost identical to the classical version with the gradient estimation replaced by Jordan’s algorithm. The analysis also looks like a rehash of the classical proofs by substituting the guarantees from Jordan’s algorithm. The lower bound constructions seem to be just borrowed from Jin et al., 2018a and Carmon et al., 2021 (up to a unitary rotation and a scaling factor).

**Questions:**

1. Can you give a more descriptive title to Section 4.3 like the surrounding sections?
2. Table 3 is for lower bounds in $\epsilon$. But the zeroth-order lower bound for deterministic classical algorithm entry is still an expression in $d$.

---

> ### Author Response · Authors · 2024-11-21
>
> We appreciate the reviewer for the great effort and valuable suggestions.
>
> The reviewer raised the question of the novelty of our algorithms and lower bounds. Regarding the comment on the novelty of our quantum algorithms, we agree with the reviewer that in the tiny and small noise regime, our algorithm replaces the classical gradient estimation using Jordan’s algorithm as quantum gradient estimation to obtain the exponential speedup. However, we would like to point out that in the intermediate noise regime, we employ a recent result of quantum mean estimation to estimate the gradient of the Gaussian smoothing of $f$ and prove a polynomial speedup. Although there are previous examples of implementing quantum mean estimation in the optimization setting to obtain polynomial speedups, this is the first example of using this technique in nonconvex optimization. In addition, we have to carefully control the variance of the stochastic gradient to use the quantum mean estimation technique.
>
> Regarding the comment on the novelty of our quantum lower bound results, we would like to highlight that, while our high-level intuition follows Jin et al., 2018a and Carmon et al., 2021, the construction details diverge significantly, and the analysis is highly technical. For our lower bound in Theorem 3.4, the lower bound construction in Jin et al., 2018a cannot be applied directly given that the input space is continuous and has infinite size. To address this, we introduced a new discretized version of the problem defined on a $\delta$-net of the unit hypercube and established a reduction from this new problem to the problem of finding an $\epsilon$-SOSP. For our lower bound in Theorem 3.5, there are two key distinctions between our approach and Carmon et al., 2021. Firstly, the presence of noise in our construction enables a non-informative region, which is central to our analysis and is not present in Carmon et al., 2021. Secondly, given that the noise is bounded, we need a rescaled version of the hard instance in Carmon et al., 2021 with a sine function as the regularizer. To the best of our knowledge, this different regularizer has not been used in previous works on zero-chain methods. The analysis of the non-informative region and the rescaling step involves significant technical challenges.
>
> We agree that a more descriptive title of Section 4.3 would benefit the presentation. Accordingly, we have updated the title in the revised version.
>
> We appreciate the reviewer for pointing out the slight mismatch between the caption of Table 3 and its content. We have resolved this issue in our revised version.

---

### Official Review · Reviewer_pk5B · 2024-11-04

**Soundness:** 4
**Presentation:** 4
**Contribution:** 3
**Rating:** 8
**Confidence:** 2

**Summary:**

This paper investigates the robustness of quantum algorithms for non-convex optimization, in the context of finding an $\epsilon$-approximate second-order stationary point ($\epsilon$-SOSP) in a d-dimensional non-convex function. As is usually done in optimization literature, the algorithm has access to the function via noisy zeroth or first-order oracles that satisfy certain standard assumptions.

The paper provides several foundational results for this setup in both upper and lower bounds. The upper bounds, under zeroth and first-order oracles, respectively, are nicely summarized in Table 1 and Table 2 under different noise regimes. Notably, under the zeroth order assumption and low noise regimes, the algorithms achieve exponential speed-ups. Additionally, they establish lower bounds for classical and quantum algorithms, summarized in Tables 1, 2, and 3, under various noise regimes and Zeroth and First order Assumptions.

**Strengths:**

The paper provides a comprehensive analysis of upper and lower bounds on query complexities of quantum algorithms for an important problem in non-convex optimization, and thus, the contributions are important to the community. The paper is well-written.

**Weaknesses:**

No major weaknesses. See questions.

**Questions:**

1. In the regime of Table 2 and Theorem 2.5 under the first-order assumption, are there known classical lower bounds?
2. What are the key algorithmic insights for results that achieve exponential speed-ups?
3. In the final version, I will encourage the author to expand on open questions more systematically. In general, there are several open gaps to establish minimax optimal query complexities for the problem.

---

> ### Author Response · Authors · 2024-11-21
>
> We thank the reviewer for the positive opinions about our paper and valuable suggestions. In particular, we appreciate that the reviewer judged our work “The paper provides a comprehensive analysis of upper and lower bounds on query complexities of quantum algorithms for an important problem in non-convex optimization, and thus, the contributions are important to the community. The paper is well-written.”
>
> To address the question on classical lower bounds in the regime of Table 2 and Theorem 2.5 under the first-order assumption, we have listed all the relevant classical lower bounds in Table 2. To the best of our knowledge, there is no existing literature that precisely matches the regime considered in Theorem 2.5 under the first-order assumption. We believe this is an interesting open question for future works.
>
> We also provide further algorithmic insights for results that achieve exponential speed-ups below Theorem 2.2 (Theorem 1.4 in the supplementary material) in our revised version. In particular, while a polynomial number of queries in dimension $d$ is required for classical algorithms to compute the gradient using zeroth-order oracles, quantum zeroth-order queries enable querying function values at different positions in parallel due to quantum superposition. As a result, Jordan’s algorithm for quantum gradient estimation computes the gradient using exponentially fewer queries compared to classical counterparts, and such speedup is robust to noise up to $O(\epsilon^5/d^4)$.
>
> We thank the reviewer for raising the helpful suggestion to expand on open questions more systematically. In our revised version, we have updated the open questions section, including the classical (quantum) query lower bounds under the first-order assumption and additional entries on the minimax optimal quantum query complexities for the problem as mentioned by the reviewer.

---

### Comment · Area_Chair_pNg2 · 2024-11-26
**Response**

Dear Reviewers,

The authors have provided their rebuttal to your questions/comments. It will be very helpful if you can take a look at their responses and provide any further comments/updated review, if you have not already done so.

Thanks!

---

### Meta-Review · Area_Chair_pNg2 · 2024-12-20

**Metareview:**

This is an interesting paper extensively studying the the complexity of finding an $\epsilon$-approximation of a second-order stationary point of a non-convex objective function via noisy zeroth-order and gradient queries. A few quantum poly(d,epsilon) algorithms were provided, and also a lower bound.

Overall the reviewers are positive about the paper. I recommend acceptance.

**Additional Comments On Reviewer Discussion:**

The author response were precise and helpful. There were no change in scores after discussion.

---

### Decision · Program_Chairs · 2025-01-22

Accept (Poster)